# Lack of CCDC146, a ubiquitous centriole and microtubule-associated protein, leads to non-syndromic male infertility in human and mouse

Jana Muroňová[1,2,3†], Zine Eddine Kherraf[1,2,3,4†], Elsa Giordani[1,2,3], Emeline Lambert[1,2,3], Simon Eckert[5], Caroline Cazin[1,2,3,4], Amir Amiri-Yekta[1,2,3,6], Magali Court[1,2,3], Geneviève Chevalier[1,2,3], Guillaume Martinez[1,2,3,7], Yasmine Neirijnck[8], Francoise Kühne[8], Lydia Wehrli[8], Nikolai Klena[9‡], Virginie Hamel[9], Lisa De Macedo[1,2,3], Jessica Escoffier[1,2,3], Paul Guichard[9], Charles Coutton[1,2,3,7], Selima Fourati Ben Mustapha[10], Mahmoud Kharouf[10], Anne-Pacale Bouin[1,2,3], Raoudha Zouari[10], Nicolas Thierry-Mieg[11], Serge Nef[8], Stefan Geimer[5], Corinne Loeuillet[1,2,3], Pierre F Ray[1,2,3,4]*, Christophe Arnoult[1,2,3]*

[1]Institute for Advanced Biosciences (IAB), INSERM 1209, Grenoble, France; [2]Institute for Advanced Biosciences (IAB), CNRS UMR 5309, Grenoble, France; [3]Institute for Advanced Biosciences (IAB), Université Grenoble Alpes, Grenoble, France; [4]UM GI-DPI, CHU Grenoble Alpes, Grenoble, France; [5]Cell Biology/ Electron Microscopy, University of Bayreuth, Bayreuth, Germany; [6]Department of Genetics, Reproductive Biomedicine Research Center, Royan Institute for Reproductive Biomedicine, ACECR, Tehran, Islamic Republic of Iran; [7]UM de Génétique Chromosomique, Hôpital Couple-Enfant, CHU Grenoble Alpes, Grenoble, France; [8]Department of Genetic Medicine and Development, University of Geneva Medical School, Geneva, Switzerland; [9]University of Geneva, Department of Molecular and Cellular Biology, Sciences III, Geneva, Switzerland; [10]Polyclinique les Jasmins, Centre d'Aide Médicale à la Procréation, Centre Urbain Nord, Tunis, Tunisia; [11]Laboratoire TIMC/MAGe, CNRS UMR 5525, Pavillon Taillefer, Faculté de Medecine, La Tronche, France

*For correspondence:
pray@chu-grenoble.fr (PFR);
christophe.arnoult@univ-grenoble-alpes.fr (CA)

†These authors contributed equally to this work

Present address: ‡Human Technopole, Milan, Italy

**Abstract** From a cohort of 167 infertile patients suffering from multiple morphological abnormalities of the flagellum (MMAF), pathogenic bi-allelic mutations were identified in the *CCDC146* gene. In somatic cells, CCDC146 is located at the centrosome and at multiple microtubule-related organelles during mitotic division, suggesting that it is a microtubule-associated protein (MAP). To decipher the molecular pathogenesis of infertility associated with *CCDC146* mutations, a *Ccdc146* knock-out (KO) mouse line was created. KO male mice were infertile, and sperm exhibited a phenotype identical to CCDC146 mutated patients. CCDC146 expression starts during late spermiogenesis. In the spermatozoon, the protein is conserved but is not localized to centrioles, unlike in somatic cells, rather it is present in the axoneme at the level of microtubule doublets. Expansion microscopy associated with the use of the detergent sarkosyl to solubilize microtubule doublets suggests that the protein may be a microtubule inner protein (MIP). At the subcellular level, the absence of CCDC146 impacted all microtubule-based organelles such as the manchette, the head–tail coupling apparatus (HTCA), and the axoneme. Through this study, a new genetic cause of infertility and a new factor in the formation and/or structure of the sperm axoneme were characterized.

## eLife assessment

This study presents **valuable** information that demonstrates CCDC146 as a novel cause of male infertility that play key role in microtubule-associated structures. The evidence supporting the claims of the authors is **solid** using a combination of human and mouse genetics, biochemical and imaging approaches. This article would be of interest to cell and developmental biologists working on genes involved in spermatogenesis and male infertility.

## Introduction

Infertility is a major health concern, affecting approximately 50 million couples worldwide (*Boivin et al., 2007*), or 12.5% of women and 10% of men. It is defined by the World Health Organization (WHO) as the "failure to achieve a pregnancy after 12 months or more of regular unprotected sexual intercourse". In almost all countries, infertile couples have access to assisted reproductive technology (ART) to try to conceive a baby, and there are now 5 million people born as a result of ART. Despite this success, almost half of the couples seeking medical support for infertility fail to successfully conceive and bear a child by the end of their medical care. The main reason for these failures is that one member of the couple produces gametes that are unable to support fertilization and/or embryonic development. Indeed, ART does not specifically treat or even try to elucidate the underlying causes of a couple's infertility, rather it tries to bypass the observed defects. Consequently, when defects in the gametes cannot be circumvented by the techniques currently proposed, ART fails. To really treat infertility, a first step would be to gain a better understanding of the problems with gametogenesis for each patient. This type of approach should increase the likelihood of adopting the best strategy for affected patients, and if necessary, should guide the development of innovative therapies.

Male infertility has several causes, such as infectious diseases, anatomical defects, or a genetic disorder, including chromosomal or single-gene deficiencies. Genetic defects play a major role in male infertility, with over 4000 genes thought to be involved in sperm production, of which more than 2000 are testis-enriched and almost exclusively involved in spermatogenesis (*Uhlén et al., 2016*). Mutations in any of these genes can negatively affect spermatogenesis and produce one of many described sperm disorders. The characterization and identification of the molecular bases of male infertility is thus a real challenge. Nevertheless, thanks to the emergence of massively parallel sequencing technologies, such as whole-exome sequencing (WES) and whole-genome sequencing (WGS), the identification of genetic defects has been greatly facilitated in recent years. As a consequence, remarkable progress has been made in the characterization of numerous human genetic diseases, including male infertility.

Today, more than 120 genes are associated with all types of male infertility (*Houston et al., 2021*), including quantitative and qualitative sperm defects. Qualitative spermatogenesis defects impacting sperm morphology, also known as 'teratozoospermia' (*Beurois et al., 2020*; *Touré et al., 2021*), are a heterogeneous group of abnormalities covering a wide range of sperm phenotypes. Among these phenotypes, some relate to the morphology of the flagellum. These defects are usually not uniform, and patients' sperm show a wide range of flagellar morphologies such as short and/or coiled and/or irregularly sized flagella. Due to this heterogeneity, this phenotype is now referred to as multiple morphological abnormalities of the sperm flagellum (MMAF) (*Touré et al., 2021*). Sperm from these patients are generally immotile, and patients are sterile.

Given the number of proteins present in the flagellum and necessary for its formation and functioning, many genes have already been linked to the MMAF phenotype. Study of the MMAF phenotype in humans has allowed the identification of around 50 genes (*Wang et al., 2022*) coding for proteins involved in axonemal organization, present in the structures surrounding the axoneme – such as the outer dense fibers and the fibrous sheath – and involved in intra-flagellar transport (IFT). Moreover, some genes have been identified from mouse models, and their human orthologs are very good gene candidates for MMAF, even if no patient has yet been identified with mutations in these genes. Finally, based on the remarkable structural similarity of the axonemal structure of motile cilia and flagella, some MMAF genes were initially identified in the context of primary ciliary dyskinesia (PCD). However, this structural similarity does not necessarily imply a molecular similarity, and only around half (10 of the 22 PCD-related genes identified so far; *Sironen et al., 2020*) are effectively associated

with male infertility. However, in most cases, the number of patients is very low and the details of the sperm tail phenotype are unknown (*Sironen et al., 2020*).

We have recruited 167 patients with MMAF. Following WES, biallelic deleterious variants in 22 genes were identified in 83 subjects. The genes identified are *AK7* (*Lorès et al., 2018*), *ARMC2* (*Coutton et al., 2019*), *CFAP206* (*Shen et al., 2021*), *CCDC34* (*Cong et al., 2022*), *CFAP251* (*Kherraf et al., 2018*), *CFAP43* and *CFAP44* (*Coutton et al., 2018*), *CFAP47* (*Liu et al., 2021a*), *CFAP61* (*Liu et al., 2021b*), *CFAP65* (*Li et al., 2020a*), *CFAP69* (*Dong et al., 2018*), *CFAP70* (*Beurois et al., 2019*), *CFAP91* (*Martinez et al., 2020*), *CFAP206* (*Shen et al., 2021*), *DNAH1* (*Ben Khelifa et al., 2014*), *DNAH8* (*Liu et al., 2020b*), *FSIP2* (*Martinez et al., 2018*), *IFT74* (*Lorès et al., 2021*), *QRICH2* (*Kherraf et al., 2019*), *SPEF2* (*Liu et al., 2020a*), *TTC21A* (*Liu et al., 2019*), and *TTC29* (*Lorès et al., 2019*). Despite this success, a molecular diagnosis is obtained in half of the patients (49.7%) with this sperm phenotype, suggesting that novel candidate genes remain to be identified. We have pursued our effort with this cohort to identify further mutations that could explain the patient MMAF phenotype. As such, we have identified bi-allelic truncating mutations in *CCDC146* in two unrelated infertile patients displaying MMAF. *CCDC146* is known to code for a centrosomal protein when heterologously expressed in HeLa cells (*Firat-Karalar et al., 2014*; *Almeida et al., 2018*), but minimal information is available on its distribution within the cell, or its function when naturally present. Moreover, this gene has never been associated with any human disease.

The centrosome, located adjacent to the nucleus, is a microtubule-based structure composed of a pair of orthogonally oriented centrioles surrounded by the pericentriolar material (PCM). The centrosome is the major microtubule-organizing center (MTOC) in animal cells, and as such regulates the microtubule organization within the cell. Therefore, it controls intracellular organization and intracellular transport, and consequently regulates cell shape, cell polarity, and cell migration. The centrosome is also crucial for cell division as it controls the assembly of the mitotic/meiotic spindle, ensuring correct segregation of sister chromatids in each of the daughter cells (*Blanco-Ameijeiras et al., 2022*). The importance of this organelle is highlighted by the fact that 3% (579 proteins) of all known human proteins have been experimentally detected in the centrosome (https://www.proteinatlas.org/human-proteome/subcellular/centrosome). Centrioles also play essential roles in spermatogenesis and particularly during spermiogenesis. In round spermatids, the centriole pair docks to the cell membrane, whereas the distal centriole serves as the basal body initiating assembly of the axoneme. The proximal centriole then tightly attaches to the sperm nucleus and gradually develops the head-to-tail coupling apparatus (HTCA), linking the sperm head to the flagellum (*Wu et al., 2020*). In human and bovine sperm, the proximal centriole is retained and the distal centriole is remodeled to produce an 'atypical' centriole *Fishman et al., 2018* in contrast, in rodents, both centrioles are degenerated during epididymal maturation (*Manandhar et al., 1998*). Despite the number of proteins making up the centrosome, and its importance in sperm differentiation and flagellum formation, very few centrosomal proteins have been linked to MMAF in humans – so far only CEP135 (*Sha et al., 2017*), CEP148 (*Zhang et al., 2022*), or DZIP1 (*Lv et al., 2020*). Moreover, some major axonemal proteins with an accessory location in the centrosome, such as CFAP58 (*Sha et al., 2021*) and ODF2 (*Zhu et al., 2022*; *Nakagawa et al., 2001*), have also been reported to be involved in MMAF syndrome. Other centrosomal proteins lead to MMAF in mice; these include CEP131 (*Hall et al., 2013*) and CCDC42 (*Pasek et al., 2016*). The discovery that MMAF in humans is linked to CCDC146, known so far as a centrosomal protein, adds to our knowledge of proteins important for axoneme biogenesis.

In this article, we first evaluated the localization of endogenous CCDC146 during the cell cycle in two types of cell cultures, immortalized HEK-293T cells and primary human foreskin fibroblasts. To validate the gene candidate and improve our knowledge of the corresponding protein, we also generated two mouse models. The first one was a *Ccdc146* KO model, with which we studied the impact of lack of the protein on the general phenotype, and in particular on male reproductive function using several optical and electronic microscopy techniques. The second model was a HA-tagged CCDC146 model, with which we studied the localization of the protein in different cell types. Data from these genetically modified mouse models were confirmed in human sperm cells.

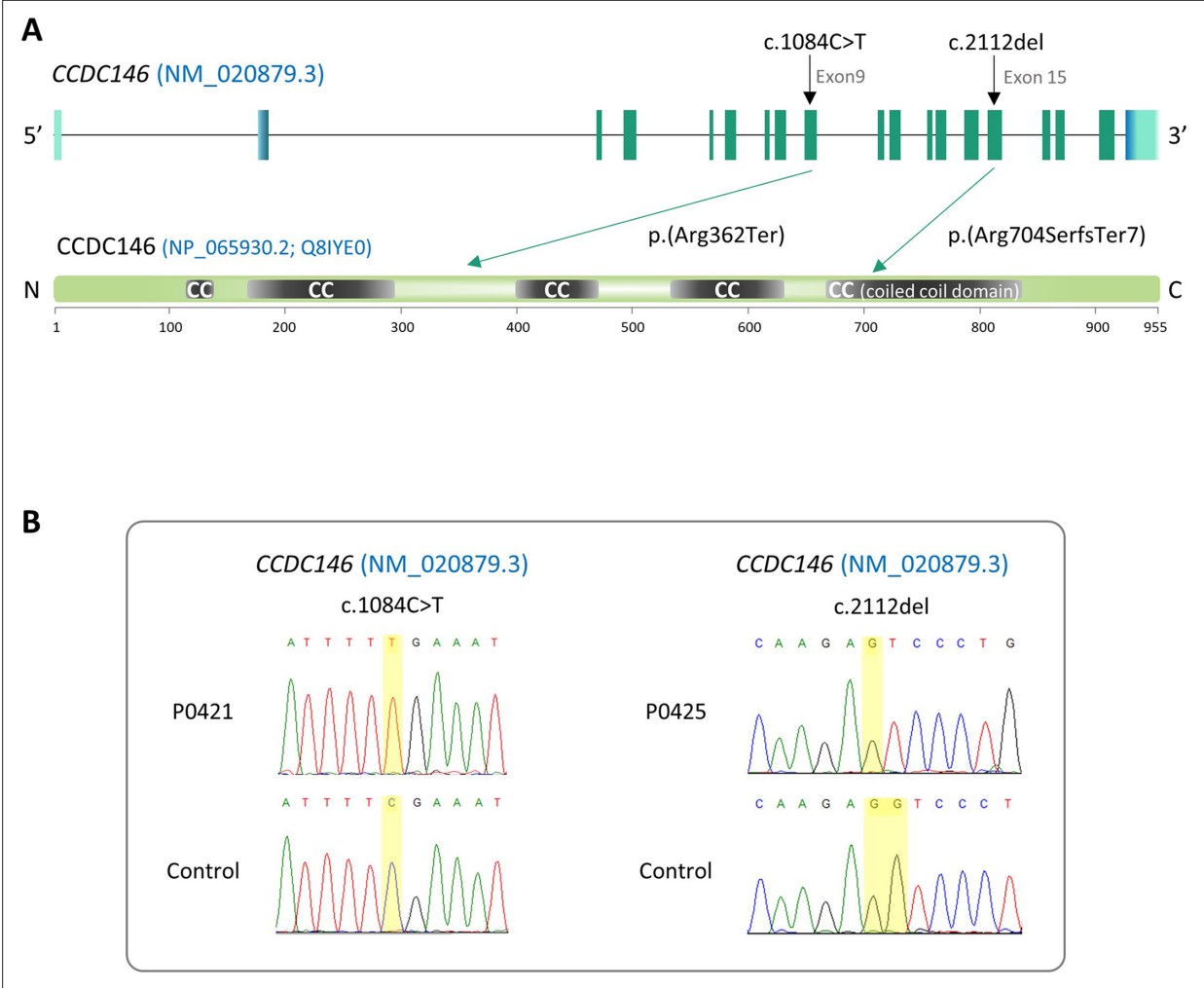

**Figure 1.** Identification of two *CCDC146* gene variants in multiple morphological abnormalities of the flagellum (MMAF) patients. (**A**) Structure of the canonical *CCDC146* gene transcript showing the position of the variants and their impact on translation. Variants are annotated according to HGVS recommendations. Position of the observed variants in both probands. (**B**) Electropherograms indicating the homozygous state of the identified variant: variant c.1084C>T is a nonsense mutation, and c.2112Del is a single-nucleotide deletion predicted to induce a translational frameshift.

The online version of this article includes the following figure supplement(s) for figure 1:

**Figure supplement 1.** Relative mRNA expression levels for human and mouse *Ccdc146* transcripts.

## Results

### WES identifies *CCDC146* as a gene involved in MMAF

We performed WES to investigate a highly selected cohort of 167 MMAF patients previously described in *Coutton et al., 2019*. The WES data was analyzed using an open-source bioinformatics pipeline developed in-house, as previously described (*Kherraf et al., 2022*). From these data, we identified two patients with homozygous truncating variants in the *CCDC146* (coiled-coil domain containing 146) gene, NM_020879.3 (*Figure 1A*), which contains 19 exons in human. Sperm parameters of both patients are presented in *Table 1*. No other candidate variants reported to be associated with male infertility were detected in these patients. Despite an ubiquitous expression, this gene is highly transcribed in human testes (*Figure 1—figure supplement 1A*). The first identified mutation is located in exon 9 and corresponds to c.1084C>T, the second is located in exon 15 and corresponds to c.2112Del (*Figure 1A*). The c.1084C>T variant is a nonsense mutation, whereas the single-nucleotide deletion c.2112Del is predicted to induce a translational frameshift. Both mutations were predicted to produce premature stop codons: p.(Arg362Ter) and p.(Arg704serfsTer7), respectively, leading either to the complete absence of the protein or to the production

**Table 1.** Detailed semen parameters of both multiple morphological abnormalities of the flagellum (MMAF) individuals harboring a *CCDC146* variant.

Values are percentages unless specified otherwise. NA: not available, Reference limits (5th centiles and their 95% confidence intervals) according to World Health Organization (WHO) standards (*Cooper et al., 2010*). Patient 08IF39 variant c.1084C>T and patient O9IF26 variant c.2112Del. Green and red colors indicate normal and abnormal values, respectively.

| | Patient # 08IF039 | Patient # 09IF026 | Lower reference limits (WHO) |
|---|---|---|---|
| Age (years) | 48 | 52 | |
| Semen volume (mL) | 5.5 | 2 | 1.5 |
| pH | NA | 7.7 | ≥7.2 |
| Viscosity (normal: 1; abnormal: 2) | 2 | 1 | |
| Sperm concentration (million/mL) | 35 | 42 | 15 |
| Total sperm number (million/ejaculate) | 192.5 | 84 | 39 |
| Non sperm cells/round cells (million/mL) | 3.2 | 1 | |
| Polynuclear neutrophils | NA | 0 | |
| Total motility (a + b + c) after 1 hr (%) | 15 | 10 | 40 |
| Progressive motility (PR; a + b) | 10 | 7 | 32 |
| Vitality (%) | 42 | 62 | 58 |
| Sperm morphology: normal forms (%) | 1 | 0 | 4 |
| Multiple anomalies index (MAI) | NA | 2.4 | – |
| Tapered head | NA | 5 | – |
| Thin head | NA | 2 | – |
| Microcephalic | NA | 6 | – |
| Macrocephalic | NA | 2 | – |
| Multiple heads | NA | 0 | – |
| Abnormal base (abnormal post-acrosomal region) | NA | 6 | – |
| Abnormal acrosomal region | NA | 34 | – |
| Excess residual cytoplasm | NA | 11 | – |
| Thin midpiece | NA | 3 | – |
| Bent or misaligned tail | NA | NA | |
| No tail | >10% | 35 | – |
| Short tail | >10% | 37 | – |
| Irregularly shaped tail | >10% | 67 | – |
| Coiled tail | >10% | 18 | – |
| Multiple tails | >10% | 3 | – |

of a truncated and non-functional protein. These variants are annotated with a high impact on the protein structure (MoBiDiC prioritization algorithm [MPA] score = 10) (*Yauy et al., 2018*). The two mutations are therefore most likely deleterious. Both variants were absent in our control cohort and their minor allele frequencies (MAF), according to the gnomAD v3 database, were $6.984 \times 10^{-5}$ and $6.5 \times 10^{-06}$, respectively. The presence of these variants and their homozygous state were verified by Sanger sequencing, as illustrated in *Figure 1B*. Taken together, these elements strongly suggest that mutations in the *CCDC146* gene could be responsible for the infertility of these two patients and the MMAF phenotype.

## *Ccdc146* knock-out mouse model confirms that lack of CCDC146 is associated with MMAF

Despite a lower expression in mouse testes compared to human (the level of expression is only medium, *Figure 1—figure supplement 1B*), we produced by CRISPR/Cas9 two mouse lines carrying each a frameshift mutation in *Ccdc146* (ENSMUST00000115245). One line (line 1) has a deletion of 4 bp (*Figure 2—figure supplement 1A and B*) and the other (line 2) an insertion of 250 bp, both in exon 2. We used these lines to address the hypothesis that CCDC146 deficiency leads to MMAF and male infertility. We analyzed the reproductive phenotype of the gene-edited mice from the F2 generation and found that homozygous males reproduced the MMAF phenotype, like the two patients carrying the homozygous variants in the orthologous gene (*Figure 2* for line 1 and *Figure 2—figure supplement 1C and D* for line 2). Based on these findings, we restricted our study to a strain with a 4 bp deletion in exon 2 (c.164_167delTTCG).

The *Ccdc146* KO mice were viable without apparent defects. The reproductive phenotypes of male and female mice were explored. WT or heterozygous animals and KO females were fertile, whereas KO males were completely infertile (*Figure 2A*). This infertility is associated with a 90% decrease in epididymal sperm concentration (from ~30 to ~3 million) (*Figure 2B*) and a significant decrease of testicle weight relative to whole body weight (*Figure 2C*), suggesting a germ cell rarefaction in the seminiferous epithelia due to high apoptosis level. A study of spermatogenic cell viability by TUNEL assay confirmed this hypothesis, with a significant increase in the number of fluorescent cells in *Ccdc146* KO animals (*Figure 2—figure supplement 2*). Closer examination revealed sperm morphology to be strongly altered, with a typical MMAF phenotype and marked defects in head morphology (*Figure 2D*) indicative of significantly impaired spermiogenesis. The percentage of abnormal form, flagellum and head anomalies is 100% (*Figure 2E*). Moreover, an almost complete absence of motility was observed (*Figure 2F*).

Comparative histological studies (*Figure 3*) showed that on sections of spermatogenic tubules, structural and shape defects were present from the elongating spermatid stage in *Ccdc146* KO mice, with almost complete disappearance of the flagella in the lumen and very long spermatid nuclei (*Figure 3A*). At the epididymal level, transverse sections of the epididymal tubules from KO males contained almost no spermatozoa, and the tubules were filled with an acellular substance (*Figure 3B*).

## *CCDC146* codes for a centriolar protein

CCDC146 has been described as a centriolar protein in immortalized HeLa cells (*Firat-Karalar et al., 2014*; *Almeida et al., 2018*). To confirm this localization, we performed immunofluorescence experiments (IF) (*Figure 4*). First, we validated the specificity of the anti-CCDC146 antibody (Ab) in HEK-293T cells by expressing DDK-tagged CCDC146. We observed a nice colocalization of the DDK and CCDC146 signals in both interphase and mitotic cells, showing that both Abs target the same protein (*Figure 4—figure supplement 1*), strongly suggesting that the anti-CCDC146 Ab is specific. Next, we focused on the centrosome (*Figure 4A*). In HEK-293T cells, using an antibody recognizing centrin (anti-centrin Ab) as a centriole marker and the anti-CCDC146 Ab, CCDC146 was shown to colocalize with centrioles. This colocalization strongly suggests that CCDC146 is a centriolar protein. However, the signal was not strictly localized to centrioles, as peri-centriolar labeling was clearly visible (*Figure 4A1d*, overlay). As this labeling pattern suggests the presence of centriolar satellite proteins, we next performed co-labeling with an antibody recognizing PCM1, a canonical centriolar satellite marker (*Odabasi et al., 2020*). Once again, the colocalization was only partial (*Figure 4—figure supplement 2*). Finally, the presence of CCDC146 was assessed at the basal body of primary cilia in cells cultured under serum-deprived conditions (*Figure 4B*). Similarly to centrosome staining, CCDC146 was observed on the basal body by also around it (*Figure 4B1d*). Based on these observations, CCDC146 has a unique localization profile in somatic cells that may indicate specific functions.

## CCDC146 co-localizes with multiple tubulin-based organelles

We next evaluated the presence of CCDC146 on centrioles during their duplication (*Figure 5A*). For this purpose, we synchronized HEK-293T cells with thymidine and nocodazole, and CCDC146 labeling was observed by IF. Both centrioles pairs were stained by CCDC146 Ab in cells blocked in the G2 phase (*Figure 5A*). Because our immunofluorescence experiments revealed that CCDC146 labeling was not strictly limited to centrosomes, we next assessed the presence of CCDC146 on

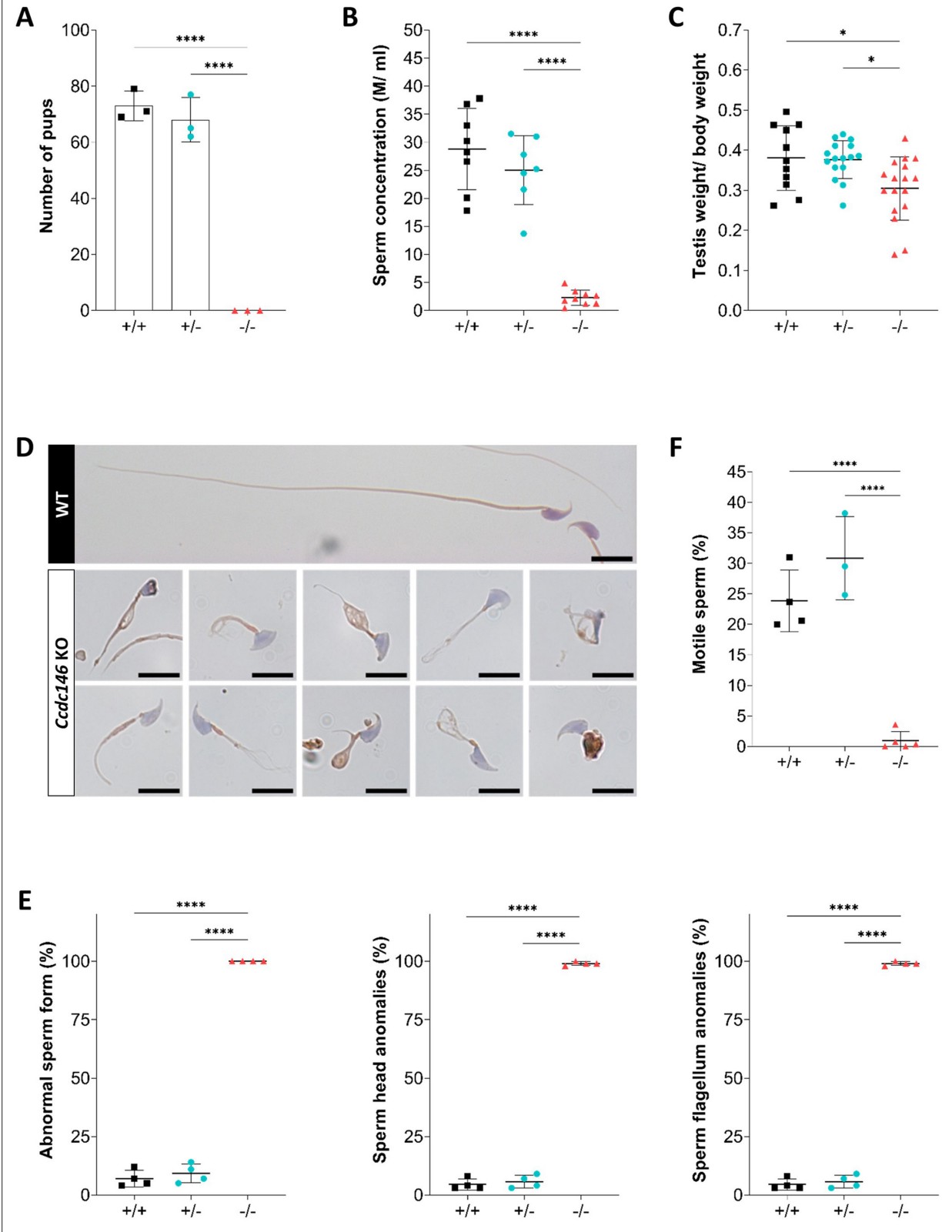

**Figure 2.** *Ccdc146* KO mice are infertile and KO sperm exhibit a typical multiple morphological abnormalities of the flagellum (MMAF) phenotype. (**A**) Number of pups produced by wild-type (+/+, WT), *CCDC146* heterozygote (+/-), and *Ccdc146* knock-out (-/-, KO) males (three males per genotype) after mating with fertile WT females (two females per male) over a period of 3 mo. (**B**) Sperm concentration and (**C**) comparison of testis weights (mg). (**D**) Illustration of WT and KO sperm morphologies stained with Papanicolaou and observed under optic microscopy. Scale bars of images represent

*Figure 2 continued on next page*

*Figure 2 continued*

10 µm. (**E**) Histograms showing proportions of total, head and flagella morphological anomalies (mean ± SD) for each *Ccdc146* genotype (n = 4). (**F**) sperm mobility. (**A–C, F**) Statistical comparisons were based on ordinary one-way ANOVA tests and (**E**) Statistical significance was assessed by applying an unpaired *t*-test;p-value of 0.05 or lower was considered statistically significant. ****p<0.0001; ***p<0.001, **p<0.01, *p<0.05. Error bars show standard deviation.

The online version of this article includes the following figure supplement(s) for figure 2:

**Figure supplement 1.** Molecular strategy used to generate *Ccdc146* KO mice by CRISPR/Cas9.

**Figure supplement 2.** Increased levels of apoptosis in testes from *Ccdc146* KO mice.

other tubulin-containing cellular substructures, particularly structures emerging during cell division (*Figure 5B–E*). In non-synchronized cells, the mitotic spindle was labeled at its base and at its ends (*Figure 5BC*). The co-labeling intensified in the midzone during chromatid separation (*Figure 5D*). Finally, the separation structure between the two cells, the midbody, was also strongly stained (*Figure 5E*). As HEK-293T cells are an immortalized cell line, we therefore verified that this labeling pattern was not due to an aberrant expression profile and that it also reflected the situation in primary cell lines. Identical labeling profiles were observed in freshly prepared human foreskin fibroblasts (HFF cells) (*Figure 5—figure supplement 1*).

## CCDC146 is present in epididymal sperm and its expression peaks at spermatid elongation stage during spermatogenesis

The presence of CCDC146 in the mature epididymal spermatozoa was assessed by WB (*Figure 6A*, *Figure 6—source data 1 and 2*). To overcome a lack of specific antibodies recognizing mouse CCDC146 (the commercial Ab works for human CCDC146 only), a mouse strain expressing a HA-CCDC146 knock-in (KI) was created. Because the functional domains of the protein are unknown, and to limit as such as possible the risk of interfering with the structure of the protein, we choose to insert the hemagglutinin (HA) tag, which contains only nine amino acids (YPYDVPDYA). HA sequence was inserted by the CRISPR/Cas9 system into the coding sequence of the *Ccdc146* gene between the two first codons to produce a tagged protein at the N-terminus domain (*Figure 6—figure supplement 1*). This insertion induced no phenotypic changes, and both female and male mice were viable with normal fertility. Interestingly, the protein was retained in epididymal sperm, where one band was observed around 120 kDa (*Figure 6A*). The theoretical MW of the tagged protein is around 116.2 kDa

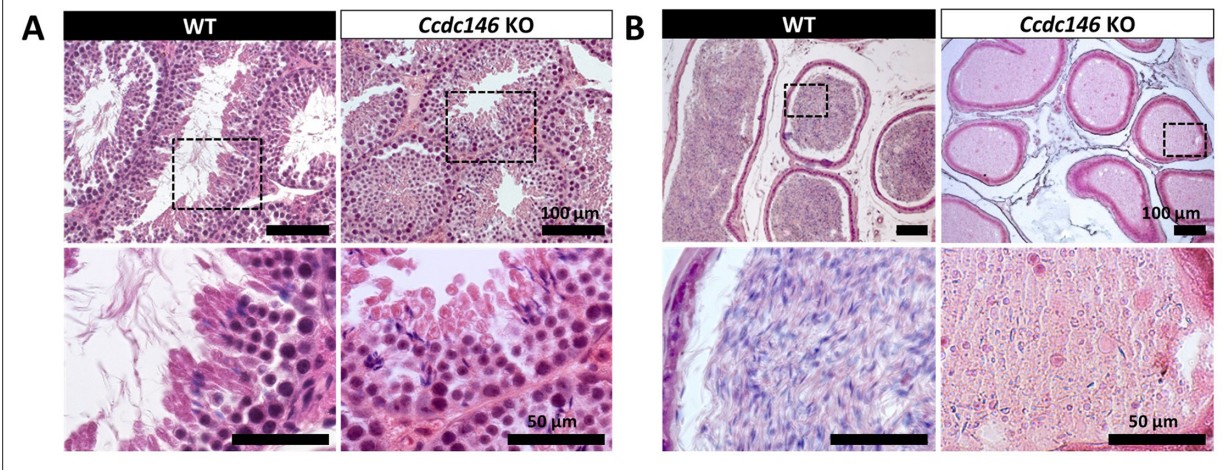

**Figure 3.** Histological evidence that spermiogenesis is disrupted in *Ccdc146* KO males and leads to a strong decrease in sperm concentration in the epididymis. (**A**) Transversal sections of WT and KO testes stained with hematoxylin and eosin. The upper images show the sections at low magnification (scale bars 100 µm) and the lower images are an enlargement of the dotted square (scale bars 50 µm). In the KO, spermatid nuclei were very elongated and remarkably thin (green arrow heads) and no flagella were visible within the seminiferous tubule lumen. (**B**) Transversal sections of WT and KO epididymides stained with hematoxylin and eosin. Despite similar epididymis section diameters in WT and KO testes, KO lumen were filled with round cells and contained few spermatozoa with abnormally shaped heads and flagella. The upper images show the sections at low magnification (scale bars 100 µm) and the lower images are an enlargement of the dotted square (scale bars 50 µm).

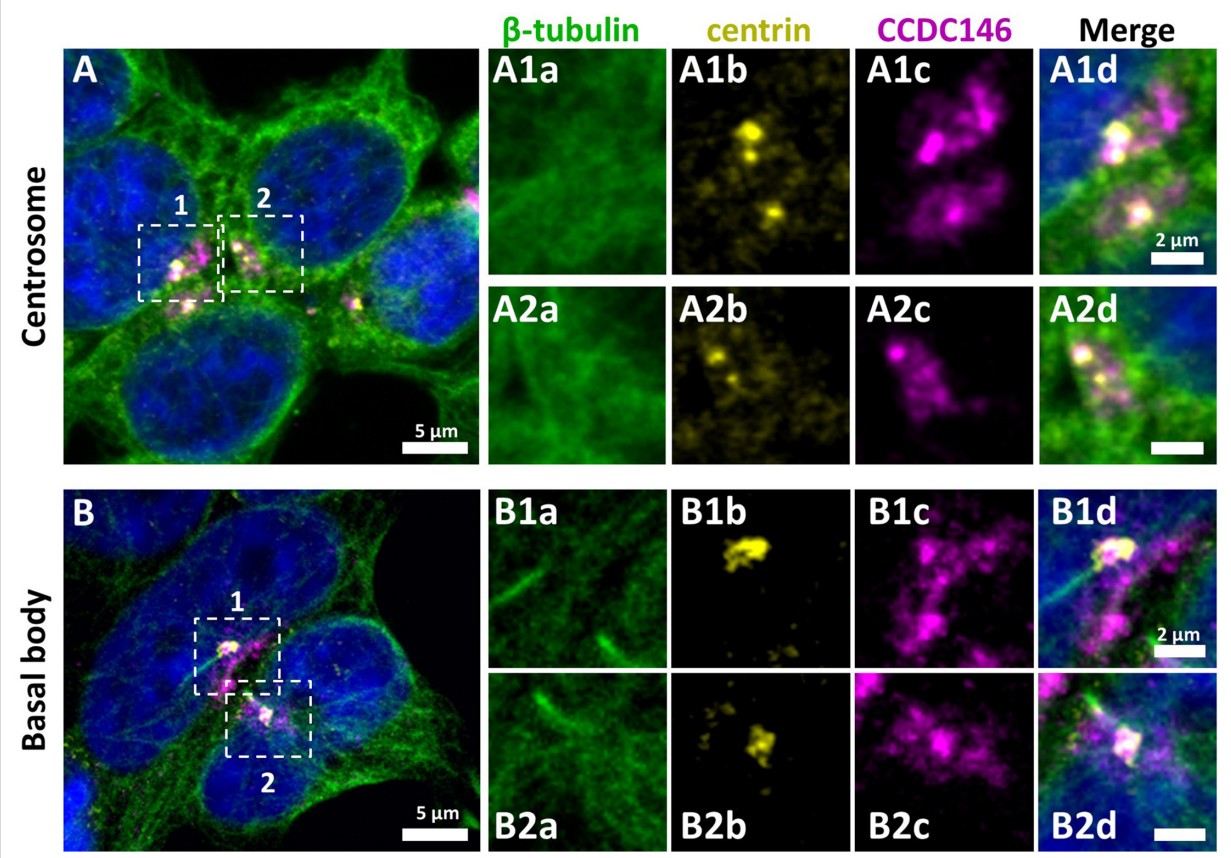

**Figure 4.** CCDC146 has a centriolar and pericentriolar localization in interphase somatic HEK-293T cells. HEK-293T cells were immunolabeled for β-tubulin (green), centrin (yellow), and CCDC146 (magenta). DNA was stained with Hoechst (blue). (**A**) CCDC146 localized to centrioles and/or to the pericentriolar material in interphase cells. The centrosome area of two cells is shown enlarged in (**A1**) and (**A2**). (**B**) In serum-starved cells with primary cilia, CCDC146 localized to the basal body of primary cilia. The basal body of two cells is shown enlarged in (**B1**) and (**B2**). CCDC146 was also present as dotted signal resembling the pattern for centriolar satellite proteins. Scale bars 5 μm and 2 μm on zoomed images.

The online version of this article includes the following figure supplement(s) for figure 4:

**Figure supplement 1.** DDK and CCDC146 Abs immunodecorate the same cellular components.

**Figure supplement 2.** CCDC146 does not colocalize with the centriolar satellite marker PCM1.

(115.1 + 1.1); the observed MW was slightly higher, suggesting that the protein migrates at a different weight or some post-translational modifications. The band is specific because it was observed in HA-CCDC146 sperm only and not in WT epididymal sperm. The protein is therefore present during spermatid differentiation and conserved in mature sperm. To better characterize the presence of CCDC146 in sperm, sperm flagella and heads were purified after mild sonication (*Figure 6B*), and protein extracts were analyzed by WB (*Figure 6A*). In the flagella fraction, the HA Ab revealed a band with an intensity similar to that of the whole sperm at around 120 kDa, whereas no staining was observed in head fraction.

From our immunofluorescence analysis of somatic cell lines, it was clear that CCDC146 expression is associated with the cell cycle. In the testis, a wide variety of cell types co-exist, including both somatic and germline cells. The germline cells can be further subcategorized into a wide variety of cells, some engaged in proliferation (spermatogonia), others in meiosis (spermatocytes), or in differentiation (spermatids). To better understand the role of CCDC146 in spermatogenesis, and thus how its absence leads to sperm death and malformation, we initially studied its expression during the first wave of spermatogenesis (*Zindy et al., 2001*). Results from this study should shed light on when CCDC146 is required for sperm formation. Expression of *Ccdc146* was normalized with two different housekeeping genes, *Actb* and *Hprt* (*Figure 6C*). Surprisingly, RT-PCR failed to detect *Ccdc146* transcripts on day 9 after birth in proliferating spermatogonia from HA-CCDC146 males. Transcription of

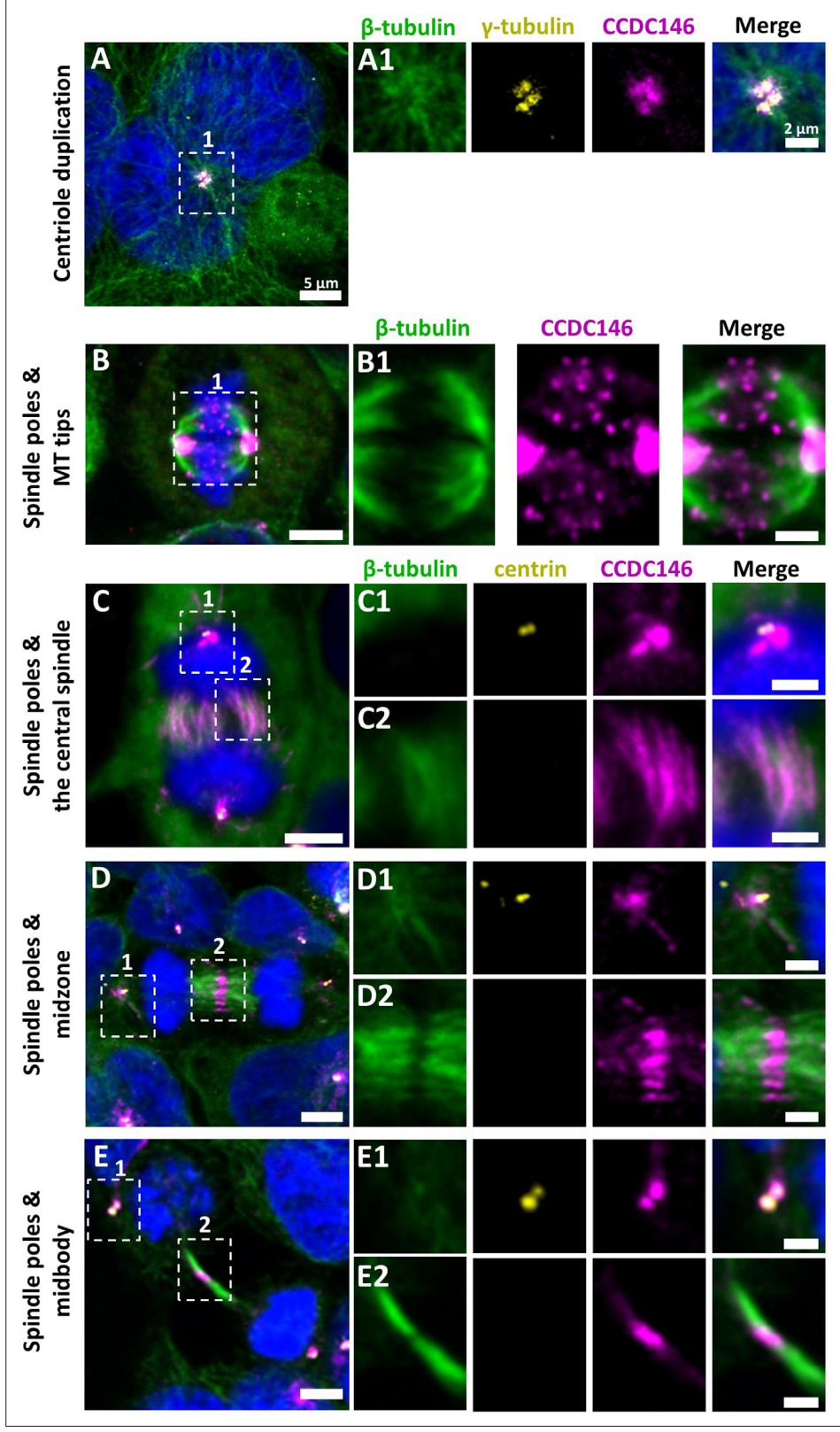

**Figure 5.** CCDC146 is a microtubule-associated protein (MAP) associating with microtubule-based structures throughout the cell cycle. HEK-293T cells were immunolabeled with anti-β-tubulin (panels **A–E**, green), anti-γ tubulin (yellow, panel **A**), anti-centrin (yellow panels **C–E**), and anti-CCDC146 (panels **A–E**, magenta) Abs. DNA was stained with Hoechst (blue). (**A**) In synchronized HEK-293T cells, CCDC146 is observed associated with mother

*Figure 5 continued on next page*

*Figure 5 continued*

centrioles and their corresponding procentrioles during centriole duplication. (**B–E**) In non-synchronized cells, CCDC146 is observed associated with spindle poles (**B–E**) and with microtubule (MT) tips during metaphase (**B**), with the central spindle during anaphase (**C**), with the midzone during telophase (**D**), and with the midbody during cytokinesis (**E**). Images on the right show the enlargement of the dotted square in the left image. Scale bars 5 μm and scale bars of zoomed insets 2 μm.

The online version of this article includes the following figure supplement(s) for figure 5:

**Figure supplement 1.** CCDC146 shows a similar localization to the centrosome and to the midbody in primary human foreskin fibroblast (HFF) cells.

*Ccdc146* started on day 18, concomitantly with the initiation of meiosis 2. Expression peaked on day 26, during the differentiation of spermatids.

## In sperm, CCDC146 is present in the flagellum, not in the centriole

To attempt to elucidate the function of CCDC146 in sperm cells, we next studied its localization by IF in mouse and human sperm. We focused successively on the anterior segment (head, neck, and beginning of the intermediate piece) of the sperm and then on the flagellum.

The IF experiments were first performed on murine epididymal spermatozoa. For this study, we used the mouse model expressing HA-tagged CCDC146 protein. In conventional IF, using an HA Tag Alexa Fluor 488-conjugated Antibody (anti-HA-AF488-C Ab), a punctiform labeling was observed along the whole flagellum (*Figure 7A*) on HA-CCDC146 only. We validated the specificity of the punctiform staining by performing a statistical comparison of the density of dots in the principal piece of WT and HA-CCDC146 sperm (*Figure 7B*). This study was carried out by analyzing 58 WT spermatozoa and 65 HA-CCDC146 spermatozoa coming from 3 WT and 3 KI males. We found a highly significant difference, with a p-value<0.0001, showing that the signal obtained on spermatozoa expressing the tagged protein is highly specific. To enhance resolution, ultrastructure expansion microscopy (U-ExM), an efficient method to study in detail the ultrastructure of organelles (*Gambarotto et al., 2019*) was used. We observed the same punctiform staining all along the axoneme, including the midpiece (*Figure 7C*), using another anti-HA Ab (#2). Unexpectedly, the mouse flagellum presented breaks that most likely resulted from the expansion procedure (*Figure 7D*). Interestingly, strong HA labeling was observed at the level of these breaks, suggesting that CCDC146 epitopes are buried inside the axonemal structure and become accessible mostly on blunt or broken microtubule doublets. The same pattern was observed with a third different anti-HA Ab (#3) (*Figure 7—figure supplement 1*).

In humans, the spermatozoon retains its two centrioles (*Fishman et al., 2018*), and they are observable in the neck, as shown by anti-centrin and anti-tubulin labeling (*Figure 8A1ac, A2ac, and A3ac*). No colocalization of the CCDC146 label with centrin was observed on human sperm centrioles (*Figure 8A1d, A2d, and A3d*), suggesting that the protein is not present in or around this structure. However, two unexpected labeling events were observed: sub-acrosomal labeling and labeling of the midpiece (*Figure 8A1b, A2b, and A3b*). At the flagellum level, faint staining was observed along the whole length (*Figure 8A4b*). To enhance resolution, U-ExM was also used (*Figure 8B*). The localization of the two centrioles was perfectly visible following anti-tubulin labeling and was confirmed by co-labeling with an anti-POC5 Ab (*Figure 8—figure supplement 1A*). Once again, no CCDC146 labeling was observed on sperm centrioles (*Figure 8B6*, merge), confirming the conventional IF results. Moreover, U-ExM unveiled that the observed CCDC146 midpiece staining (*Figure 8A*) seems in fact associated with isolated structures, which we hypothesize could be mitochondria, now visible by expansion (*Figure 8B1*). This suggests that the labeling on the midpiece observed in IF corresponds to non-specific mitochondrial labeling. This conclusion is supported by the fact that the same isolated structures were also labeled with anti-POC5 Ab (*Figure 8—figure supplement 1B*). In contrast, at the flagellum level, clear punctiform labeling was observed along the whole length of the principal piece (*Figure 8B1 and B5*), confirming that the protein is present in the sperm flagellum.

The study of CCDC146 localization was made in two species and with two different Abs. We observed slight differences in the CCDC146 staining between human and mouse. However, the antibodies do not target the same part of the CCDC146 protein (the tag is placed at the N-terminus of the protein, and the HPA020082 Ab targets the last 130 amino acids of the Cter), a difference that may change their accessibility to the antigenic site and explain the difference. Nevertheless, overall,

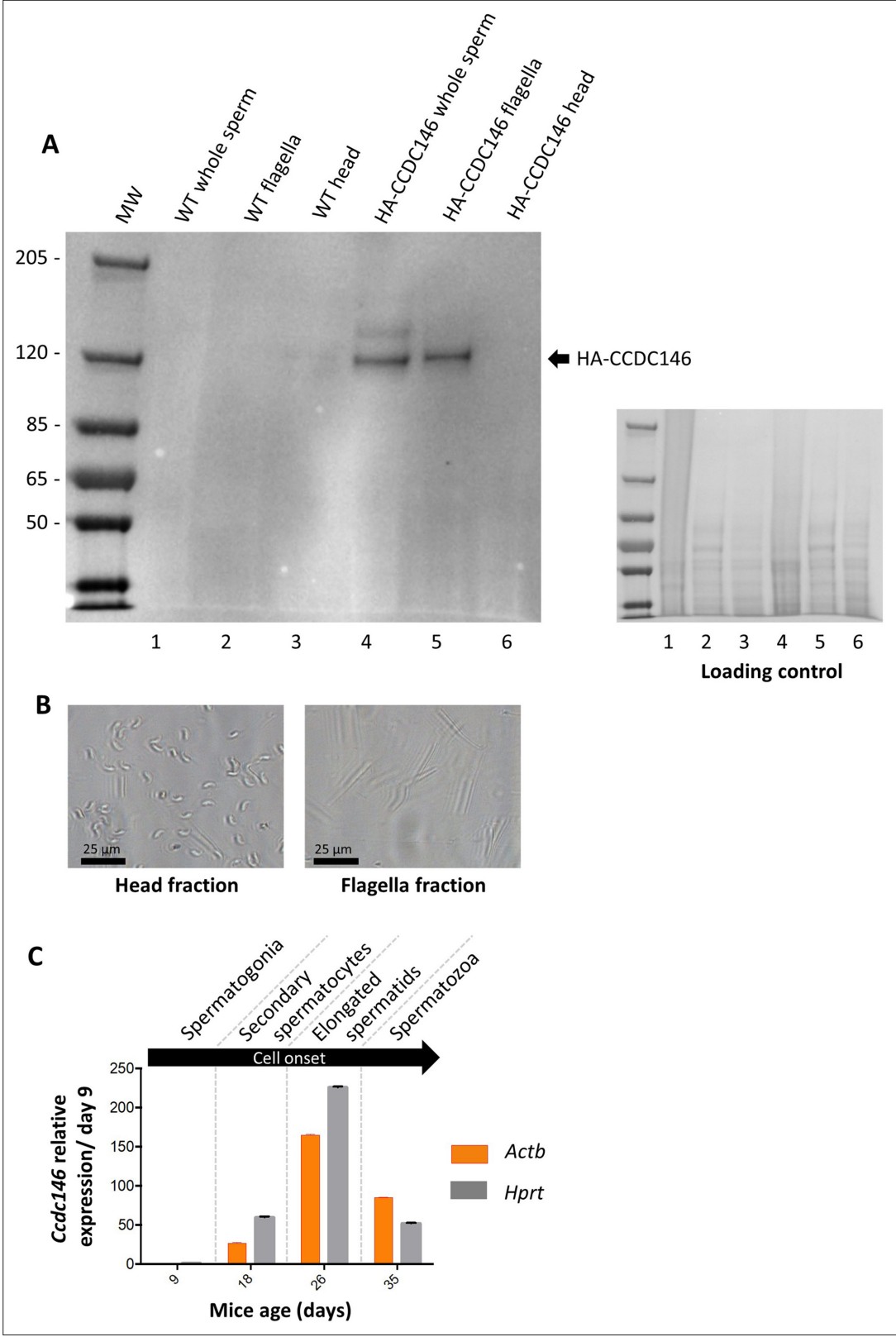

**Figure 6.** CCDC146 protein is present in epididymal spermatozoa in mouse and *Ccdc146* mRNA is expressed in late spermatocyte and in spermatids. (**A**) CCDC146 is retained in epididymal sperm. Western blot of HA-tagged epididymal whole sperm shows a band (arrow) corresponding to HA-tagged CCDC146 (HA-CCDC146), whereas no band was observed with WT epididymal sperm, demonstrating the specificity of the band observed

*Figure 6 continued on next page*

*Figure 6 continued*

in HA-CCDC146 sperm lane. Sperm extracts obtained from flagella and head fractions of WT and HA-CCDC146 were also analyzed and the presence of HA-CCDC146 was revealed by an anti-HA Ab in flagella fraction from HA-CCDC146 sperm only. Loading control (ponceau staining) of the gel is shown on the right. (**B**) Image of heads and flagella fractions. Scale bars as indicated (**C**) mRNA expression levels of *Ccdc146* relative to *Actb* and *Hprt* in HA-CCDC146 mouse pups' testes during the first wave of spermatogenesis (n = 3). Extremely low *Ccdc146* expression was detected at day 9, corresponding to testes containing spermatogonia and Sertoli cells only. *Ccdc146* expression was observed from postnatal day 18 (formation of secondary spermatocytes), peaked at day 26 (formation of elongated spermatids), and subsequently decreased from day 35 (formation of spermatozoa), suggesting that *Ccdc146* is particularly expressed in elongated spermatids during spermatogenesis. One-way ANOVA test was used for statistical comparisons. p-value of 0.05 or lower was considered statistically significant. ****p<0.0001; ***p<0.001; **p<0.01, *p<0.05. Error bars show standard deviation.

The online version of this article includes the following source data and figure supplement(s) for figure 6:

**Source data 1.** Uncropped gel of *Figure 6A*: whole sperm, flagella fraction and head fraction.

**Source data 2.** Uncropped gel of *Figure 6A*: loading control.

**Figure supplement 1.** Molecular strategy used to generate HA-tagged CCDC146 mice by CRISPR/Cas9.

the results are fairly consistent and both antibodies target the flagellum. It should be noted that unlike in humans, in mice, centrioles are no longer present in epididymal spermatozoa (*Manandhar et al., 1998*). Therefore, the WB result, showing the presence of the protein in sperm flagella fraction in mouse sperm, demonstrates unequivocally that the protein is located in the flagellum.

## CCDC146 labeling associates with microtubule doublets

We next wanted to determine whether CCDC146 staining was associated with the axoneme or accessory structures of the flagellum (outer dense fibers or fibrous sheath), and if yes, whether it was associated with microtubule doublets or the central pair. To do so, we used U-ExM on human sperm and quantified the relative position of each CCDC146 dot observed (outside the axoneme, outer left and right; microtubule doublets left and right and central pair – *Figure 9A–C*). The different parts of the flagellum were identified from the tubulin staining (*Figure 9B*). The distribution of localizations was summarized in a bar graph, generated for two distinct sperm cells. Labeling was preferentially located on the right and left doublets (*Figure 9D*), indicating that CCDC146 associates more with microtubule doublets. Analysis of protein distribution in mice was not easy because signal tended to concentrate at breaks. Nevertheless, using U-ExM, on some spermatozoa with frayed microtubule doublets, that is, with the flagellum taking on the shape of a hair, we could find that isolated doublets carried the punctiform labeling confirming the results of analyses on human spermatozoa (*Figure 9E*). Taken together, these results from mouse and human sperm demonstrate that CCDC146 is an axonemal protein, probably associated with microtubule doublets.

## CCDC146 is solubilized by the sarkosyl detergent

The presence of extensive labeling at axoneme breaks suggests that the antigenic site is difficult to access in an intact flagellum. We therefore hypothesized that CCDC146 could be a MIP. MIPs are generally resistant to solubilization by detergents. However, N-lauroylsarcosine (sarkosyl) can solubilize microtubule doublets, with increasing concentrations destabilizing first the A-tubule, then the B-tubule (*Yanagisawa et al., 2014*; *Witman et al., 1972*). Microtubule solubilization allows release of the MIPs contained within the tubules, and the method is recognized (*Linck, 1976*; *Kirima and Oiwa, 2018*). To test whether our hypothesis that CCDC146 may be a MIP, we treated spermatozoa from HA-tagged CCDC146 mice with sarkosyl and performed a WB on the supernatant (*Figure 10A*, *Figure 10—source data 1 and 2*). The protein was effectively solubilized, leading to the appearance of bands at around 120 kDa on an SDS-PAGE gel following migration of extracts from sperm treated with 0.2 and 0.4% sarkosyl concentrations. Interestingly, a second band around 90 kDa was also immunodecorated by anti-HA antibody in the sarkosyl treated sample, a band not present in the protein extracts from WT males. This band may correspond to proteolytic fragment of CCDC146, the solubilization of microtubules by sarkosyl may have made CCDC146 more accessible to endogenous proteases. The other detergents and buffers tested – RIPA, CHAPS, or Tris-HCl – barely solubilized CCDC146 or led to no solubilization (*Figure 10B*, *Figure 10—source data 3*). This result confirms the unique

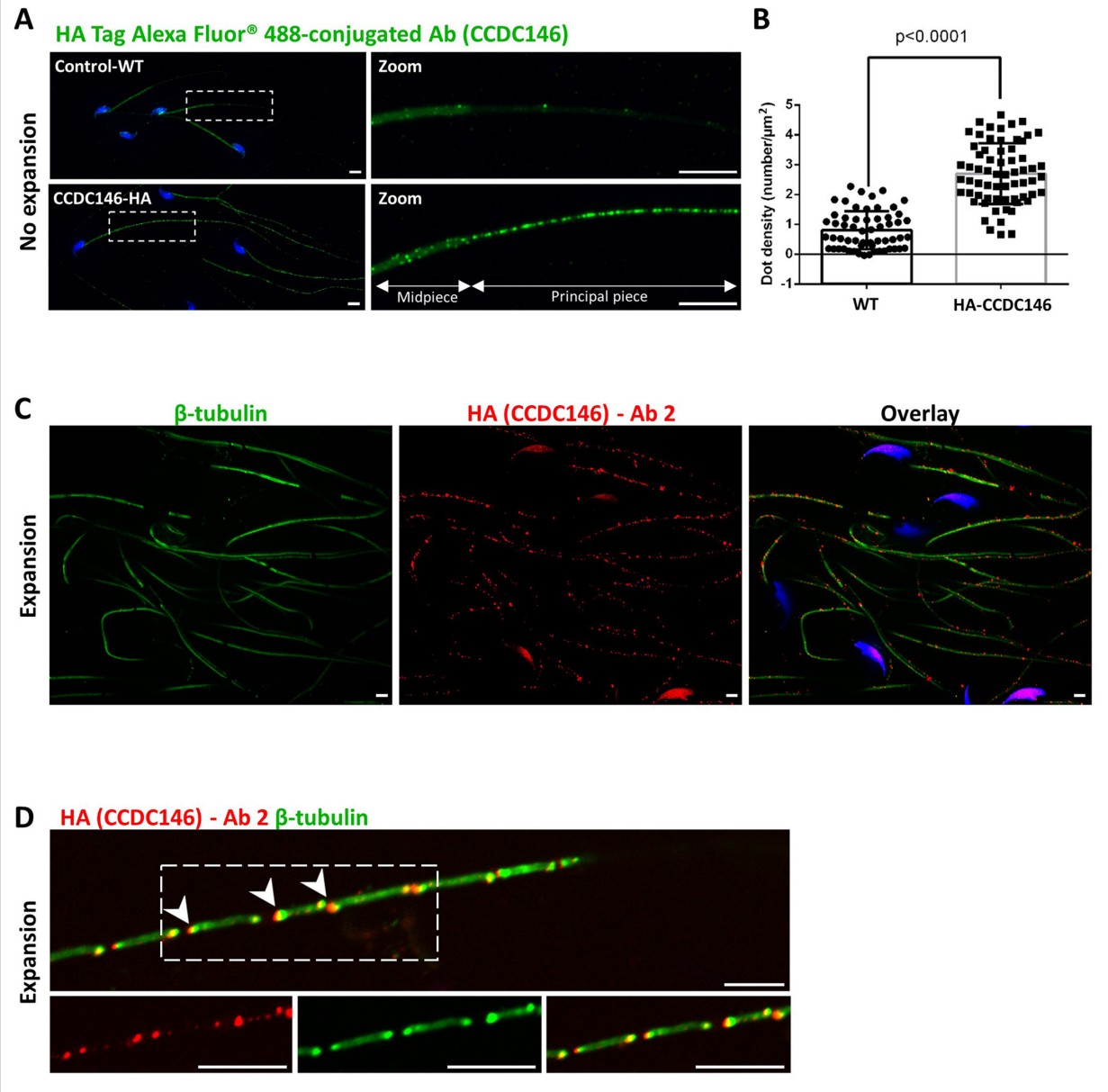

**Figure 7.** CCDC146 localizes to the flagellum of mouse epididymal spermatozoa. (**A**) Mouse epididymal spermatozoa observed with conventional IF. WT and HA-CCDC146 sperm were labeled with anti-HA Tag Alexa Fluor 488-conjugated 3 (green) Abs (anti-HA #1). DNA was stained with Hoechst (blue). The upper image shows the staining of a WT sperm and the lower images, the staining of a HA-CCDC146 sperm. A punctuate signal is observed in HA-CCDC146 sperm (scale bars 5 µm). (**B**) Quantification of the density of dots per square µm (µm²) and statistical significance between WT and HA-CCDC146 sperm was assessed by Mann–Whitney test (n=58 for WT and 65 for HA_CCDC146), p-value as indicated. p-value of 0.05 or lower was considered statistically significant. Error bars show standard deviation. (**C**) Mouse epididymal spermatozoa observed with expansion microscopy. HA-CCDC146 sperm were immunolabeled with anti-β-tubulin (red) and anti-HA #2 (green) Abs, and DNA was stained with Hoechst (blue). The right image shows the sperm with merged immunostaining. Scale bars 5 µm. (**D**) Mouse epididymal spermatozoa observed with expansion microscopy. The lower images show the staining, HA-CCDC146 (red), tubulin (green), and merge observed in the principal piece of the flagellum. Strong red punctiform signals were observed at the level of axonemal breakages induced by the expansion process. White arrows indicate the zones of the micro breaks. Scale bars 10 µm.

The online version of this article includes the following figure supplement(s) for figure 7:

**Figure supplement 1.** Staining by anti-HA Abs of axonemal breaks induced by expansion.

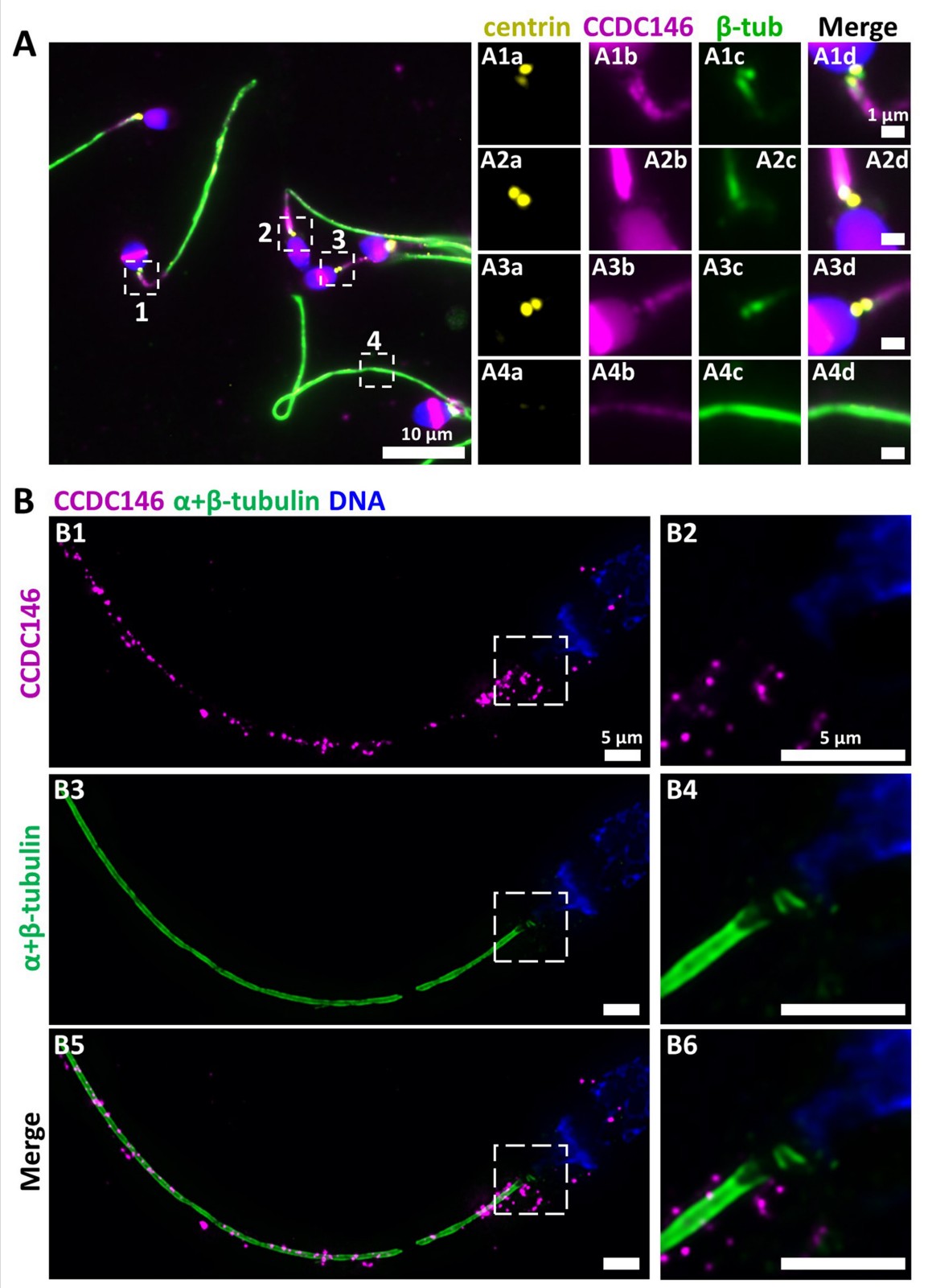

**Figure 8.** CCDC146 localizes to the flagellum but not to the centrioles of ejaculated human spermatozoa. (**A**) Human ejaculated sperm were immunolabeled with Abs recognizing centrin (yellow), CCDC146 (magenta), and β-tubulin (green). DNA was stained with Hoechst (blue). (**A1–A3**) Enlargement of dotted square focused on sperm neck: no colocalization between CCDC146 and centrin. (**A4**) A faint signal for CCDC146 is present along the length of the sperm flagellum. Scale bar of zoomed images: 1 µm. (**B**) Human ejaculated sperm observed by expansion microscopy. Sperm

*Figure 8 continued on next page*

*Figure 8 continued*

were immunolabeled with anti-CCDC146 (magenta) and anti-β-tubulin (green) abs, and DNA was stained with Hoechst (blue). (**B1** and **B5**) show strong staining for CCDC146 in the axoneme, (**B3**) shows the localization of the axoneme through tubulin staining. (**B2, B4,** and **B6**) show enlargements of the dotted squares focused on the sperm neck. CCDC146 did not colocalize with the centrioles at the base of the axoneme. The CCDC146 staining observed probably corresponds to non-specific labeling of mitochondria, as suggested by *Figure 8—figure supplement 1*. Scale bars 5 μm.

The online version of this article includes the following figure supplement(s) for figure 8:

**Figure supplement 1.** Centrioles are identified by anti-POC5 Abs in expanded human ejaculated spermatozoa.

action of sarkosyl on CCDC146 and strengthens the hypothesis that the protein is associated with the doublets of microtubules. To confirm the action of sarkosyl on the accessibility of the antigenic site, murine spermatozoa were labeled with an anti-HA antibody after treatment with sarkosyl or no treatment. CCDC146 labeling in the principal piece was significantly increased in the presence of sarkosyl (*Figure 10C*). The full image panel can be found in *Figure 10—figure supplement 1A*. The presence of the protein in the midpiece was difficult to assess because non-specific staining was observed in this area, the non-specific staining being due to secondary antibodies as shown in *Figure 10—figure supplement 1B*, where they were used alone. Despite all these results, the CCDC146 localization in the lumen of microtubules requires at this stage more proofs.

## KO models show defects in tubulin-made organelles

To better characterize the function of CCDC146 in mouse sperm, we went on to perform a detailed morphological analysis of tubulin-made organelles by IF and scanning microscopy and examined the morphological defects induced by the absence of CCDC146 at the subcellular level by transmission electron microscopy. This work was performed on immature testicular sperm and on seminiferous tubule sections from adult WT and *Ccdc146* KO males. Mouse testicular sperm were used because they still contain centrioles that become disassembled as sperm transit through the epididymis (*Simerly et al., 2016*). We mainly focused our analyses on the centrioles, the manchette, and the axoneme.

The connecting piece between the head and the flagellum, known as the sperm head-tail coupling apparatus (HTCA), is a complex structure containing several substructures including both centrioles, the capitulum and the segmented columns (*Wu et al., 2020*). The distal centriole is embedded in the segmented column and the axoneme emerges from the distal centriole. This structure has a specific shape when observed by scanning electron microscopy (*Figure 11A*). In *Ccdc146* KO sperm, there was a great variability in the morphological defects of the connecting piece, some were almost intact (*Figure 11A2*), whereas others were severely damaged (*Figure 11A3 and A4*). In WT testicular sperm, IF experiments show that the centrioles, identified by anti-tubulin Ab, are very close to each other and adjacent to the nucleus (*Figure 11B1*). For *Ccdc146* KO males, although some sperm cells presented centrioles with almost normal localization (*Figure 11B2*), centriole separation was visible in numerous spermatozoa (*Figure 11B3 and B4*), with the structures located far away from the connecting piece (*Figure 11B5*) or duplicated (*Figure 11B6*). These defects were not observed in WT sperm, demonstrating that these defects were caused by the lack of CCDC146. Interestingly, the overall structure of the HTCA under construction in spermatids, observed by TEM, was conserved in *Ccdc146* KO spermatids, with the presence of both centrioles, containing nine triplets of microtubules, as well as accessory cytoskeletal structures, including the capitulum and the segmented columns (*Figure 12A1* for WT and A2 for *Ccdc146* KO). The adjunct of the proximal centriole was also normal in *Ccdc146* KO spermatids (*Figure 12B*) and no difference were observed with adjunct of WT spermatids (*Figure 12—figure supplement 1*). Remarkably, in a very large proportion of sections, no singlet or doublet of microtubules emerged from the distal KO centriole, suggesting that the process of tubulin polymerization is somehow hampered in these cells (*Figure 12A2*). The absence of microtubules at the end of the distal centriole was confirmed by analysis of serial sections of the sperm centrioles (*Figure 12C*). Moreover, the defects observed in IF experiments, such as duplication, or defective attachment to the nuclear membrane, were frequently confirmed in TEM images (*Figure 12—figure supplement 2*). Such defects were not observed in WT spermatids.

Manchette formation – which occurs from the round spermatid to fully elongated spermatid stages – was next studied by IF using β-tubulin Ab. Simultaneously, we monitored formation of the acrosome using an antibody binding to DPY19L2 (*Figure 13*). Initial acrosome binding at step 3 spermatid was

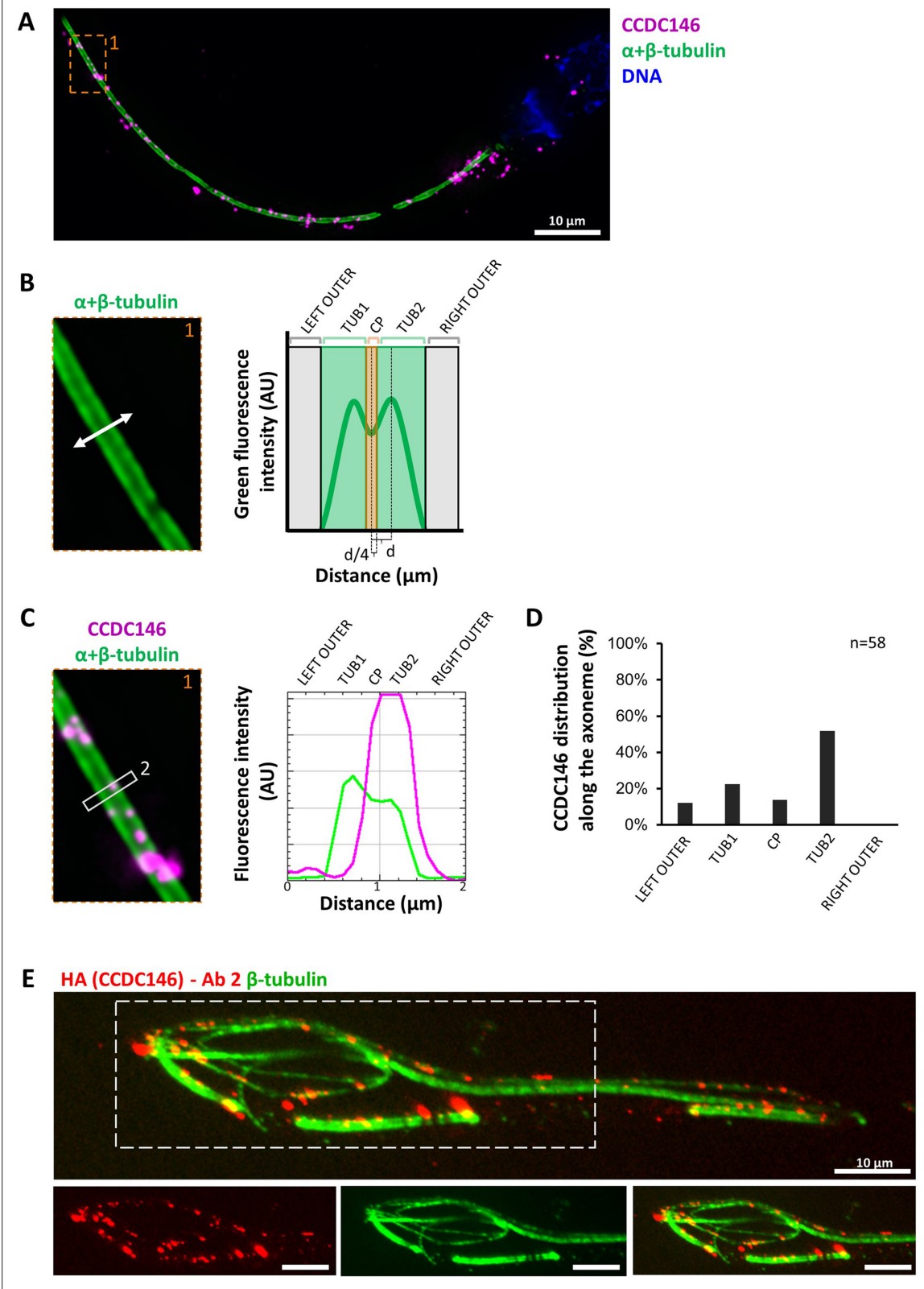

**Figure 9.** CCDC146 localizes to the microtubule doublets of the axoneme in human and mouse. (**A**) Sperm was double-stained with anti-tubulin (green) and anti-CCDC146 (magenta) Abs and observed by expansion microscopy. Scale bar 10 μm. (**B**) Measurement of the green-tubulin signal intensity perpendicularly to the axoneme is quite characteristic with two peaks corresponding to the left and right microtubule doublet. It identifies five axonemal compartments (left outer, left doublet [Tub1], central pair [CP], right doublet [Tub2], and right outer). To determine the CP area, the distance (d) between

*Figure 9 continued on next page*

*Figure 9 continued*

the center of the flagella and the peak of Tub2 was measured, and all fluorescent dots located in between -d/4 to +d/4 from the center were counted as CP dots. (**C**) Example of the measurement of the tubulin (green) and the CCDC146 signal intensities measured at the white rectangle (2). The image corresponds to the orange rectangle in (**A**). In this example, the CCDC146 signal is localized in the right doublet of microtubules (Tub2). (**D**) The position of the CCDC146 signal with respect to the tubulin signal was measured along the entire flagellum. Each CCDC146 signal was assigned to a different compartment of the axoneme, allowing to obtain an histogram showing the distribution of CCDC146 labeling in ejaculated human sperm (n = 2, 38 dots analyzed). (**E**) Flagellum of a mouse epididymal spermatozoa observed with expansion microscopy. HA-CCDC146 sperm were immunolabeled with anti-HA #2 (red) and anti-β-tubulin (green). The upper image shows the sperm with merged immunostaining, and the lower images, the staining (red, green, and merge) observed in the principal piece of the flagellum. Scale bars 10 µm.

not disrupted (not shown), and we observed no differences in covering of the anterior portion of spermatid nucleus at steps 7–8 spermatids (*Figure 13A1/A2*). The development of the manchette in between steps 9–12 spermatids was not hampered, but noticeable defects appeared indicating defective manchette organization– such as a random orientation of the spermatids (*Figure 13B1/B2*) and abnormal acrosome shapes (*Figure 13C1/C2*). Finally, at steps 13–15 spermatids, the manchette was clearly longer and wider than in WT cells (*Figure 13D1/D2*, *Figure 13—figure supplement 1A*), suggesting that the control of the manchette dynamic is defective: in KO spermatids, the reduction and disappearance of the manchette is hampered and contrary to WT continue to expand (*Figure 13—figure supplement 1B*). In the meantime, the acrosome is strongly remodeled, as shown in Figure 16H, and is detached from the nucleus. This latter morphological defect is associated with a loss of the DPY19L2 staining (*Figure 13D2*). In TEM, microtubules of the manchette were clearly visible surrounding the compacting nucleus of elongating spermatids in both WT and KO. The manchette normally anchors on the perinuclear ring, which is itself localized just below the marginal edge of the acrosome and separated by the groove belt (*Figure 14A and B*, blue arrows; *Kierszenbaum et al., 2004*). In KO spermatids, many defects such as marked asymmetry and enlargement were visible (*Figure 14C–F*). The perinuclear ring was no longer localized in the vicinity of the acrosome (*Figure 14C–F*, red arrows) and was often spread into the cytoplasm, providing a large nucleation structure for the manchette (*Figure 14D–F*, black double arrows), which explains its width and irregularity.

Numerous defects were also observed in the axoneme. These axoneme defects were generally similar to the defects observed in other MMAF mouse models, with disorganization of the axoneme structure (*Figure 15A*) accompanied by fragmentation of the dense fibers and their random arrangement in cell masses anchored to the sperm head (*Figure 15B*). The absence of emerging singlet or doublet microtubules at the base of the distal centriole leads to a complete disorganization of the flagellum and the presence of notably dense fiber rings devoid of internal tubulin elements (*Figure 15BC*, red squares and enlargements). Typical cross-sections of the midpiece and principal piece of WT sperm are shown in *Figure 15—figure supplement 1*.

Finally, the shape of the nucleus presented numerous abnormalities during elongation (*Figure 16*). The emergence of head defects was concomitant to the implantation of the manchette (*Figure 16E*, red arrow), with no defects observed on round spermatids at stages III–VIII. Unexpectedly, a distortion was created in the center of the nucleus at the anterior pole, causing the formation of a bilobed compacted nucleus (*Figure 16E and F*, red arrows) and likely responsible of the presence of visible vacuoles inside the nuclei (*Figure 16G and H*, white arrows). Moreover, acrosomes were severely impacted at steps 13–15 spermatid, with strong deformations and detachment (*Figure 16H*, blue arrowheads).

## Discussion

This study allowed us to identify and validate the involvement of a new candidate gene in male infertility, *CCDC146*. This gene was first described in mice by our team earlier this year in the context of a research project on the different types of genetic causes of sperm abnormalities in mouse (*Martinez et al., 2022*). However, no data other than the sperm phenotype (MMAF) was presented in this initial report. In this article, we present for the first time, the evidence of the presence of mutations in humans and a description of the localization of the protein in somatic and germ cells. We also examined the

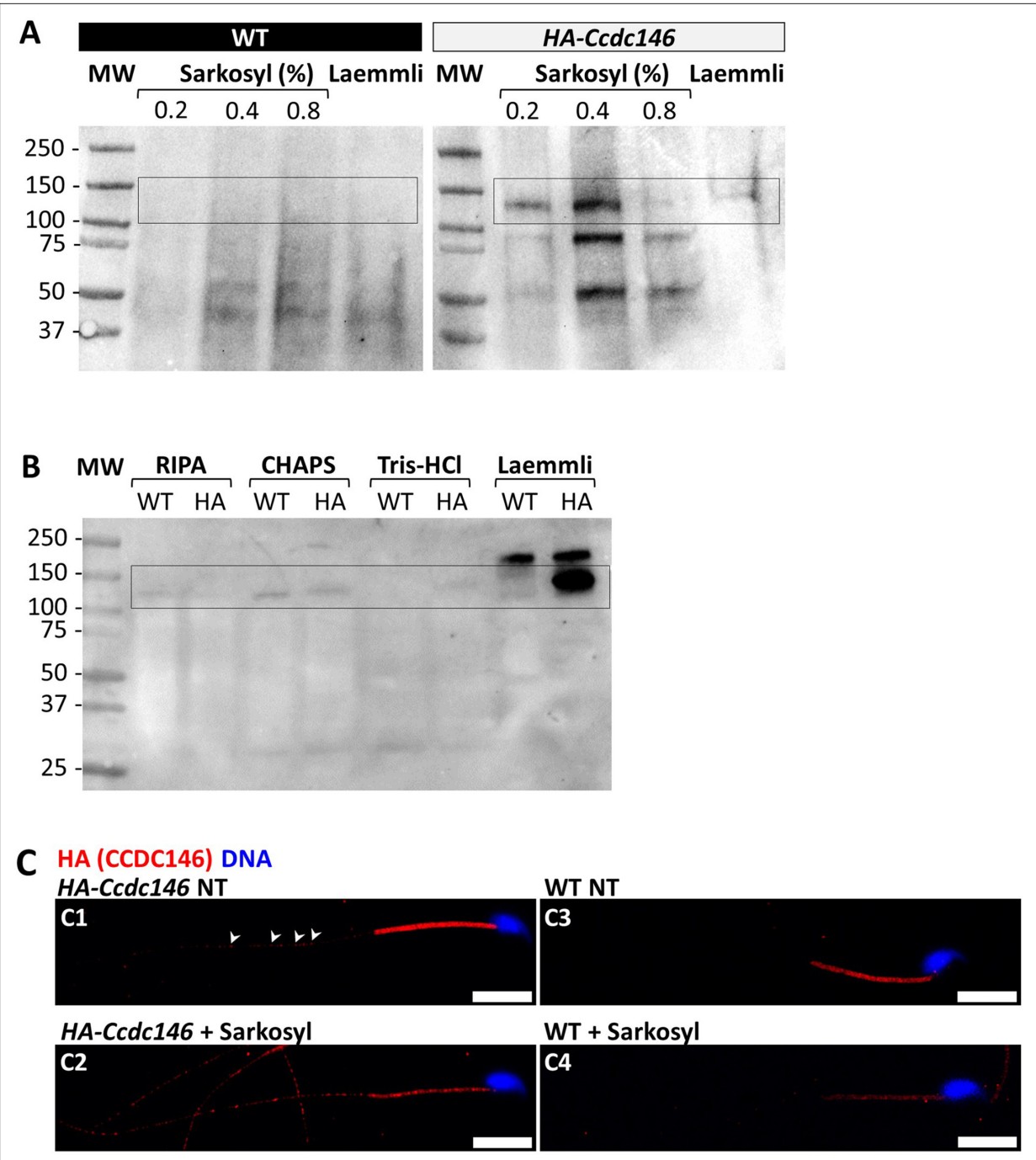

**Figure 10.** CCDC146 is solubilized by sarkosyl and sarkosyl treatment enhances IF signal. (**A**) Western blot of WT and HA-CCDC146 sperm extract solubilized with N-lauroylsarcosine (sarkosyl), an anionic detergent. Sarkosyl was used at increasing concentrations (0.2 and 0.4%). The presence of HA-CCDC146 was detected by an anti-HA Ab. (**B**) Western blot of WT and HA-CCDC146 sperm extracts solubilized with alternative detergents (RIPA, CHAPS, Tris-HCl) and whole sperm extract solubilized in Laemmli. The presence of HA-CCDC146 was revealed by an anti-HA Ab. (**C**) Epididymal HA-CCDC146 sperm (**C1–C2**) and WT sperm (**C3–C4**), treated with sarkosyl (5 min, 0.2% sarkosyl) or not (NT), were immunostained to reveal the HA-tag (red) and counterstained with Hoechst (blue). (**C1**) Without treatment, a faint CCDC146 signal (white arrowheads) is observed along the flagellum from HA-CCDC146 sperm. (**C2**) Treatment with sarkosyl enhanced the HA-CCDC146 signal along the sperm flagellum. (**C3**) The HA signal present the midpiece is likely non-specific since it is present in WT non-treated (NT) sperm. See also *Figure 10—figure supplement 1B*, suggesting that this signal is due to secondary Abs. (**C4**) The HA signal in WT sperm is not enhanced by sarkosyl treatment. Scale bars 10 µm.

The online version of this article includes the following source data and figure supplement(s) for figure 10:

**Source data 1.** Uncropped gel of *Figure 10A*: tagged sperm.

*Figure 10 continued on next page*

*Figure 10 continued*

**Source data 2.** Uncropped gel of *Figure 10A*: WT sperm.

**Source data 3.** Uncropped gel of *Figure 10B*.

**Figure supplement 1.** Sperm sarkosyl treatment corroborates the presence of CCDC146 along the mouse flagellum.

lesion spectrum at the subcellular level, to obtain a detailed view of the molecular pathogenesis associated with defects in this gene.

## Genetic complexity of MMAF

The question of how to interpret and annotate the mutations identified by high-throughput sequencing is an important issue in clinical genetics. The most difficult step is certainly the identification of genes associated with a given disease. Beyond pure genetic data and the pathogenicity of mutations, a fundamental element is the reproduction of the disease when the gene is absent in an animal model. When these two elements are combined – KO photocopying disease and pathogenicity of the identified mutations – the probability of mutation causality in the studied disease is very high. Both mutations are predicted to produce a premature stop codons: p.Arg362Ter and p.Arg704serfsTer7, leading either to the complete absence of the protein in case of non-sense mediated mRNA decay or to the production of a truncated protein missing almost two-third or one-fourth of the protein, respectively. *CCDC146* is very well conserved throughout evolution (*Supplementary file 1*), including the 3′ end of the protein which contains a large coil-coil domain (*Figure 1*). In view of the very high degree of conservation, it is most likely that the 3′ end of the protein, absent in both subjects, is critical for the CCDC146 function and hence that both mutations are deleterious. These results, combined with the fact that the absence of CCDC146 is deleterious in mice and induces an infertility similar to that observed in the human, enable us to associate *CCDC146* with MMAF syndrome in humans.

It is essential to characterize all genes involved in male infertility. Therefore, the characterization of a new gene involved in MMAF syndrome is not just 'one more gene'. First, it further confirms that MMAF syndrome is a heterogeneous recessive genetic disease, currently associated with defects in more than 50 genes. Second, it helps direct the diagnostic strategy to be directed during genome-wide searches. The type of profiling required for MMAF is very different from that performed for instance for cystic fibrosis, where one mutation is responsible for more than 50% of cases. Third, the discovery of all genes involved in MMAF is important in the context of oligogenic heterozygous inheritance of sperm abnormalities in human (*Martinez et al., 2022*). Indeed, a homozygous deleterious mutation is found in only half of all MMAF patients, suggesting that other modes of inheritance must be involved. Only by establishing an extended list of genes linked to MMAF will it be possible to determine whether oligogenic heterozygous inheritance is a relevant cause of this syndrome in humans. Finally, some of the genes involved in MMAF syndrome are also involved in ciliary diseases (*Sironen et al., 2020*). By further exploring MMAF-type infertility, we can hope to enhance our understanding of another underlying pathology. Although the vast majority of patients with PCD are diagnosed before 5 years of age (*Collins et al., 2014*), one study showed that in a cohort of 240 adults with a mean age of 36 y (36 ± 13) presenting with chronic productive cough and recurrent chest infections, PCD was identified for the first time in 10% of patients (*Shoemark et al., 2007*). This result suggests that infertility diagnosis can occur before PCD diagnosis, and consequently that infertility management might improve PCD diagnosis and care.

## CCDC146 is present in flagellum and absent in sperm centriole

Several proteomics studies of the sperm centriole identified CCDC146 as a centrosome-associated protein in bovine sperm (*Firat-Karalar et al., 2014*; *Amargant et al., 2021*), a species where centrioles are present in ejaculated sperm. Using a different methodological approach, based on IF with conventional and expansion microscopy, we observed no CCDC146 signal in centrioles from testicular mouse and ejaculated human sperm. The same result was obtained with two different types of antibodies: with human sperm, we used a commercial anti-CCDC146 Ab, whereas with mouse sperm we used several anti-HA antibodies. The same IF approach allowed us to clearly identify the sperm centrosome using a number of antibodies such as anti-POC5 and anti-β tubulin, ruling out a possible failure of our IF protocol. It is worth noting that centrosomal proteins were isolated from whole flagella

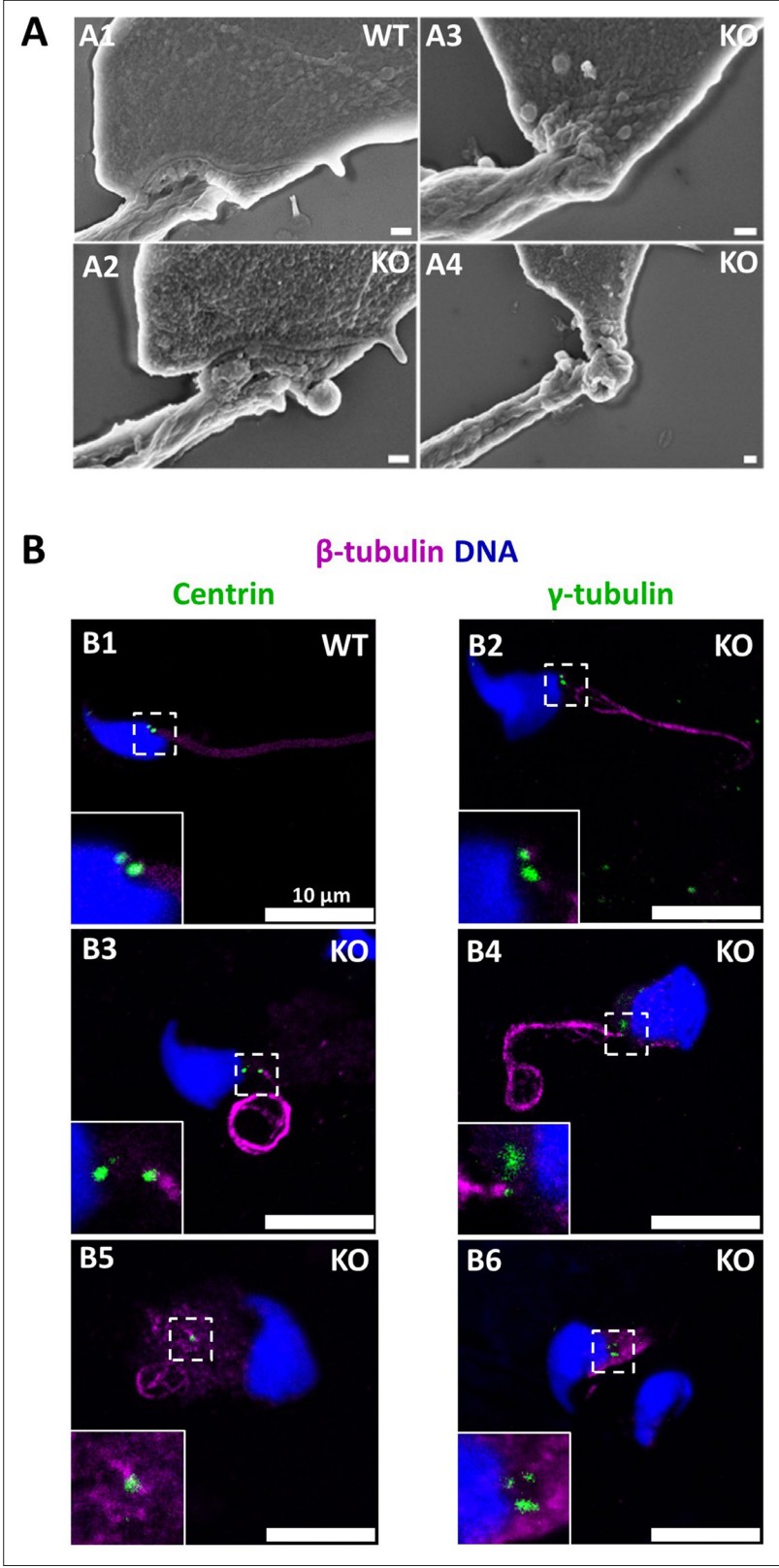

**Figure 11.** The absence of CCDC146 causes defects of the head-tail coupling apparatus in epididymal spermatozoa and duplication and mislocalization of centriole in testicular sperm. (**A**) Scanning electron microscopy of WT and *Ccdc146* KO epididymal spermatozoa shows aberrant head morphologies and irregular head-tail coupling apparatus (HTCA) linking the sperm head with the flagellum. There is a great variability in the

*Figure 11 continued on next page*

*Figure 11 continued*

morphological damage, with sperm presenting almost intact HTCA (**A2**), whereas other were strongly impacted by the absence of the protein (**A3–4**). Scale bars 200 nm. (**B**) Testicular spermatozoa from WT (**B1**) and *Ccdc146* KO (**B2–B6**) mice immunolabeled with anti-β-tubulin (magenta) and anti-centrin (**B1–B3**) or anti-γ-tubulin (**B4–B6**) (green) Abs. Centrioles appeared to be normal (**B2**) in some spermatozoa, separated but partially attached to the head (**B3, B4**), completely detached from the sperm head (**B5**) or duplicated (**B6**). Scale bar 10 μm.

---

using several strategies including sequential use of multiple detergents, and that the centrosomal proteins were purified in the last fractions. These results indicate that CCDC146 is in fact barely soluble in conventional buffers, which could explain why it is co-purified with the centrosomal fraction in proteomics studies.

In epididymal mouse sperm, the biochemical analyses showed the presence of the protein in the flagellum only, and not in the head fractions: the detection of the band appeared very quickly at visualization and became very strong after few minutes, demonstrating that the protein is abundant in the flagella. It is important to note that epididymal sperm do not have centrioles and therefore this signal is not a centriolar signal. We also performed a statistical analysis showing that the immunostaining observed in the principal piece of mouse sperm is very specific (*Figure 7B*). Altogether, these results demonstrate unequivocally the intracellular localization of CCDC146 in the flagellum of mouse sperm. In human, we obtained very similar results, with a flagellar localization. Using expansion microscopy, we show that the protein is associated with the microtubule doublets in both mouse sperm (*Figure 9E*) and human sperm (*Figure 9B–D*).

Moreover, CCDC146 was solubilized with the sarkosyl detergent only, known to destabilize microtubules and generally used to identify microtubule inner proteins (MIP) and the CCDC146 fluorescent signal was enhanced when sperm were treated with this peculiar detergent. Expansion microscopy also revealed a strong signal in mouse sperm at the point of axoneme rupture. Taken together, these elements strongly suggest that CCDC146 is a microtubule-associated protein (MAP) and may be a MIP. However, more experiments need to be performed to validate the inner localization of CCDC146 because there are other proteins, located in the peri-axonemal structure of the axoneme, that are difficult to solubilize as well.

Finally, in human sperm, staining was observed on acrosome and in the midpiece. Staining on acrosome should always be taken with caution in sperm. Indeed, numerous glycosylated proteins are present at the surface of the plasma membrane regarding the outer acrosomal membrane for sperm attachment and are responsible to numerous nonspecific staining. Moreover this acrosomal staining was not observed in mouse sperm, strongly suggesting that it is not specific.

Concerning the staining in the midpiece observed in both conventional and expansion microscopy, it also seems to be nonspecific and associated with secondary Abs. In *Figure 8—figure supplement 1B* showing the localization of POC5, a very similar staining of the midpiece was also observed, although POC5 was never described to be present in the midpiece, therefore questioning the specificity of the signal observed with the anti-CCDC146 antibody in the midpiece. POC5 and CCDC146 staining experiments shared the same secondary Ab and the midpiece signal was likely due to it. For midpiece staining observed in *Figure 10C* and *Figure 10—figure supplement 1A* by IF, it is worth to note that a similar staining was observed when secondary Ab were used alone (see *Figure 10—figure supplement 1B*). The strong fluorescent signal is clearly due to secondary Abs, likely masking the punctuate staining of the axoneme.

## Functional complexity of CCDC146

The results presented in this report show striking differences in localization of CCDC146 between somatic and germ cells. Our results show that the protein displays several subcellular localizations, which vary according to the cell cycle, and the cell nature. In somatic cells, the protein is mostly associated with the centrosome and other microtubular structures – in particular, the mitotic spindle – at the end of microtubules and at the level of the kinetochore and the midbody. These observations also identify CCDC146 as a MAP. In contrast, in spermatozoa, our results indicate that CCDC146 is an axonemal protein that seems to be associated with the microtubule doublets. Despite its broad cellular distribution, the association of CCDC146 with tubulin-dependent structures is remarkable. However, centrosomal and axonemal localizations in somatic and germ cells, respectively, have also

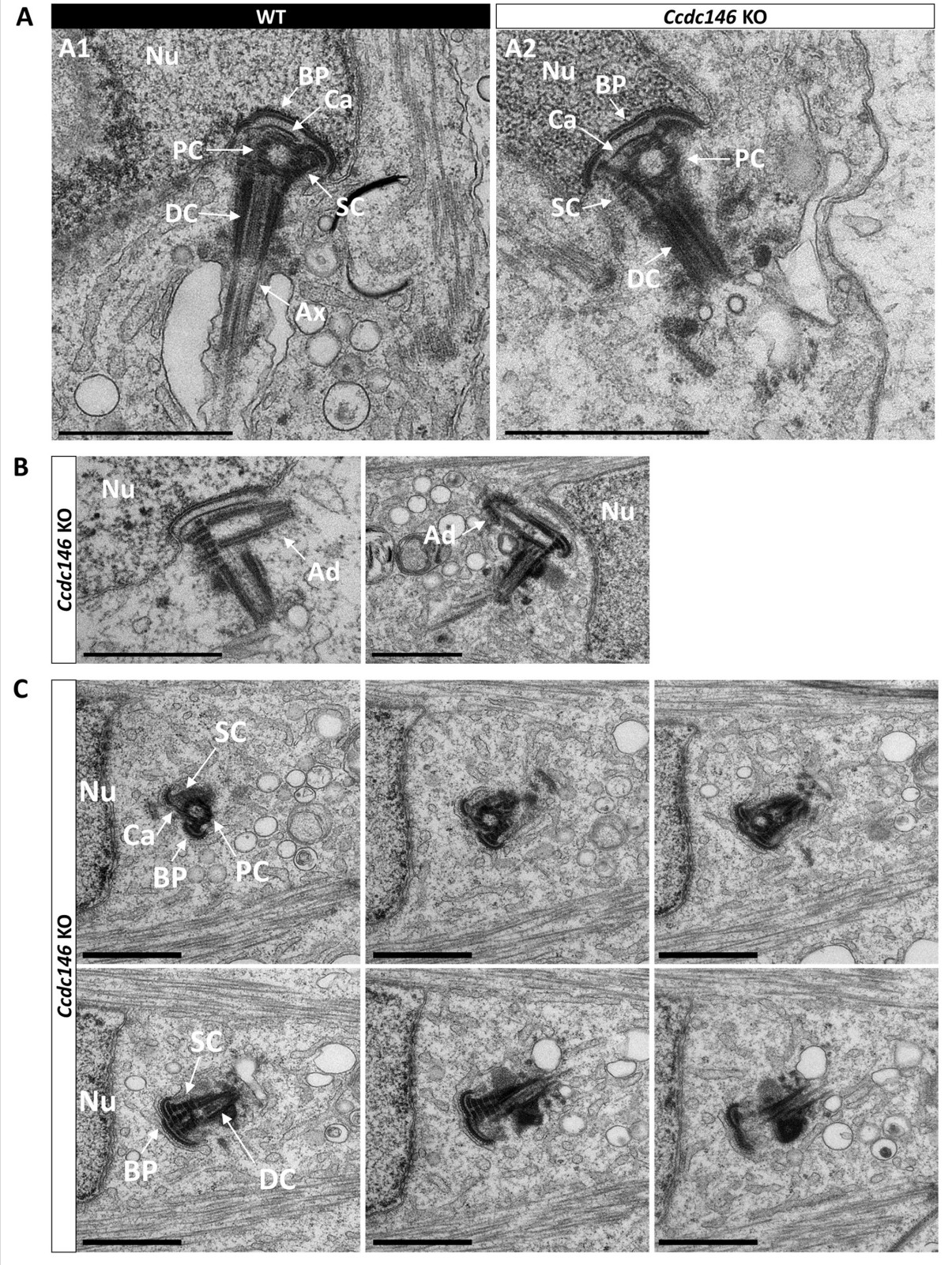

**Figure 12.** Absence of elongation of axonemal microtubules at the base of the distal centriole. (**A**) In WT spermatids (**A1**), the proximal centriole (PC) is linked to the base of the compacting nucleus (Nu) through the basal plate (BP) and the capitulum (Ca), and the distal centriole (DC) is embedded in the segmented column (SC). All these sperm-specific cytoskeletal structures make up the head-tail coupling apparatus (HTCA). At the base of the distal centriole, axonemal microtubules (Ax) grow. In *Ccdc146* KO elongating spermatids (**A2**), the overall structure of the HTCA is conserved, with the

*Figure 12 continued on next page*

Figure 12 continued

presence of the centrioles and the accessory cytoskeletal structures. However, no axonemal microtubules are visible, emerging from the DC. (**B**) The adjunct (Ad) of the proximal centriole is also preserved in *Ccdc146* KO spermatids. (**C**) Serial sections of the HTCA of a *Ccdc146* KO spermatid confirm the absence of axonemal microtubules at the base of the DC during spermatid elongation. Scale bars 1 μm.

The online version of this article includes the following figure supplement(s) for figure 12:

**Figure supplement 1.** Ultrastructure of WT sperm showing the head to tail coupling apparatus (HTCA) in elongating spermatids.

**Figure supplement 2.** Lack of CCDC146 causes centriole duplication and mislocalization in *Ccdc146* KO spermatids.

been reported for CFAP58 (*Sha et al., 2021*; *Li et al., 2020b*), thus the reuse of centrosomal proteins in the sperm flagellar axoneme is not unheard of. In addition, 80% of all proteins identified as centrosomal are found in multiple localizations (https://www.proteinatlas.org/humanproteome/subcellular/centrosome). The ability of a protein to home to several locations depending on its cellular environment has been widely described, in particular for MAP. The different localizations are linked to the presence of distinct binding sites on the protein. For example, MAP6 binds and stabilizes microtubules, through Mc modules, and associates with membranes and neuroreceptors through palmitoylated cysteines. MAP6 can also localize in the microtubule lumen, in its role as MIP, thanks to its Mn modules. Finally, in addition to its associations with subcellular compartments and receptors, the presence of proline-rich domains (PRD) in the MAP6 sequence allows it to bind to SH3-domain-containing proteins, and thus triggering activation of signaling pathways (*Cuveillier et al., 2021*). Another example of a protein with multiple localizations is CFAP21, which is a MIP but also a cytosolic calcium sensor. In the latter capacity, CFAP21 modulates the interaction of STIM1 and ORAI1 upon depletion of calcium stores (*Ma et al., 2019*; *Jardin et al., 2021*). These examples illustrate the complexity of function of some multiple-domain proteins.

The fact that CCDC146 can localize to multiple subcellular compartments suggests that it also contains several domains. Interestingly, a PF05557 motif (Pfam mitotic checkpoint protein, https://www.ebi.ac.uk/interpro/protein/UniProt/E9Q9F7/) has been identified in the mouse CCDC146 sequence between amino acids 130 and 162. Proteins belonging to the 'mitotic spindle checkpoint' monitor correct attachment of the bipolar spindle to the kinetochores. The presence of this motif likely explains the ability of CCDC146 to localize to cell cycle-dependent subcellular compartments containing tubulin. However, the most important structural motifs identified in CCDC146 are the coiled-coil domains. Although coiled-coil domains play a structural role in a variety of protein interactions, their presence in CCDC146 remains mysterious, and how they contribute to its function remains to be elucidated. Nevertheless, this motif is compatible with a MIP function for this protein since several MIP proteins, including CCDC11 (FAP53), CCDC19 (FAP45), and CCDC173 (FAP210), are coiled-coil proteins (*Ma et al., 2019*). It is worth noting that the ortholog of *CCDC146* in *Chlamydomonas*, *MBO2*, codes for a protein required for the beak-like projections of doublets 5 and 6, located inside the lumen of the tubule B (*Segal et al., 1984*), in the proximal part of the *Chlamydomonas* flagellum. Although no beak-like projections are present in the mammalian axoneme, the location inside the tubule seems to be evolutionarily conserved. A more recent study of MBO2 showed that the protein is also present all along the flagellum of *Chlamydomonas* and is tightly associated with microtubule doublets (*Tam and Lefebvre, 2002*). These observations support our results showing association of CCDC146 with this axonemal structure.

The results presented here also show a striking difference in the phenotype induced by the lack of the protein in somatic and male germ cells. This protein is essential for spermatogenesis, and its absence leads to immotile non-functional sperm and to complete infertility in both humans and mice. Conversely, both patients and *Ccdc146* KO mice seem to be healthy and present no other conditions such as PCD. CCDC146, despite its wide expression profile in many tissues, therefore seems to be dispensable except during spermatogenesis. Nevertheless, because this protein is localized on the tips of the spindle and may be involved in mitotic checkpoints, its absence may lead to late proliferative disorders. This hypothesis is supported by the fact that *CCDC146* is reported to be downregulated in thyroid cancer (*Shi et al., 2021*). The link between infertility and risk of cancer was recently underlined, with mutations found in genes like *FANCM* (*Kasak et al., 2018*). Therefore, it would be interesting to monitor aging in *Ccdc146* KO mice and to study their life expectancy and cancer rates compared to WT mice.

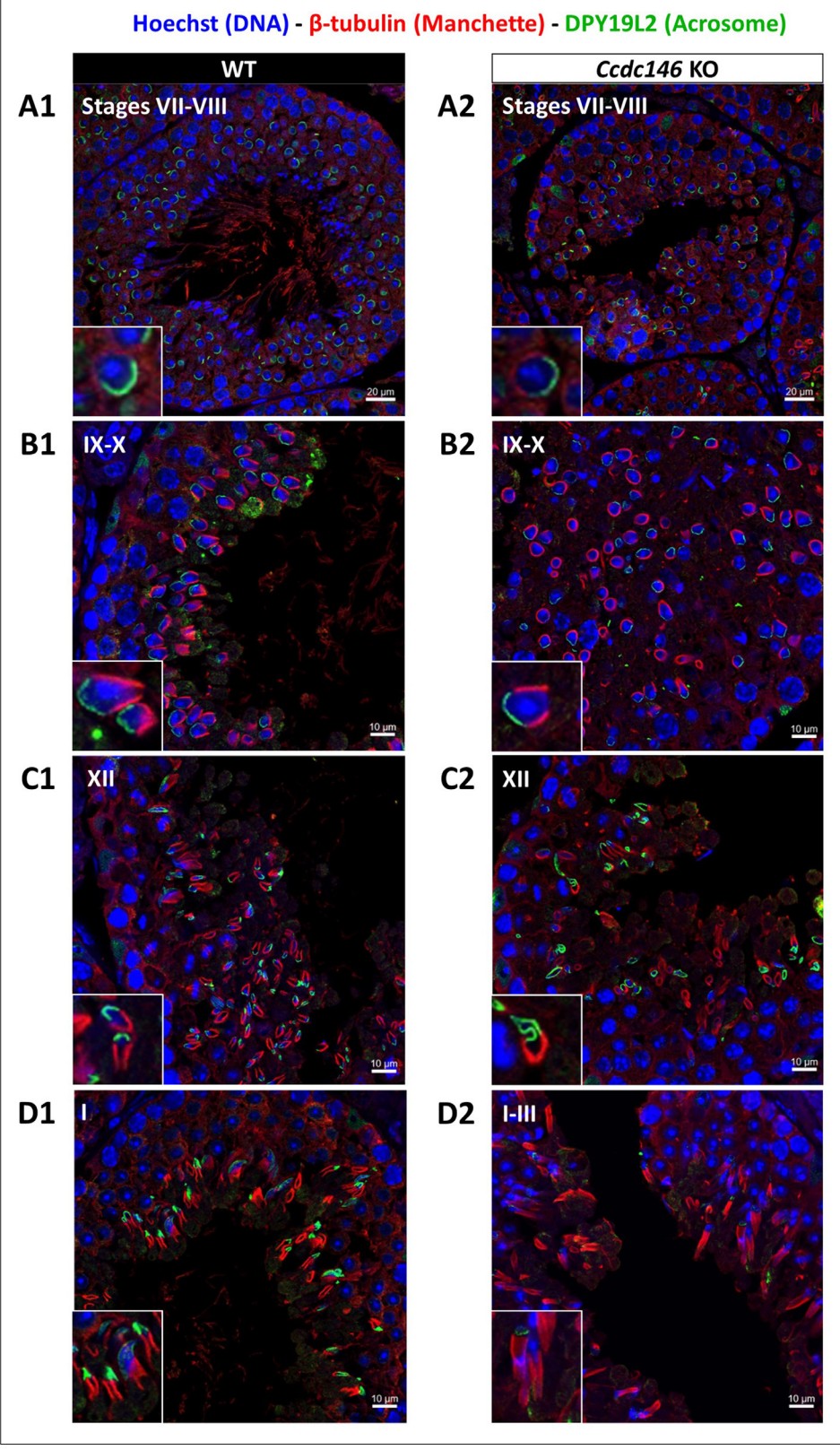

**Figure 13.** Analysis of stages of spermatogenesis by IF reveals acrosome formation and manchette elongation defects in *Ccdc146* KO spermatids. Cross-sections of WT (**A1–D1**) and *Ccdc146* KO (**A2–D2**) testes showing different stages of mouse spermatogenesis (I, VII–XII). Stages were determined by double immunostaining for β-tubulin (red; manchette elongation) and DPY19L2 (green; acrosome localization), and DNA was stained with

*Figure 13 continued on next page*

*Figure 13 continued*

Hoechst (blue). (**A1/A2**) At stage VII–VIII, acrosome spreading on round spermatids appeared similar between WT and KO. However, very few mature spermatozoa lined the lumen in the KO. (**B1/B2**) Cell orientations appeared random from stage IX–X in the KO and the tubules contained more advanced spermatid stages. Random orientations of cells are evidenced by manchette cross-sections in various planes. (**C2**) Abnormal acrosomes of elongating KO spermatids are observed by stage XII. (**D1/D2**) The manchette of elongated spermatids at stage I–III was longer in the KO compared with the WT. Insets in **A1–D1** and **A2–D2** show typical spermatids from stages I–XII from WT and KO males, respectively, showing details of acrosome formation (green) and manchette elongation (red). Scale bars stages VII–VIII 20 μm, scale bars IX–X, XII and I–III 10 μm.

The online version of this article includes the following figure supplement(s) for figure 13:

**Figure supplement 1.** The manchette of elongating spermatids from *Ccdc146* KO male is longer than those from WT males.

## TEM reveals that lack of CCDC146 severely impacts microtubule-based organelles

No morphological defects were observed before the elongating spermatid stage, and in round spermatids, the acrosome started to spread on the nucleus in a normal way (*Figure 13*). Morphological defects appeared clearly from the onset of spermatid elongation. This result indicates that the protein is only necessary for late spermiogenesis, from the phase corresponding to flagellum biogenesis. All the organelles composed mainly of tubulin were strongly affected by the absence of the protein.

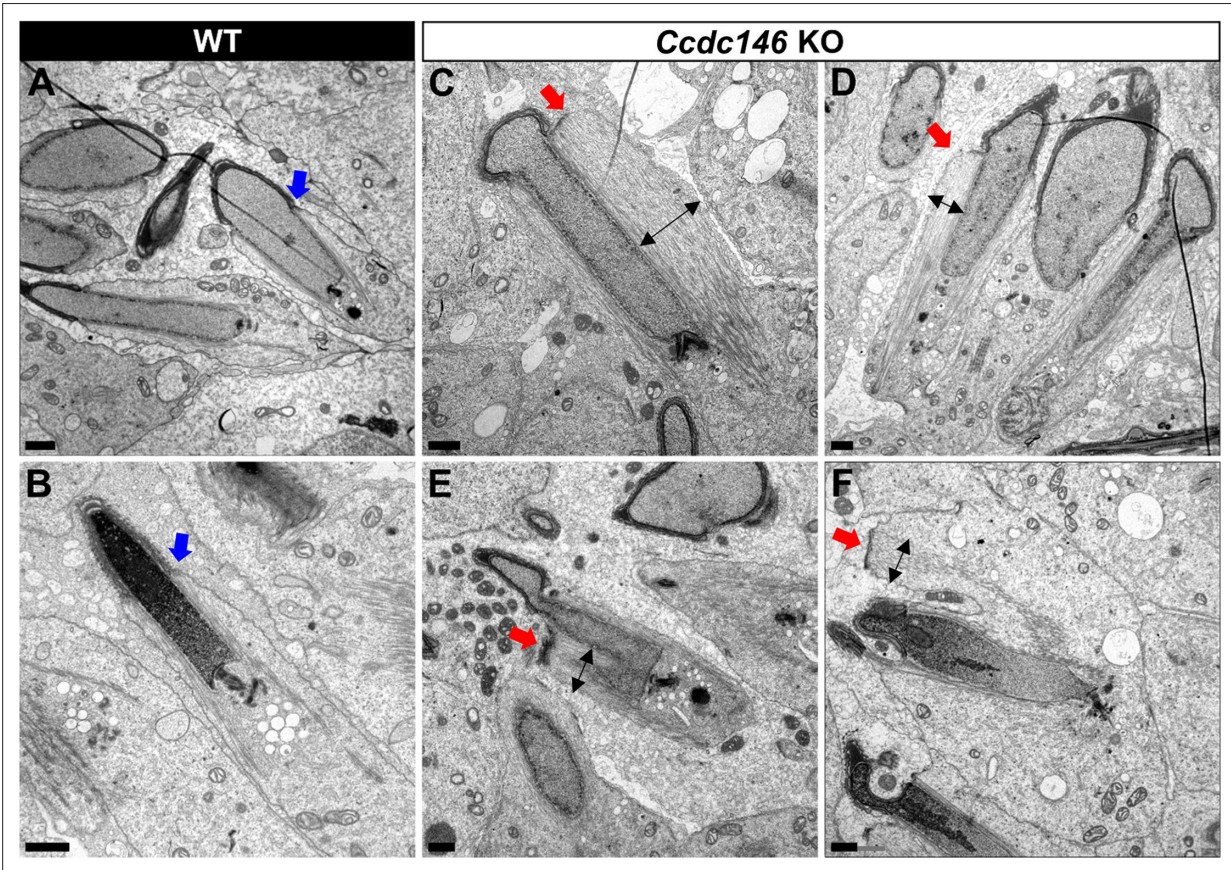

**Figure 14.** TEM of elongating spermatids from *Ccdc146* KO male shows ultrastructural defects of the manchette. (**A, B**) Ultrastructural analysis of the manchette in WT elongating spermatids shows the normal thin perinuclear ring, anchored below the acrosome (blue arrows) and allowing a narrow array of microtubules to anchor. (**C–F**) In elongating spermatids from *Ccdc146* KO animals, the perinuclear ring was abnormally broad, usually located on one side of the spermatid (red arrows), creating an asymmetric and wide bundle of microtubules. The resulting manchette was wider and often longer than in WT animals (black double arrows). (**F**) The tubulin nucleation location was sometimes ectopic in the KO (red arrow) and coincided with irregularly shaped sperm heads. Scale bars 1 μm.

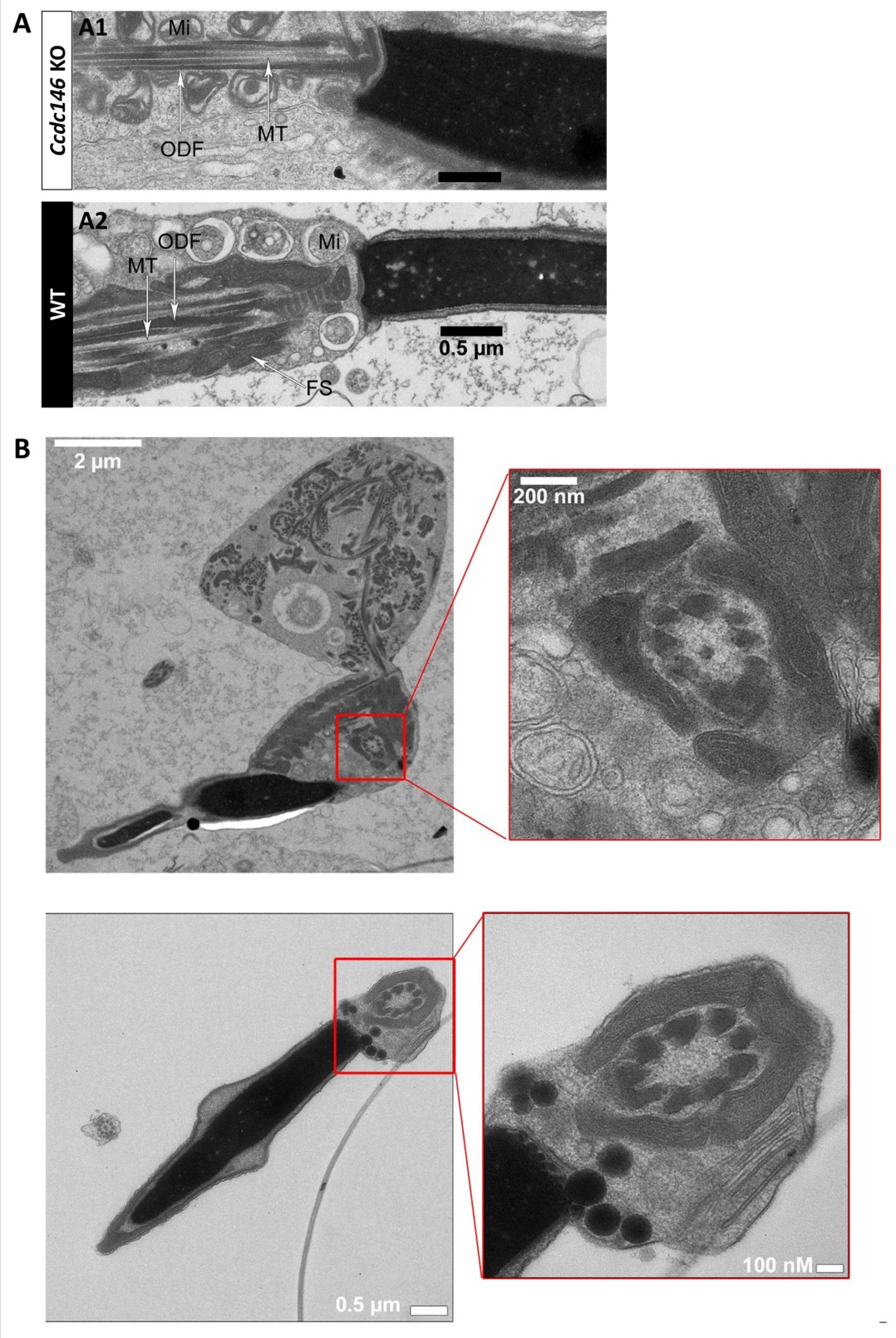

**Figure 15.** The axonemes of *Ccdc146* KO spermatids present multiple defects visible under TEM. (**A1**) A longitudinal section of a WT flagellum shows a typical structure of the principal piece, with outer dense fibers (ODF) at the periphery, microtubules (MT) in the center, and mitochondria (Mi) aligned along the flagellum. (**A2**) A longitudinal section of a *Ccdc146* KO flagellum shows a disorganized midpiece, with altered mitochondria, the presence of an amorphous fibrous sheath (FS) and altered microtubules. Scale bars 0.5 µm (**B**) Longitudinal section of a *Ccdc146* KO sperm showing dispersed

*Figure 15 continued on next page*

*Figure 15 continued*

and non-assembled flagellar material in a cytoplasmic mass. The right-hand image is the enlargement of the red square, showing the presence of an external ring of mitochondria surrounding an ODF ring devoid of microtubular material. (**C**) Longitudinal section of another *Ccdc146* KO sperm showing a similar abnormal midpiece structure. The right-hand image is the enlargement of the red square, showing the presence of an external ring of mitochondria surrounding an ODF ring devoid of microtubular material. Scale bars as indicated.

The online version of this article includes the following figure supplement(s) for figure 15:

**Figure supplement 1.** Cross-sections of the midpiece, principal piece, and tail piece of WT elongating spermatids showing the ultrastructure of the different pieces.

The manchette structure in elongating *Ccdc146* KO spermatids was asymmetric, abnormally broad, and ectopic, leading to the formation of aberrantly shaped sperm heads. So far, manchette defects have been associated with defects in IFT, and intra-manchette transport (IMT) (*Lehti and Sironen, 2016*). CCDC146 was only localized to the sperm axoneme by IF, and no signal was observed in the manchette, suggesting that CCDC146 is probably not involved in the transport machinery. Moreover, our results indicated that the manchette was remarkably long in elongated spermatids. A similar phenotype was observed in Katanin80-deficient animals (*O'Donnell et al., 2012*). Katanin80 is a microtubule-severing enzyme that is important for manchette reduction. Interestingly, the absence of WDR62, a scaffold protein involved in centriole duplication, leads to defective katanin80 expression, and the presence of elongated manchettes in mice (*Ho et al., 2021*). In combination with our results, this detail suggests that the manchette's structure and its reduction are influenced by centrosomal proteins, possibly through katanin80 defects. The precise molecular link between CCDC146 and manchette assembly and reduction remains to be identified.

The HTCA was also aberrant in *Ccdc146* KO spermatids. Centrioles in elongating spermatids were frequently displaced from their implantation fossa at the nuclear envelope. We observed some correctly lodged centrioles in round spermatids; however, we are unable to determine with certainty whether the majority of centrioles failed to correctly attach or whether they detached from the nuclear envelope during spermatid elongation. Defects in cohesion of the HTCA have been associated with the acephalic spermatozoa syndrome and were shown to involve a number of proteins such as SUN5, SPATA6, and ODF1 (*Beurois et al., 2020*; *Zhu et al., 2016*; *Yuan et al., 2015*; *Hoyer-Fender, 2022*). Here, we did not observe any sperm decapitation, suggesting that CCDC146 is involved in a different pathway controlling the HTCA. Moreover, elongating *Ccdc146* KO spermatids displayed supernumerary centrioles. Abnormal centriole numbers have also been reported in the absence of a few other centrosome-associated proteins including DZIP1 (*Lv et al., 2020*) and CCDC42 (*Pasek et al., 2016*), and of microtubule-regulating proteins such as katanin like-2 (*Dunleavy et al., 2017*) and tubulin deglutamylase CCP5 (*Giordano et al., 2019*).

Overall, these results suggest a relationship between the manchette and the centrioles. In the literature, there is limited information about this relationship. Studies of the spermiogenesis defects observed in different models deficient for centrosomal proteins show some common features such as abnormal manchette and duplicated centrosomes. The absence of these proteins does not appear to be directly responsible for these defects, rather it seems to modify the expression of microtubule regulatory proteins such as katanins (*O'Donnell et al., 2012*; *Ho et al., 2021*; *Dunleavy et al., 2017*). Moreover, there is a report showing that both organelles share molecular components (*Tapia Contreras and Hoyer-Fender, 2019*).These modifications could explain the pleiotropic effect of the absence of CCDC146 on microtubule-based organelles.

In conclusion, by characterizing the genetic causes of human infertility, we not only improve the diagnosis and prognosis of these pathologies but also pave the way for the discovery of new players in spermatogenesis. We are constantly adding to the number of proteins present in the flagella of the mammalian spermatozoa that are necessary for its construction and functioning. This study showed that CCDC146, a protein previously described as a centrosomal protein in somatic and germ cells, localizes in spermatozoa's axonemal microtubule doublets. The presence of CCDC146 in somatic cells' centrosomes gives weight to the idea of a centrosome with a dynamic composition, allowing it to fulfill its multitude of functions throughout all the phases of cellular life.

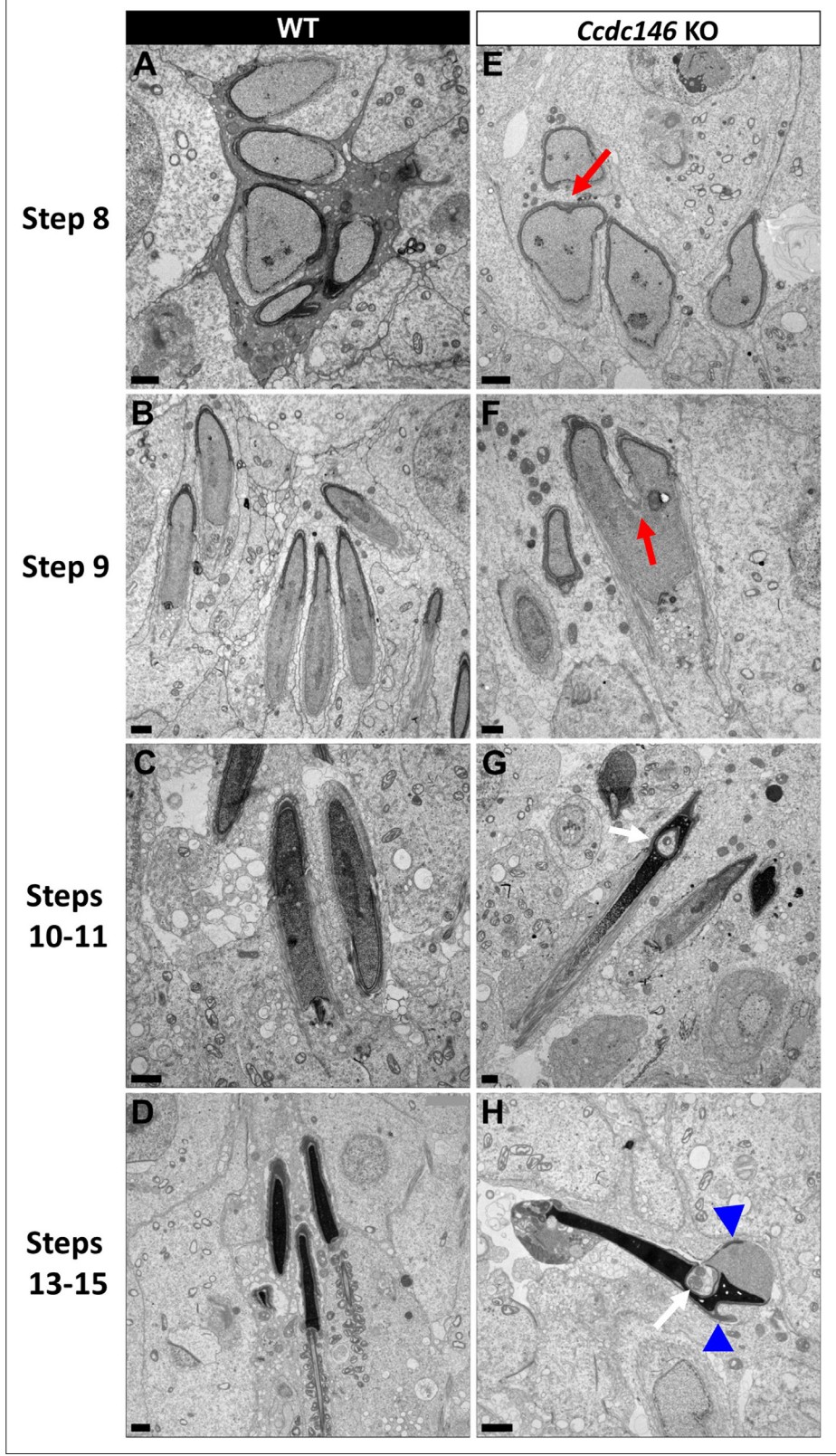

**Figure 16.** Spermatid head shape is aberrant in the absence of CCDC146. Comparative ultrastructural analysis of the spermatid head in WT (**A–D**) and *Ccdc146* KO (**E–H**) testis sections. (**A, E**) Spermatid nuclei at the beginning of elongation. KO spermatid nuclei showing nuclear membrane invaginations and irregular shape that were not present in the WT (red arrow). The acrosome of KO spermatids appeared intact. (**B, F**) Morphology

*Figure 16 continued on next page*

*Figure 16 continued*

of nuclei in elongating spermatids. Whereas nucleus elongation is symmetric in the WT, in the *Ccdc146* KO, more pronounced head invaginations are observed (red arrow). (**C, G**) Elongated spermatids. Although nuclear condensation appeared normal in both WT and KO nuclei, vacuolization is observed in KO nucleus (white arrow). (**D, H**) Elongated KO spermatids showed malformed elongated nuclear shapes with frequent invaginations (white arrow) and absence of flagella (**H**) compared to the WT (**D**). The acrosome of KO spermatids showed numerous defects such as detachment induced by swelling/bubbling of the plasma membrane (blue arrowheads). Scale bars 1 µm.

# Materials and methods

**Key resources table**

| Reagent type (species) or resource | Designation | Source or reference | Identifiers | Additional information |
|---|---|---|---|---|
| Gene (human) | *CCD146* | GenBANK | Gene ID: 57639 | |
| Gene (*Mus musculus*) | *Ccdc146* | GenBANK | Gene ID: 75172 | |
| Cell line (human) | HEK293T | ATCC | | |
| Transfected construct (*M. musculus*) | Myc-DDK-tagged Ccdc146 plasmid DNA | OriGene Technologies, Inc | NM_029195 | Validation of anti-CCDC146-human |
| Biological sample (human) | Human Foreskin Fibroblasts | This paper | | Primary cell line |
| Antibody | Anti-human CCDC146 (rabbit polyclonal) | Atlas Antibodies | HPA020082 | IF: 1/200 |
| Antibody | Anti-centrin- clone 20H5 (mouse monoclonal) | Merck | 04-1624 | IF: 1/200 |
| Antibody | Anti-γ-tubulin (mouse monoclonal) | Santa Cruz Biotechnology | sc-17787 | IF: 1/500 |
| Antibody | Anti-β-tubulin (guinea pig polyclonal) | Geneva Antibody Facility | AA344-GP | IF: 1/500 |
| Antibody | Anti-β-tubulin (mouse monoclonal) | Cell Signaling Technology | 2128 | IF: 1/100 |
| Antibody | Anti-PCM1 (G-6) (mouse monoclonal) | Santa Cruz Biotechnology | sc-398365 | IF: 1/200 |
| Antibody | Anti-α-tubulin (recombinant antibody-isotype mouse monoclonal) | Geneva Antibody Facility | AA-344 | U-ExM: 1/250 IF: 1/500 https://doi.org/10.24450/journals/abrep.2019.e108 |
| Antibody | Anti-β-tubulin (recombinant antibody-isotype mouse monoclonal) | Geneva Antibody Facility | AA-345 | U-ExM: 1/250 IF: 1/500 https://doi.org/10.24450/journals/abrep.2019.e108 |
| Antibody | Anti-POC5 (rabbit polyclonal) | Bethyl | A303-341A | IF: 1/250 U-ExM: 1/200 |
| Antibody | Anti-high affinity (HA) (rat monoclonal) | Roche | 11867423001 | IF: 1/400 U-ExM: 1/400 WB: 1/2500 |
| Antibody | Anti-high affinity (HA) (recombinant antibody isotype rabbit monoclonal) | Cell Signaling Technology | 3724 | U-ExM: 1/100 |

*Continued on next page*

*Continued*

| Reagent type (species) or resource | Designation | Source or reference | Identifiers | Additional information |
|---|---|---|---|---|
| Antibody | Anti-high affinity (HA) (rabbit polyclonal) | Sigma-Aldrich | H6908 | U-ExM: 1/200 |
| Antibody | HA Tag Alexa Fluor 488-conjugated antibody (recombinant Ab, rabbit monoclonal) | Cell Signaling | 28427 | IF:1/400 |
| Antibody | Anti- Dpy19L2 (rabbit polyclonal) | Home made as described in **Pierre et al., 2012** | | IF: 1/100 |
| Sequence-based reagent | gRNA-Ex2 | GenScript | gRNA for exon 2 of Ccdc146 KO | 5'-CCT ACA GTT AAC ATT CGG G-3' |
| Sequence-based reagent | gRNA-EX4 | GenScript | gRNA for exon 4 of Ccdc146 KO | 5'-GGG AGT ACA ATA TTC AGT AC-3' |
| Sequence-based reagent | gRNA-Ex2 for HA-tag | GenScript | gRNA for exon 2 of Ccdc146 tag | 5'-TAC TTT AGA ACT GTG AAA AA-3' |
| Sequence-based reagent | *Ccdc146-F* | Eurogentec | Forward primer for RT-PCR experiment | 5'-TGCTGCATGACGCCGTGATG-3' |
| Sequence-based reagent | *Ccdc146-R* | Eurogentec | Reverse primer for RT-PCR experiment | 5'-GGAGACCTCCGTGGAGAATGCTTC-3' |
| Sequence-based reagent | *Hprt-F* | Eurogentec | Forward primer for RT-PCR experiment | 5'-CCTAATCATTATGCCGAGGATTTGG-3' |
| Sequence-based reagent | *Hprt-R* | Eurogentec | Reverse primer for RT-PCR experiment | 5'-TCCCATCTCCTTCATGACATCTCG-3' |
| Sequence-based reagent | *Actb-F* | Eurogentec | Forward primer for RT-PCR experiment | 5'-CTTCTTTGCAGCTCCTTCGTTGC-3' |
| Sequence-based reagent | *Actb-R* | Eurogentec | Reverse primer for RT-PCR experiment | 5'-AGCCGTTGTCGACGACCAGC-3' |
| Commercial assay or kit | Click-iT Plus TUNEL Assay kit | Invitrogen | C10617 | |

## Human subjects and controls

We analyzed WES data from a cohort of 167 MMAF individuals previously established by our team (*Coutton et al., 2019*). All individuals presented with a typical MMAF phenotype characterized by severe asthenozoospermia (total sperm motility below 10%) with at least three of the following flagellar abnormalities present in >5% of the spermatozoa: short, absent, coiled, bent, or irregular flagella. All individuals had a normal somatic karyotype (46,XY) with normal bilateral testicular size, normal hormone levels and secondary sexual characteristics. Sperm analyses were carried out in the source laboratories during routine biological examination of the individuals according to World Health Organization (WHO) guidelines (*Wang et al., 2014*). Informed and written consents were obtained from all the individuals participating in the study and institutional approval was given by the local medical ethical committee (CHU Grenoble Alpes institutional review board). Samples were stored in the Fertithèque collection declared to the French Ministry of Health (DC-2015-2580) and the French Data Protection Authority (DR-2016-392).

## Sanger sequencing

*CCDC146* single-nucleotide variants identified by exome sequencing were validated by Sanger sequencing as previously described (*Coutton et al., 2019*). PCR primers used for each individual are listed in *Supplementary file 2a*.

## Cell culture

HEK-293T (human embryonic kidney) and human foreskin fibroblast (HFF) cells were grown in D10 medium consisting of DMEM with GlutaMAX (Dulbecco's Modified Eagle's Medium, Sigma-Aldrich)

supplemented with 10% heat-inactivated fetal bovine serum (FBS, Life Technologies) and 10% of penicillin-streptomycin (Sigma-Aldrich) in a 5% $CO_2$ humidified atmosphere at 37°C. HEK-293T cells were divided twice weekly by 1/10 dilution. HFFs cells were divided 1/5 one time a week.

## Generation of *Ccdc146* KO and HA-tagged CCDC146 mice

*Ccdc146* KO mice were generated using the CRISPR/Cas9 technology as previously described *Martinez et al., 2022*. Briefly, to maximize the chances of generating deleterious mutations, two gRNAs located in two distinct coding exons located at the beginning of the targeted gene were used. For each gene, the two gRNAs (5′-CCT ACA GTT AAC ATT CGG G-3′ and 5′-GGG AGT ACA ATA TTC AGT AC-3′), respectively targeting exons 2 and 4, were inserted into two distinct plasmids, each plasmid also contained the Cas9 sequence. The Cas9 gene was controlled by a CMV promoter and the gRNA and its RNA scaffold by a U6 promoter. Full plasmids (pSpCas9 BB-2A-GFP [PX458]) containing the specific sgRNA were ordered from GenScript (https://www.genscript.com/crispr-synthetic-sgrna.html). Both plasmids were co-injected into the zygotes' pronuclei at a concentration of 2.5 ng/mL. Plasmids were directly injected as delivered by the supplier, without in vitro production and purification of Cas9 proteins and sgRNA. PCR primers used for genotyping are listed in *Supplementary file 2b*.

HA-*Ccdc146* knock-in mice were also generated by CRISPR/Cas9. Twenty-seven nucleotides encoding the HA (hemagglutinin) tag (5′-TAC CCA TAC GAT GTT CCA GAT TAC GCT TAG-3′) were inserted immediately after the start codon of *Ccdc146*. One plasmid containing one sgRNA (5′-TAC TTT AGA ACT GTG AAA AA-3′) and Cas9 was injected (5 ng/µL) with a single-stranded DNA (150 nucleotides, 50 ng/µL) as a template for the homology-directed repair (HDR) *Figure 6—figure supplement 1*. PCR primers used for genotyping are listed in *Supplementary file 2c*. Genetically modified *Ccdc146* strains were bred in the Grenoble university animal platform (HTAG) and housed under specific-pathogen-free conditions. Animals were euthanized by cervical dislocation at the indicated ages.

## Phenotypic analysis of *Ccdc146* KO mice

### Fertility test

Three adult males of each genotype were housed individually with two fertile WT B6D2 females for 12 wk. The date of birth and the number of pups were recorded.

### Sperm analysis

Epididymal sperm were obtained by making small incisions in the mouse caudae epididymides placed in 1 mL of warm M2 medium (Sigma-Aldrich), and the sperm were allowed to swim up for 10 min at 37°C. Sperm samples (10 µL) were used for computer-assisted semen analysis (CASA, Hamilton Thorn Research, Beverley, MA) using a 100-µm-deep analysis chamber (Leja Products B.V., Nieuw-Vennep, the Netherlands). A minimum of 100 motile sperm was recorded in each assay. The remaining sperm samples were washed in 1× phosphate-buffered saline (PBS, Life Technologies), 10 µL were spread onto slides pre-coated with 0.1% poly-L-lysine (Epredia), fixed in 70% ethanol (Sigma-Aldrich) for 1 hr at room temperature (RT) and submitted to a Papanicolaou staining (WHO laboratory manual) to assess sperm morphology. Images were obtained using a Zeiss AxioImager M2 fitted with a ×40 objective (color camera AxioCam MRc) and analyzed using ZEN (Carl Zeiss, version 3.4).

## Testis and epididymides histology

Testis and epididymitis samples from 8- to 16-week-old mice were fixed for 24 hr in PBS/4% paraformaldehyde (PFA) (Electron Microscopy Sciences), dehydrated in a graded ethanol series, embedded in paraffin wax, sectioned at 5 µm, and placed onto Superfrost slides (Fisher Scientific). For both, slides were deparaffinated and rehydrated prior to use. Tissue morphology and structure were observed after coloration by Mayer's hematoxylin and eosin phloxine B (WHO protocols) using a Zeiss AxioImager M2 (color camera AxioCam MRc).

## Terminal deoxynucleotidyl transferase dUTP nick-end labeling (TUNEL) assay on testes

Testes samples from three adult individuals for each genotype were analyzed. Apoptotic cells in testis sections were identified using the Click-iT Plus TUNEL Assay kit (Invitrogen) in line with the manufacturer's instructions. DNA strand breaks for the positive control were induced by DNase I treatment. Each slide contained up to eight testis sections. DNA was stained with Hoechst (2 µg/mL). Images were acquired and reconstituted using a Zeiss Axioscan Z1 slide scanner and analyzed using Fiji (*Schindelin et al., 2012*). The total number of seminiferous tubules and the number of tubules containing at least one TUNEL-positive cell were counted in each testis section.

## Conventional immunofluorescence (IF)

### Somatic cells

HEK-293T or HFFs (10,000 cells) were grown on 10 mm coverslips previously coated with poly-D-lysine (0.1 mg/mL, 1 hr, 37°C, Gibco) placed in a well on a 24-well plate. For cell synchronization experiments, cells underwent S-phase blockade with thymidine (5 mM, Sigma-Aldrich) for 17 hr followed by incubation in a control culture medium for 5 hr, then a second blockade at the G2-M transition with nocodazole (200 nM, Sigma-Aldrich) for 12 hr. For *Ccdc146* transfection experiments, $1 \times 10^5$ HEK cells were cultivated on poly-D-lysine (100 µg/mL, Gibco, one night, 4°C) pre-coated 25 mm glass coverslips placed in Nunclon Delta Surface 4-well plates (Thermo Fisher Scientific) for 24 hr at 37°C, 5% $CO_2$ in a humid atmosphere. Cells were transfected with 0.15 µg of Myc-DDK-tagged *Ccdc146* plasmid DNA (NM_029195, OriGene Technologies, Inc) according to the conditions of the jetPRIME kit (Polyplus, Illkirch, France).

Cells were then fixed with cold methanol (Sigma-Aldrich) for 10 min at different times for IF labeling. After washing twice in PBS, non-specific sites were blocked with PBS/5% FBS/5% NGS (normal goat serum, Life Technologies) for 1 hr at RT. After washing three times in PBS/1% FBS, primary antibody was added in PBS/1% FBS and incubated overnight at 4°C. Coverslips were washed three times in PBS/1% FBS before adding secondary antibody in PBS/1% FBS and incubating for 2 hr at RT. After washing three times in PBS, nuclei were stained with 2 µg/mL Hoechst 33342 in PBS (Sigma-Aldrich). Coverslips were once again washed three times in PBS, then carefully placed on Superfrost slides (cells facing the slide) and sealed with nail polish.

### Spermatogenic cells

Seminiferous tubules were isolated from mouse testes (8–16 weeks old). After removing of the tunica albuginea, the testes were incubated at 37°C for 1 hr in 3 mL of a solution containing (mM) NaCl (150), KCl (5), $CaCl_2$ (2), $MgCl_2$ (1), $NaH_2PO_4$ (1), $NaHCO_3$ (12), D-glucose (11), Na-lactate (6), HEPES (10) pH 7.4, and collagenase type IA (1 mg/mL – Sigma-Aldrich). Tubules were rinsed twice in collagenase-free medium and cut into 2 mm sections. Spermatogenic cells were obtained by manual trituration and filtered through a 100 µm filter. The isolated cells were centrifuged (10 min, 500 × *g*), resuspended in 500 µL PBS, and 50 µL was spread onto slides pre-coated with 0.1% poly-L-lysine (Epredia) and allowed to dry. Dried samples were fixed for 5 min in PBS/4% PFA. After washing twice in PBS, slides were placed in PBS/0.1% Triton/5% BSA (Euromedex) for 90 min at RT. Following two washes in PBS, the primary antibody was added in PBS/1% BSA overnight at 4°C. After washing three times in PBS/1% BSA, secondary antibody was added in PBS/1% BSA for 2 hr at RT. After washing three times in PBS, nuclei were stained with 2 µg/mL Hoechst 33342 in PBS (Sigma-Aldrich) and slides were mounted with DAKO mounting media (Agilent).

### Testis sections

After deparaffination and rehydration, testis sections were subjected to heat antigen retrieval for 20 min at 95°C in a citrate-based solution at pH 6.0 (VectorLabs). Tissues were then permeabilized in PBS/0.1% Triton X-100 for 20 min at RT. After washing three times in PBS (5 min each), slides were incubated with blocking solution (PBS/10% BSA) for 30 min at RT. Following three washes in PBS, slides were incubated with primary antibodies in PBS/0.1% Tween/5% BSA overnight at 4°C. Slides were washed three times in PBS before applying secondary antibodies in blocking solution and incubating for 2 hr at RT. After washing in PBS, nuclei were stained with 2 µg/mL Hoechst 33342 in PBS

(Sigma-Aldrich) for 5 min at RT. Slides were washed once again in PBS before mounting with DAKO mounting media (Agilent).

## Spermatozoa

Mouse spermatozoa were recovered from the caudae epididymides. After their incision, sperm were allowed to swim in 1 mL of PBS for 10 min at 37°C. They were washed twice with 1 mL of PBS 1× at 500 × *g* for 5 min, and 10 µL of each sample was smeared onto slides pre-coated with 0.1% poly-L-lysine (Epredia). Sperm were fixed in PBS/4% PFA for 45 s, washed twice in PBS, and permeabilized in PBS/0.1% Triton for 15 min at RT. After incubation in PBS/0.1% Triton/2% NGS for 2 hr at RT, primary antibody was added in PBS/0.1% Triton/2% NGS overnight at 4°C. After washing three times in PBS/0.1% Triton, the secondary antibody was applied in PBS/0.1% Triton/2% NGS for 90 min at RT. Slides were washed three times in PBS before staining nuclei with 2 µg/mL Hoechst 33342 in PBS (Sigma-Aldrich). Slides were washed once in PBS before mounting with DAKO mounting media (Agilent).

For human spermatozoa, the protocol was based on that of *Fishman et al., 2018*. Straws were thawed at RT for 10 min and resuspended in PBS. Sperm were washed twice in 1 mL of PBS (10 min, 400 × *g*), 50 µL was spread onto slides pre-coated with 0.1% poly-L-lysine and left to dry. Dry slides were fixed in 100% ice-cold methanol for 2 min and washed twice in PBS. Cells were permeabilized in PBS/3% Triton X-100 (PBS-Tx) for 1 hr at RT. Slides were then placed in PBS-Tx/1% BSA (PBS-Tx-B) for 30 min at RT. Sperm were then incubated with primary antibodies in PBS-Tx-B overnight at 4°C. After washing three times in PBS-Tx-B for 5 min each, slides were incubated with secondary antibodies for 1 hr at RT in PBS-Tx-B. After washing three times in PBS, nuclei were stained with 2 µg/mL Hoechst 33342 in PBS (Sigma-Aldrich) and slides were mounted with DAKO mounting media (Agilent).

## Sarkosyl protocol for mouse sperm

Mouse sperm from caudae epididymides were collected in 1 mL PBS 1×, washed by centrifugation for 5 min at 500 × *g* and then resuspended in 1 mL PBS 1×. Sperm cells were spread onto slides pre-coated with 0.1% poly-L-lysine (Epredia), treated or not for 5 min with 0.2% sarkosyl (Sigma) in Tris-HCl 1 mM, pH7.5 at RT and then fixed in PBS/4% PFA for 45 s at RT. After washing twice for 5 min with PBS, sperm were permeabilized with PBS/0.1% Triton X-100 (Sigma-Aldrich) for 15 min at RT and unspecific sites were blocked with PBS/0.1% Triton X-100/2% NGS for 30 min at RT. Then, sperm were incubated overnight at 4°C with primary antibodies diluted in PBS/0.1% Triton X-100/2% NGS. Slides were washed three times with PBS/0.1% Triton X-100 before incubating with secondary antibody diluted in PBS/0.1% Triton X-100/2% NGS for 90 min at RT. Finally, sperm were washed three times in PBS/0.1% Triton X-100, adding 2 µg/mL Hoechst 33342 (Sigma-Aldrich) during the last wash to counterstain nuclei. Slides were mounted with DAKO mounting media (Agilent).

## Image acquisition

For all immunofluorescence experiments, images were acquired using ×63 oil objectives on a multimodal confocal Zeiss LSM 710 or Zeiss AxioObserver Z1 equipped with ApoTome and AxioCam MRm or NIKON eclipse A1R/Ti2. Images were processed using Fiji (*Schindelin et al., 2012*) and Zeiss ZEN (Carl Zeiss, version 3.4). Figures for cultured cells were arranged using QuickFigures (*Mazo, 2021*).

## Image analysis

To quantify the number of spots in the flagella the brightfield and fluorescent images were preprocessed with a home-made macro in ImageJ (https://imagej.net/ij/) to improve the contrast and decrease noise. Masks of whole cells, midpieces, and head were then realized using Ilastik (*Berg et al., 2019*) on brightfield and fluorescent images. The masks of midpieces and head were then subtracted from the mask of the whole cell to obtain a mask of the flagella. This mask was used as an ROI to quantify intensity maxima in the flagella of control and CCDC146 expressing cells. To compare the number of spots in images with a random distribution, we quantified the total number of intensity maxima in the fluorescent image and a new image was generated with a random distribution of the same number of spots. Subsequently, the same quantification of intensity maxima in the flagella was

carried out on the random images. A second home-made macro in ImageJ allows to automatize these two steps. Both macros are available on demand.

## Expansion microscopy (U-ExM)

Coverslips used for either sample loading (12 mm) or image acquisition (24 mm) were first washed with absolute ethanol and dried. They were then coated with poly-D-lysine (0.1 mg/mL) for 1 hr at 37°C and washed three times with ddH$_2$O before use. Sperm from cauda epididymides or human sperm from straws were washed twice in PBS. $1 \times 10^6$ sperm cells were spun onto 12 mm coverslips for 3 min at $300 \times g$. Crosslinking was performed in 1 mL of PBS/1.4% formaldehyde/2% acrylamide (Thermo Fisher) for 5 hr at 37°C in a wet incubator. Cells were embedded in a gel by placing a 35 µL drop of a monomer solution consisting of PBS/19% sodium acrylate/10% acrylamide/0.1% N,N'-methylenbisacrylamide/0.2% TEMED/0.2% APS (Thermo Fisher) on parafilm and carefully placing coverslips on the drop, with sperm facing the gelling solution. Gelation proceeded in two steps: 5 min on ice followed by 1 hr at 37°C. Coverslips with attached gels were transferred into a 6-well plate for incubation in 5 mL of denaturation buffer (200 mM SDS, 200 mM NaCl, 50 mM Tris in ddH$_2$O, pH 9) for 20 min at RT until gels detached. Then, gels were transferred to a 1.5 mL microtube filled with fresh denaturation buffer and incubated for 90 min at 95°C. Gels were carefully removed with tweezers and placed in beakers filled with 10 mL ddH$_2$O to cause expansion. The water was exchanged at least twice every 30 min. Finally, gels were incubated in 10 mL of ddH$_2$O overnight at RT. The following day, a 5 mm piece of gel was cut out with a punch. To remove excess water, gels were placed in 10 mL PBS for 15 min, the buffer was changed, and incubation repeated once. Subsequently, gels were transferred to a 24-well plate and incubated with 300 µL of primary antibody diluted in PBS/2% BSA at 37°C for 3 hr with vigorous shaking. After three washes for 10 min in PBS/0.1% Tween 20 (PBS-T) under agitation, gels were incubated with 300 µL of secondary antibody in PBS/2% BSA at 37°C for 3 hr with vigorous shaking. Finally, gels were washed three times in PBS-T for 10 min with agitation. Hoechst 33342 (2 µg/mL) was added during the last wash. The expansion resolution was between 4× and 4.2× depending on sodium acrylate purity. For image acquisition, gels were placed in beakers filled with 10 mL ddH$_2$O. Water was exchanged at least twice every 30 min, and then expanded gels were mounted on 24 mm round poly-D-lysine-coated coverslips, placed in a 36 mm metallic chamber for imaging. Confocal microscopy was performed using either a Zeiss LSM 710 using a ×63 oil objective or widefield was performed using a Leica THUNDER widefield fluorescence microscope, using a ×63 oil objective and small volume computational clearing.

## *Ccdc146* expression

Testis from HA-CCDC146 pups were collected at days 9, 18, 26, and 35 after birth and directly cryopreserved at –80°C before RNA extraction (n = 3 for each day). RNA was extracted as follows. Frozen testes were placed in RLT buffer (QIAGEN)/1% β-mercaptoethanol (Sigma), cut in small pieces and lysis performed for 30 min at RT. After addition of 10 volumes of TRIzol (5 min, RT, Thermo Fisher) and 1 volume of chloroform (2 min, RT, Sigma-Aldrich), the aqueous phase was recovered after centrifugation at $12,000 \times g$, 15 min, 4°C. RNA was precipitated by the addition of one volume of isopropanol (Sigma) and of glycogen (20 mg/mL, Thermo Fisher) as a carrier, tube was placed overnight at –20°C. The day after, after centrifugation (15 min, $12,000 \times g$, 4°C), RNA pellet was washed with ethanol 80%, air-dried and resuspended in 30 µL of ultrapure RNAse-free water (Gibco). RNA concentrations were determined by using the Qubit RNA assay kit (Thermo Fisher). 800 ng of total RNA were used to perform the RT step using the iScript cDNA synthesis kit (Bio-Rad) in a total volume of 20 µL.

**Table 2.** List of primers used for RT-PCR experiments.

| Genes | Forward primer | | Reverse primer | |
| | Sequence | Concentration (nM) | Sequence | Concentration (nM) |
|---|---|---|---|---|
| *Ccdc146* | 5'-TGCTGCATGACGCCGTGATG-3' | 750 | 5'-GGAGACCTCCGTGGAGAATGCTTC-3' | 500 |
| *Hprt* | 5'-CCTAATCATTATGCCGAGGATTTGG-3' | 500 | 5'-TCCCATCTCCTTCATGACATCTCG-3' | 250 |
| *Actb* | 5'-CTTCTTTGCAGCTCCTTCGTTGC-3' | 250 | 5'-AGCCGTTGTCGACGACCAGC-3' | 250 |

Gene expression was assessed by qPCR (1 µL of undiluted cDNA in a final volume of 20 µL with the appropriate amount of primers, see *Table 2*) using the SsoAdvanced Universal SYBR Green Supermix (Bio-Rad). The qPCR program used was 94°C 15 min (94°C 30 s, 58°C 30 s, 72°C 30 s) ×40 followed by a melt curve analysis (58°C 0.05 s, 58–95°C 0.5°C increment 2–5 s/step). Gene expression was calculated using the $2^{-\Delta CT}$ method. Results are expressed relative to *Ccdc146* expression on day 9.

## CCDC146 detection by WB

### Protein extraction from whole sperm head and sperm flagella

Spermatozoa from HA-CCDC146 males (9 weeks old) were isolated from both epididymis in 1 mL PBS and washed twice with PBS by centrifugation at RT (5 min, 500 × *g*). Half of the sperm were incubated with 1× protease inhibitor cocktail (mini cOmplete EDTA-free tablet, Roche Diagnostics), incubated for 15 min on ice and sonicated. Separated flagella were isolated by centrifugation (600 × *g*, 20 min, 4°C) in a Percoll gradient. Percoll concentrations used were 100, 80, 60, 34, 26, and 23%, and sperm flagella were isolated from the 60% fraction and heads form the 100% fraction. Samples were washed with PBS by centrifugation (500 × *g*, 10 min, 4°C). Whole sperm or sperm flagella were incubated in 2× Laemmli buffer (Bio-Rad), heated at 95°C for 10 min and centrifuged (15,000 × *g*, 10 min, 4°C). The supernatants were incubated with 5% β-mercaptoethanol, boiled (95°C, 10 min), cooled down, and placed at –20°C until use.

### Protein solubilization from total sperm

HA-CCDC146 sperm were recovered from caudae epididymides in 1 mL PBS and washed twice in 1 mL PBS by centrifugation 500 × *g*, 5 min. Lysis was then performed for 2 hr at 4°C on wheel in either Chaps buffer (10 mM Chaps/10 mM HEPES/137 mM NaCl/10% glycerol), in RIPA Buffer (Pierce IP Lysis buffer, Thermo Fisher), in Tris 10 mM/HCl 1 M buffer or in Tris 10 mM/HCl 1 M/0.2–0.8% sarkosyl buffer. After centrifugation at 15,000 × *g*, 4°C, 15 min, the supernatants were recovered and 5% of β-mercaptoethanol added. After boiling (95°C, 10 min), the samples were cooled down and placed at –20°C until use.

### Western blot

The different protein lysates were fractionated on 5–12% SDS-PAGE precast gels (Bio-Rad) and transferred onto Trans-Blot Turbo Mini 0.2 µm PVDF membranes using the Trans-Blot Turbo Transfer System (Bio-Rad) and the appropriate program. Membranes were then blocked in PBS/5% milk/0.1% Tween 20 (PBS-T) for 2 hr at RT before incubating with the primary antibody in PBS-T overnight at 4°C with agitation. Membranes were then washed three times for 5 min in PBS-T and incubated with the secondary HRP-antibody in PBS-T for 1 hr at RT. After three washes (PBS-T, 10 min), the membrane was revealed by chemiluminescence using the Clarity ECL substrate (Bio-Rad) and images were acquired on a Chemidoc apparatus (Bio-Rad).

## Transmission electron microscopy

Testis from adult mice were fixed in PBS/4% PFA. They were then decapsulated from the tunica albuginea and the seminiferous tubules were divided into three to four pieces using a razor blade (Gillette Super Sliver). The seminiferous tubules were incubated for 60 min at RT in fixation buffer (100 mM HEPES pH 7.4, 4 mM $CaCl_2$, 2.5% glutaraldehyde, 2% PFA, all from Sigma-Aldrich) and then the buffer was exchanged with fresh fixation buffer and the samples left overnight fixed at 4°C. After washing three times for 10 min in 100 mM HEPES pH 7.4, 4 mM $CaCl_2$, samples were post-fixed in 1% osmium tetroxide (Carl Roth, Karlsruhe, DE) in distilled water for 120 min at 4°C. After three additional 10 min washes in distilled water, the tissue pieces were embedded in 1.5% Difco Agar noble (Becton, Dickinson and Company, Sparks, MD) and dehydrated using increasing concentrations of ethanol. The samples were then embedded in glycidyl ether 100 (formerly Epon 812; Serva, Heidelberg, Germany) using propylene oxide as an intermediate solvent according to the standard procedure. Ultrathin sections (60–80 nm) were cut with a diamond knife (type ultra 35°; Diatome, Biel, CH) on the EM UC6 ultramicrotome (Leica Microsystems, Wetzlar, Germany) and mounted on single-slot pioloform-coated copper grids (Plano, Wetzlar, Germany). Finally, sections were stained with uranyl acetate and lead citrate (*Reynolds, 1963*). The sectioned and contrasted samples were analyzed under a JEM-2100 transmission electron microscope (JEOL, Tokyo, JP) at an acceleration voltage of 80 kV. Images were acquired using a 4080 × 4080

charge-coupled device camera (UltraScan 4000, Gatan, Pleasanton, CA) and Gatan Digital Micrograph software. The brightness and contrast of images were adjusted using the ImageJ program.

## Scanning electron microscopy

The two epididymides of mature males were recovered in 1 mL of 0.1 M sodium cacodylate buffer (pH 7.4, Electron Microscopy Sciences) and the sperm were allowed to swim for 15 min at 37°C. After centrifugation for 10 min, 400 × $g$, RT, the supernatant was discarded and the pellet resuspended in primary fixating buffer (2% glutaraldehyde/0.1 M sodium cacodylate buffer, pH 7.4, Electron Microscopy Sciences) for 30 min at 4°C. After washing three times in 0.1 M sodium cacodylate buffer (400 g, 10 min), the pellet was submitted to post fixation using 1% osmium tetroxide 2% (OsO$_4$, Electron Microscopy Sciences) in 0.1 M sodium cacodylate buffer for 30 min at 4°C. Fixed cells were washed three times in 0.1 M sodium cacodylate buffer (400 × $g$, 5 min), the sample was then placed on a coverslip and treated with Alcian blue 1% (Electron Microscopy Sciences) to improve attachment. The sample was then dehydrated in graded ethanol series: 50, 70, 80, 90, 96, and 100% (10 min, once each). Final dehydration was performed for 10 min in a v/v solution of 100% ethanol/100% hexam-ethyldisilazane (HMDS) followed by 10 min in 100% HMDS. Samples were left to dry overnight before performing metallization. Samples were analyzed using a Zeiss Ultra 55 microscope at the C.M.T.C. – Consortium des Moyens Technologiques Communs (Material characterization platform), Grenoble INP.

## Statistical analysis

Statistical differences were assessed by applying unpaired $t$-tests, Mann–Whitney tests or one-way ANOVA using GraphPad Prism 8 and 9. Histograms show mean ± standard deviation, and p-values were considered significant when inferior to 0.05. The $t$-test was used when the dataset was assumed to have a normal distribution, while the Mann–Whitney test was used when the normality of the distribution was unknown.

## Antibodies used

Primary antibodies

| Target | Host species | Reference | | Dilution |
|---|---|---|---|---|
| CCDC146 | Rabbit | Atlas Antibodies | HPA020082 | IF: 1/200 U-ExM: 1/200 |
| Centrin, 20H5 | Mouse | Merck | 04-1624 | IF: 1/200 |
| γ-Tubulin | Mouse | Santa Cruz Biotechnology | sc-17787 | IF: 1/500 |
| β-Tubulin | Guinea pig | Geneva Antibody Facility | AA344-GP | IF: 1/500 |
| β-Tubulin | Rabbit | Cell Signaling Technology | 2128 | IF: 1/100 |
| PCM1 (G-6) | Mouse | Santa Cruz Biotechnology | sc-398365 | IF: 1/200 |
| α-Tubulin β-tubulin * | Mouse | Geneva Antibody Facility | AA-345 AA-344 | U-ExM: 1/250 IF: 1/500 |
| POC5 | Rabbit | Bethyl | A303-341A | IF: 1/250 U-ExM: 1/200 |
| High affinity (HA) | Rat | Roche | 11867423001 | IF: 1/400 U-ExM: 1/400 WB: 1/2500 |
| High affinity (HA) | Rabbit | Cell Signaling Technology | 3724 | U-ExM: 1/100 |
| High affinity (HA) | Rabbit | Sigma-Aldrich | H6908 | U-ExM: 1/200 |
| HA Tag Alexa Fluor–488-conjugated Antibody | | Cell Signaling | 28427 | IF:1/400 |

*Continued on next page*

*Continued*

| Primary antibodies | | | | |
|---|---|---|---|---|
| Dpy19L2 | Rabbit | † | | IF: 1/100 |

| Target | Fluorophore | Reference | Dilution | Target |
|---|---|---|---|---|
| Goat anti-rabbit | Alexa Fluor 488 | Jackson ImmunoResearch | 111-545-144 | IF: 1/800 U-ExM: 1/250 |
| Goat anti-mouse | DyLight 549 | Jackson ImmunoResearch | 115-505-062 | IF: 1/400 |
| Goat anti-guinea pig | Alexa Fluor 647 | Invitrogen | A-21450 | IF: 1/800 |
| Goat anti-rabbit | Alexa Fluor 568 | Life Technologies | A11036 | U-ExM: 1/250 |
| Goat anti-mouse | Alexa Fluor 488 | Life Technologies | A11029 | U-ExM: 1/250 |
| Goat anti-rat | Alexa Fluor 549 | Jackson ImmunoResearch | 112-505-175 | IF: 1/800 |
| Goat anti-rat | HRP conjugate | Merck | AP136P | 1/10,000 |

*α-tubulin and β-tubulin were used together and noted as α + β-tubulin.
†Dpy19l2 antibodies are polyclonal antibodies produced in rabbit that were raised against RSKLREGSSDRPQSSC and CTGQARRRWSAATMEP peptides corresponding to amino acids 6–21 and 21–36 of the N-terminus of mouse Dpy19l2, as described in *Pierre et al., 2012*.

## Materials availability statement

Biological material created in this article can be obtained by contacting the corresponding authors.

## Acknowledgements

This work was supported by INSERM, CNRS, Université Grenoble Alpes, the French Agence Nationale pour la Recherche (ANR) grants 'MAS-Flagella' (ANR-19-CE17-0014), and 'FLAGELOME' (ANR-19-CE17-0014) to PFR, 'MIP-MAP' (ANR-20-CE13-0005) to CA, the Direction Générale de l'Offre de Soin (DGHOS) for the program PRTS 2014 to PFR, the Fondation Maladies Rares (FMR) grant 'Whole genome sequencing of subjects with Flagellar Growth Defects (FGD)' financed by for the program Séquençage à haut débit 2012 to PFR and the European Research Council (ERC) ACCENT Starting Grant 715289 to PG.

## Additional information

### Funding

| Funder | Grant reference number | Author |
|---|---|---|
| Agence Nationale de la Recherche | ANR-19-CE17-0014 | Pierre F Ray |
| Agence Nationale de la Recherche | ANR-20-CE13-0005 | Christophe Arnoult |
| Direction Générale de l'offre de Soins | PRTS 2014 | Pierre F Ray |
| Fondation Maladies Rares | FGD-2012 | Pierre F Ray |
| European Research Council | 715289 | Paul Guichard |

The funders had no role in study design, data collection and interpretation, or the decision to submit the work for publication.

### Author contributions

Jana Muroňová, Investigation, Methodology, Writing – original draft; Zine Eddine Kherraf, Emeline Lambert, Investigation, Methodology; Elsa Giordani, Simon Eckert, Caroline Cazin, Amir

Amiri-Yekta, Magali Court, Geneviève Chevalier, Guillaume Martinez, Yasmine Neirijnck, Francoise Kühne, Lydia Wehrli, Nikolai Klena, Lisa De Macedo, Investigation; Virginie Hamel, Paul Guichard, Charles Coutton, Supervision, Validation; Jessica Escoffier, Supervision, Investigation; Selima Fourati Ben Mustapha, Mahmoud Kharouf, Raoudha Zouari, Resources; Anne-Pacale Bouin, Software, Validation, Writing – review and editing; Nicolas Thierry-Mieg, Resources, Software, Validation, Writing – review and editing; Serge Nef, Supervision, Writing – review and editing; Stefan Geimer, Supervision, Methodology, Writing – review and editing; Corinne Loeuillet, Supervision, Investigation, Writing – review and editing; Pierre F Ray, Conceptualization, Resources, Supervision, Funding acquisition, Writing – review and editing; Christophe Arnoult, Conceptualization, Formal analysis, Supervision, Funding acquisition, Validation, Writing – original draft, Project administration, Writing – review and editing

### Author ORCIDs
Jana Muroňová ![ORCID] http://orcid.org/0009-0002-3433-4028
Guillaume Martinez ![ORCID] http://orcid.org/0000-0002-7572-9096
Lydia Wehrli ![ORCID] http://orcid.org/0000-0003-1132-9699
Virginie Hamel ![ORCID] http://orcid.org/0000-0001-5092-2343
Jessica Escoffier ![ORCID] http://orcid.org/0000-0001-8166-5845
Paul Guichard ![ORCID] http://orcid.org/0000-0002-0363-1049
Charles Coutton ![ORCID] http://orcid.org/0000-0002-8873-8098
Nicolas Thierry-Mieg ![ORCID] http://orcid.org/0000-0002-7667-2853
Serge Nef ![ORCID] http://orcid.org/0000-0001-5462-0676
Christophe Arnoult ![ORCID] http://orcid.org/0000-0002-3753-5901

### Ethics
Informed and written consents were obtained from all the individuals participating in the study and institutional approval was given by the local medical ethical committee (CHU Grenoble Alpes institutional review board). Samples were stored in the Fertitheque collection declared to the French Ministry of health (DC-2015-2580) and the French Data Protection Authority (DR-2016-392).
Breeding and experimental procedures were carried out in accordance with national and international laws relating to laboratory animal welfare and experimentation (EEC Council Directive 2010/63/EU, September 2010). Experiments were performed under the supervision of C.L. (agreement 38 10 38) in the Plateforme de Haute Technologie Animale (PHTA) animal care facility (agreement C3851610006 delivered by the Direction Departementale de la Protection des Populations) and were approved by the ethics committee of the PHTA and by the French government (APAFIS#7128-2016100609382341. v2).

Reviewer #1 (Public Review): https://doi.org/10.7554/eLife.86845.3.sa1
Reviewer #3 (Public Review): https://doi.org/10.7554/eLife.86845.3.sa2
Author Response https://doi.org/10.7554/eLife.86845.3.sa3

# Additional files

### Supplementary files
• Supplementary file 1. Amino acid conservation of CCDC146 orthologs from man to *X. tropicalis*. The amino acids altered and missing from the presence of the variant p.Arg704serfsTer7 are highlighted in yellow.

• Supplementary file 2. Lists of primers for genotyping. (a) List of primers used for Sanger verification of the identified variants by WES. (b) List of primers used for knock-out mice genotyping. (c) List of primers used for knock-in mice (HA-Tag) genotyping.

• MDAR checklist

### Data availability
All data generated or analysed during this study are included in the manuscript and supporting files.

The following datasets were generated:

| Author(s) | Year | Dataset title | Dataset URL | Database and Identifier |
|-----------|------|---------------|-------------|-------------------------|
| Ray P | 2024 | CCDC146.1 | http://www.ncbi.nlm.nih.gov/clinvar/?term=SCV004232657 | ClinVar, SCV004232657 |
| Ray P | 2024 | CCDC146.2 | http://www.ncbi.nlm.nih.gov/clinvar/?term=SCV004232656 | ClinVar, SCV004232656 |

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
