## [Editor Report · eLife assessment]

This study presents **valuable** information that demonstrates CCDC146 as a novel cause of male infertility that play key role in microtubule-associated structures. The evidence supporting the claims of the authors is **solid** using a combination of human and mouse genetics, biochemical and imaging approaches. This article would be of interest to cell and developmental biologists working on genes involved in spermatogenesis and male infertility.

---

## [Referee Report · Reviewer #1 (Public Review)]

Here, Muronova et al., demonstrate the physiological importance of a centriole and microtubule-associated protein, CCDC146, in sperm flagellar formation and male reproduction. This study identifies novel causal variants to cause male infertility and resolves the pathogenicity by the mutation with characterizing mouse models. Furthermore, the authors' claims are well supported by the biochemical and imaging approaches used in this study.

---

## [Referee Report · Reviewer #3 (Public Review)]

Male infertility is an important health problem. Among pathologies with multiple morphological abnormalities of the flagellum (MMAF), only 50% of the patients have no identified genetic causes. It is thus primordial to find novel genes that cause the MMAF syndrome. In the current work, the authors follow up the identification of two patients with MMAF carrying a mutation in the CCDC146 gene. To understand how mutations in CCDC146 lead to male infertility, the authors generated two mouse models: a CCDC146-knockout mouse, and a knockin mouse in which the CCDC146 locus is tagged with an HA tag. Male CCDC146-knockout mice are infertile, which proves the causative role of this gene in the observed MMAF cases. Strikingly, animals develop no other obvious pathologies, thus underpinning the specific role of CCDC146 in male fertility. The authors have carefully characterised the subcellular roles of CCDC146 by using a combination of expansion and electron microscopy. They demonstrate that all microtubule-based organelles, such as the sperm manchette, the centrioles, as well as the sperm axonemes are defective when CCDC146 is absent. Their data show that CCDC146 is a microtubule-associated protein, and indicate, but do not prove beyond any doubt, that it could be a microtubule-inner protein (MIP).

This is a solid work that defines CCDC146 as a novel cause of male infertility. The authors have performed comprehensive phenotypic analysis to define the defects in CCDC146 knockout mice. The manuscript text is well written and easy to follow also for non-specialists. The introduction and discussion chapters contain important background information that allow to put the current work into the greater context of fertility research. Overall, this manuscript provides convincing evidence for CCDC146 being essential for male fertility and illustrates this with a large panel of phenotypic observations. Together, the work provides important first insights into the role of a so-far unexplored proteins, CCDC146, in spermatogenesis, thereby broadening the spectrum of genes involved in male infertility.

---

## [Author Response]

The following is the authors’ response to the original reviews.

First of all, we'd like to thank the three reviewers for their meticulous work that enable us to present now an improved manuscript and substantial changes were made to the article following reviewers' and editors' recommendations. We read all their comments and suggestions very carefully. Apart from a few misunderstandings, all comments were very pertinent. We responded positively to almost all the comments and suggestions, and as a result, we have made extensive changes to the document and the figures. This manuscript now contains 16 principal figures and 15 figure supplements.

The number of principal figures is now 16 (1 new figure), and additional panels have been added to certain figures. On the other hand, we have added 7 additional figures (supplement figures) to answer the reviewers' questions and/or comments.

Main figures

▪ Figures 1, 4, 5, 10, 11, 12, 13, 14: unchanged ▪ Figure 7 and 8 were switched.

▪ Figure 2: we added panel F in response to reviewer 3's and request for sperm defect statistics

▪ Figure 3: the contrast in panel B has been taken over to homogenize colors

▪ Figure 6: This figure was recomposed. The WB on testicular extract was suppressed and we present a new WB allowing to compare the presence of CCDC146 in the flagella fraction. Using an anti-HA Ab, we demonstrate that the protein is localized in the flagella in epididymal sperm. Request of the 3 reviewers.

▪ Figure 7 (old 8): to avoid the issue of the non-specificity of secondary antibodies, we performed a new set of IF experiments using an HA Tag Alexa Fluor 488-conjugated Antibody (anti-HA-AF488-C Ab) on WT and HA-CCDC146 sperm. These results are now presented in figure 7 panel A (new). The specificity of the signal obtained with the anti-HA-AF488-C Ab on mouse spermatozoa was evaluated by performing a statistical study of the density of dots in the principal piece of the flagellum from HA-CCDC146 and WT sperm. These results are now presented in figure 7 panel B (new). This study was carried out by analyzing 58 WT spermatozoa and 65 CCDC146 spermatozoa coming from 3 WT and 3 KI males. We found a highly significant difference, with a p-value <0.0001, showing that the signal obtained on spermatozoa expressing the tagged protein is highly specific. We have added a paragraph in the MM section to describe the process of image analysis. We finally present new images obtained by ExM showing no staining in the midpiece (figure 7C new). Altogether, these results demonstrate unequivocally the presence of the protein in the flagellum. Moreover, the WB was removed and is now presented in figure 6 (improved as requested).

▪ Figure 8. Was old figure 7

▪ Figure 9: figure 9 was recomposed and improved for increased clarity as suggested by reviewer 2 and 3.

▪ Figure 16 was before appendix 11

Figure supplements and supplementary files

▪ Figure 1-Figure supplement 1 New. Sperm parameters of the 2 patients. requested by editor (remark #1) by the reviewer 1 (Note #3)

▪ Figure 2-Figure supplement 1 new. Sperm parameters of the line 2 (KO animals) requested by the reviewer 1 (Note #5)

▪ Figure 4-Figure supplement 1 New. Experiment to evaluate the specificity of the human CCDC146 antibody. Minimal revision request and reviewer 1 note #8

▪ Figure 6-Figure supplement 1 New. Figure recomposed; Asked by reviewer 2 note #4 and reviewer 3

▪ Figure 8-Figure supplement 1 New. We now provide new images to show the non-specific staining of the midpiece of human sperm by secondary Abs in ExM experiments; Asked by reviewer 2

▪ Figure 10-Figure supplement 1 New. We added new images to show the non-specific staining of the midpiece of mouse sperm by secondary Abs in IF (panel B). Rewiever 1 note #9 and reviewer 2 note #5

▪ Figure 12-Figure supplement 1 New. Control requested by reviewer 3 Note #23

▪ Figure 13-Figure supplement 1 New. We provide a graph and a statistical analysis demonstrating the increase of the length of the manchette in the Ccdc146 KO. Requested by editor and reviewer 3 Note 24

▪ Figure 15-Figure supplement 1 New. Control requested by reviewer 2. Minor comments

▪ Figure supplementary 1 New. Answer to question requested by reviewer 2 note #1

All the reviewers' and editors’ comments have been answered (see our point to point response) and we resubmit what we believe to be a significantly improved manuscript. We strongly hope that we meet all your expectations and that our manuscript will be suitable for publication in "eLife". We look forward to your feedback,

Point by point answerPlease note that there has been active discussion of the manuscript and the summarize points below is the minimal revision request that the reviewers think the authors should address even under this new review model system. It was the reviewers' consensus that the manuscript is prepared with a lot of oversights - please see all the minor points to improve your manuscript.

All minimal revision requests have been addressed

Minimal revision request1. Clinical report/evaluation of the two patients should be given as it was not described even in their previous study as well as full description of CCDC146.

We provide now a new Figure 1-figure supplement 1 describing the patients sperm parameters

1. Antibody specificity should be provided, especially given two of the reviewers were not convinced that the mid piece signal is non-specific as the authors claim. As both KO and KI model in their hands, this should be straightforward.

To validate the specificity of the Antibody, we transfected HEK cells with a human DDK-tagged CCDC146 plasmid and performed a double immunostaining with a DDK antibody and the CCDC146 antibody. We show that both staining are superimposable, strongly suggesting that the CCDC146 Ab specifically target CCDC146. This experiment is now presented in Figure 4-Figure supplement 1.Next, to avoid the issue of the non-specificity of secondary antibodies, we performed a new set of IF experiments using an HA Tag Alexa Fluor 488-conjugated Antibody (anti-HA-AF488-C Ab) on WT and HA-CCDC146 sperm. These results are now presented in figure 7 panel A (new). The specificity of the signal obtained with the anti-HA-AF488-C Ab on mouse spermatozoa was evaluated by performing a statistical study of the density of dots in the principal piece of the flagellum from HA-CCDC146 and WT sperm. These results are now presented in figure 7 panel B (new). This study was carried out by analyzing 58 WT spermatozoa and 65 CCDC146 spermatozoa coming from 3 WT and 3 KI males. We found a highly significant difference, with a p-value <0.0001, showing that the signal obtained on spermatozoa expressing the tagged protein is highly specific. We have added a paragraph in the MM section to describe the process of image analysis. We finally present new images obtained by ExM showing no staining in the midpiece (figure 7C new). Altogether, these results demonstrate unequivocally the presence of the protein in the flagellum.

1. The authors should improve statistical analysis to support their experimental results for the reader can make fair assessment. Combined with clear demonstration of ab specificity, this lack of statistical analysis with very few sample number is a major driver of dampening enthusiasm towards the current study.

Several statistical analyses were carried out and are now included:

1. distribution of the HA signal in mouse sperm cells (see point 2 Figure 7 panel B)

2. quantification and statistical analyses of the defect observed in Ccdc146 KO sperm (figure 2 panelE)

3. Quantification and statistical analyses of the length of the manchette in spermatids 13-15 steps (Figure 13-Figure supplement 1 new)

1. The authors need to clarify (peri-centriolar vs. centriole)

In figure 4A, we have clearly shown that the protein colocalizes with centrin, a centriolar core protein in somatic cells. This colocalization strongly suggests that CCDC146 is therefore a centriolar protein, and this is now clearly indicated lines 211-212. However, its localization is not restricted to the centrioles and a clear staining was also observed in the pericentriolar material (PCM). The presence of a protein in PCM and centriole was already described, and the best example is maybe gamma-tubulin (PMID: 8749391).

or tone down (CCDC146 to be a MIP) of their claim/description.

Concerning its localization in sperm, we agree with the reviewer that our demonstration that CCDC146 is MIP would deserve more results. Because of that, we have toned down the MIP hypothesis throughout the manuscript. See lines 491495

Testis-specific expression of CCDC146 as it is not consistent with their data.

We have also modified our claim concerning the testis-expression of CCDC146. Line 176

**Reviewer #1 (Recommendations For The Authors):**
Major comments1. As described in general comments, this study limits how the CCDC146 deficiency impairs abnormal centriole and manchette formation. The authors should explain their relationship in developing germ cells.

In fact, there are limited information about the relationship between the manchette and the centriole. However, few articles have highlighted that both organelles share molecular components. For instance, WDR62 is required for centriole duplication in spermatogenesis and manchette removal in spermiogenesis (Commun Biol. 2021; 4: 645. doi: 10.1038/s42003-021-02171-5). Another study demonstrates that CCDC42 localizes to the manchette, the connecting piece and the tail (Front. Cell Dev. Biol. 2019 https://doi.org/10.3389/fcell.2019.00151). These articles underline that centrosomal proteins are involved in manchette formation and removal during spermiogenesis and support our results showing the impact of CCDC146 lack on centriole and manchette biogenesis. This information is now discussed. See lines 596-603

1. The authors generated knock-in mouse model. If then, are the transgene can rescue the MMAF phenotype in CCDC146-null mice? This reviewer strongly suggest to test this part to clearly support the pathogenicity by CCDC146.

We indeed wrote that we created a “transgenic mice”, which was misleading. We actually created a CCDC16 knock-in expressing a tagged-protein. The strain was actually made by CRISPR-Cas9 and a sequence coding for the HA-tag was inserted just before the first amino acid in exon 2, leading to the translation of an endogenous HA-tagged CCDC146 protein. We have removed the word transgenic from the text and made changes accordingly (see lines 250-253). We can therefore not use this strain to rescue the MMAF phenotype as suggested by the reviewer.

1. Although the authors cite the previous study (Coutton et al., 2019), the study does not describe any information for CCDC146 and clinical information for the patients. The authors must show the results for clinical analysis to clarify the attended patients are MMAF patients without other phenotypic defects.

We have now inserted a table, indicating all sperm parameters for the patients harboring a mutation in the CCDC146 gene (Figure 1-Figure supplement 1) and is now indicated lines 159-160

1. The authors describe CCDC146 expression is dominant in testes, However, the level in testis is only moderate in human (Supp Figure 1). Thus, this description is not suitable.

In Figure 1-figure supplement 2 (old FigS1), the median of expression in testis is around 12 in human, a value considered as high expression by the analysis software from Genevestigator. However, for mouse, it is true that the level of expression is medium. We assumed that reviewer’s comment concerned testis expression in mouse. To take into account this remark, we changed the text accordingly. See line 176.

1. Although the authors mentioned that two mice lines are generated, only one line information is provided. Authors must include information for another line and provide basic characterization results to support the shared phenotype within the lines.

We now provide a revised Figure 2-figure supplement 1CD, presenting the second line and the corresponding text in the main text is found lines 178-183.

1. In somatic cells, the CCDC146 localizes at both peri-centriole and microtubule but its intracellular localization in sperm is distinguished. The authors should explain this discrepancy.

The multi-localization of a centriolar protein is already discussed in detail in discussion lines 520-526. We have written:

“Despite its broad cellular distribution, the association of CCDC146 with tubulin-dependent structures is remarkable. However, centrosomal and axonemal localizations in somatic and germ cells, respectively, have also been reported for CFAP58 [37, 55], thus the re-use of centrosomal proteins in the sperm flagellar axoneme is not unheard of. In addition, 80% of all proteins identified as centrosomal are found in multiple localizations (https://www.proteinatlas.org/humanproteome/subcellular/centrosome). The ability of a protein to home to several locations depending on its cellular environment has been widely described, in particular for MAP. The different localizations are linked to the presence of distinct binding sites on the protein…. “

1. Authors mention CCDC146 is a centriolar protein in the title and results subtitle. However, the description in results part depicts CCDC146 is a peri-centriolar protein, which makes confusion. Do the authors claim CCDC146 is centrosomal protein?

In figure 4A, we have clearly shown that the protein colocalizes with centrin, a centriolar core protein. This colocalization strongly suggests that CCDC146 is therefore a centriolar protein in somatic cells, and is now clearly indicated lines 211-212. However, its localization is not restricted to the centrioles and a clear staining was also observed in the pericentriolar material (PCM). The presence of a protein in PCM and centriole was already described and the best example is maybe gamma-tubulin (PMID:8749391).

1. Verification of the antibody against CCDC146 must be performed and shown to support the observed signal are correct. 2nd antibody only signal is not proper negative control.

It is a very important remark. The commercial antibody raised against human CCDC146 was validated in HEK293-cells expressing a DDK-tagged CCDC146 protein. Cells were co-marked with anti-DDK and anti-CCDC146 antibodies. We have a perfect colocalization of the staining. This experiment is now presented in Figure 4-figure supplement 1 and presented in the text (lines 206-208).

1. In human sperm, conventional immunostaining reveals CCDC146 is detected from acrosome head and midpiece. However, in ExM, the signal at acrosome is not detected. How is this discrepancy explained? The major concern for the ExM could be physical (dimension) and biochemical (properties) distortion of the sample. Without clear positive and negative control, current conclusion is not clearly understood. Furthermore, it is unclear why the authors conclude the midpiece signal is non-specific. The authors must provide experimental evidence.

Staining on acrosome should always be taken with caution in sperm. Indeed, numerous glycosylated proteins are present at the surface of the plasma membrane regarding the outer acrosomal membrane for sperm attachment and are responsible for numerous nonspecific staining. Moreover, this acrosomal staining was not observed in mouse sperm, strongly suggesting that it is not specific.

Concerning the staining in the midpiece observed in both conventional and Expansion microscopy, it also seems to be nonspecific and associated with secondary Abs.

For IF, we now provide new images showing clearly the nonspecific staining of the midpiece when secondary Ab were used alone (see Figure 10-figure supplement 1B).

For ExM, we provide new images in Figure 8-figure supplement 1B (POC5 staining) showing a staining of the midpiece (likely mitochondria), although POC5 was never described to be present in the midpiece. Both experiments (CCDC146 and POC5 staining by ExM) shared the same secondary Ab and the midpiece signal was likely due to it.

Moreover, we now provide new images (figure 7C) in ExM on mouse sperm showing no staining in the midpiece and demonstrating that the punctuated signal is present all along the flagellum. Finally, we would like to underline that we now provide new IF results, using an anti-HA conjugated with alexafluor 488 and confirming the ExM results.

These points are now discussed lines 498-502 for acrosome and lines 503-511 for midpiece staining.

1. For intracellular localization of the CCDC146 in mouse sperm, the authors should provide clear negative control using WT sperm which do not carry the transgene.

This experiment was performed.

To avoid the issue of the non-specificity of secondary antibodies, we performed a new set of IF experiments using an HA Tag Alexa Fluor 488-conjugated Antibody (anti-HA-AF488-C Ab) on WT and HA-CCDC146 sperm. These results are now presented in figure 7 panel A (new). The specificity of the signal obtained with the anti-HA-AF488-C Ab on mouse spermatozoa was evaluated by performing a statistical study of the density of dots in the principal piece of the flagellum from HA-CCDC146 and WT sperm. These results are now presented in figure 7 panel B (new). This study was carried out by analyzing 58 WT spermatozoa and 65 CCDC146 spermatozoa coming from 3 WT and 3 KI males. We found a highly significant difference, with a p-value <0.0001, showing that the signal obtained on spermatozoa expressing the tagged protein is highly specific. We have added a paragraph in the MM section to describe the process of image analysis. We finally present new images obtained by ExM showing no staining in the midpiece (figure 7C new). Altogether, these results demonstrate unequivocally the presence of the protein in the flagellum.

1. Current imaging data do not clearly support the intracellular localization of the CCDC146. Although western blot imaging reveal that CCDC146 is detected from sperm flagella, this is crude approach. Thus, this reviewer highly recommends the authors provide more clear experimental evidence, such as immuno EM.

We provide now a WB comparing the presence of the protein in the flagellum and in the head fractions; see new figure 6. We show that CCDC146 is only present in the flagellum fraction; The detection of the band appeared very quickly at visualization and became very strong after few minutes, demonstrating that the protein is abundant in the flagella. It is important to note that epididymal sperm do not have centrioles and therefore this signal is not a centriolar signal. We also now provide new statistical analyses showing that the immuno-staining observed in the principal piece is very specific (Figure 7B). Altogether, these results demonstrate unequivocally the intracellular localization of CCDC146 in the flagellum. This point is now discussed lines 480-489

1. Although sarkosyl is known to dissociate tubulin, it is not well understood and accepted that the enhanced detection of CCDC146 by the detergent indicates its microtubule inner space. Sperm axoneme to carry microtubule is also wrapped peri-axonemal components with structural proteins, which are even not well solubilized by high concentration of the ionic detergent like SDS.

We agree with the reviewer that the solubilization of the protein by sarkozyl is not a proof of the presence of the protein inside microtubule. Taking into account this point, the MIP hypothesis was toned down and we now discuss alternative hypothesis concerning these results; See discussion lines 490-497

1. SEM image is not suitable to explain internal structure (line 317-323).

We agree with the reviewers and changes were made accordingly. See lines 354-357

Minor comments1. In main text, supplementary figures are cited "Supp Figure". And the corresponding legends are written in "Appendix - Figure". Please unify them.

Done Labelled now “Figure X-figure supplement Y”

1. Line 159, "exon 9/19" is not clear.

We have written now exons 9 and indicated earlier that the gene contains 19 exons

1. Line 188, "positive cells" are vague.

Positive was changed by “fluorescent”

1. Representative TUNEL assay image for knockout testes were not shown in Supp Figure 3B.

It was a mistake now Figure 2-figure supplement 2C

1. Please provide full description for "IF" and "AB" when described first.

Done

1. Line 262, It is unclear what is "main piece".

Changed to principal piece

1. Line 340, Although the "stage" information might be applicable, this is information for "seminiferous tubule" rather than "spermatid". This reviewer suggests to provide step information rather than stage information.

We agree with the reviewer that there was a confusion between “stage” and “step”. We change to step spermatids

1. Line 342, Step 1 is not correct in here.

OK corrected. now steps 13-15 spermatids

1. Line 803, "C." is duplicated.

Removed

1. Figure 3A, it will be good to mark the defective nuclei which are described in figure legends.

These cells are now indicated by white arrow heads

1. Figure 5, Please provide what MT stands for.

Now explained in the legend of figure 5

1. Figure 6. Author requires clear blot images for C. In addition, Panel B information is not correct. If the blot was performed using HA antibody, then how "WT" lane shows bands rather than "HA" bands?

The reviewer is correct. It was a mistake; The figure was recomposed and improved.

**Reviewer #2 (Recommendations For The Authors):**
Overall, editing oversights are present throughout the manuscript, which has made the review process quite difficult. Some repetitive figures can be removed to streamline to grasp the overall story easier. Some claims are not fully supported by evidence that need to tone down. Some figures not referenced in the main text need to be mentioned at least once.

All figures are now referenced in the text

Major comments:1. 163-164 - Please clarify the claim that there is going to be an absence of the protein or nonfunctional protein, especially for the patient with a deletion that could generate a truncated protein at two third size of the full-length protein. Similarly, 35% of the protein level is present for the patient with a nonsense mutation. Some in silico structural analysis or analysis of conserved domains would be beneficial to support these claims.

Both mutations are predicted to produce a premature stop codons: p.Arg362Ter and p.Arg704serfsTer7, leading either to the complete absence of the protein in case of non-sense mediated mRNA decay or to the production of a truncated protein missing almost two third or one fourth of the protein respectively. CCDC146 is very well conserved throughout evolution (Figure supplementary 1), including the 3’ end of the protein which contains a large coil-coil domain (Figure 1B). In view of the very high degree of conservation, it is most likely that the 3’ end of the protein, absent in both subjects, is critical for the CCDC146 function and hence that both mutations are deleterious. This explanation is now added to the discussion. see lines 439-448

1. 173, 423 - Please clearly state a rationale of your mouse model design (i.e., why a mouse model that recapitulate human mutation is not generated) as the truncations identified in human patients are located further towards the C-terminus, and it is not clear whether truncated proteins are present, and if so, they could still be functional. Basically, the current mouse model supports the causality of the human mutations.

This is an important question, which goes beyond the scope of this article, and raises the question of how to confirm the pathogenicity of mutations identified by high-throughput sequencing. The production of KO or KI animals is an important tool to help confirm one’ suspicions but the first element to take into consideration is the nature of the genetic data.

Here we had two patients with homozygous truncating variants. In human, it is well established that the presence of premature stop codons usually induces non-sense mediated mRNA decay (NMD), inducing the complete absence of the protein or a strong reduction in protein production. In the unlikely absence of NMD in our two patients, the identified variants would induce the production of proteins missing 60% and 30% of their C terminal part. Often (and it is particularly true for structural proteins) the production of abnormal proteins is more deleterious than the complete absence of the protein (and it is most likely the purpose of NMD, to limit the production of abnormal “toxic” proteins). For these reasons, to try to recapitulate the most likely consequences of the human variants, without risking obtaining an even more severe effect, we decided to introduce a stop codon in the first exon in order to remove the totality of the protein in the KO mice.

The second element is to interpret the phenotype of the KO animals. Here, the human sperm phenotype is perfectly recapitulated in the KO mice.

Overall, we have strong genetic arguments in human and the reproduction of the phenotype in KO mice confirming the pathogenicity of the variants identified in men.

This point is now discussed see lines 433-438

1. Figure 6A - the labelling is misleading as it seems to suggest that the specific cells were isolated from the testes for RT-PCR.

We have modified the labelling to avoid any confusion.

Figure 6B -Signal of HA-tag is shown in WT, not in transgenic. Please check the order of the labels. Figure 6C - This blot is NOT a publication-quality figure. The bands are very difficult to observe, especially in lane D18. Because it is one of the important data of this study, replacing this figure is a must.

The figure has been completely remade, including new results. See new figure 6. Figure 6C was suppressed.

1. Supplementary fig 6 is also not a publication-level figure, and the top part seems largely unnecessary (already in the figure legend).

The figure has been completely remade as well (now Figure 6-Figure Supplement 1).

1. 261/267- The conclusion that mitochondrial staining in the flagellum (in both mice and humans) is non-specific is not convincing. Supplementary fig 8 shows that the signal from secondary only IF possibly extends beyond the midpiece - but it is hard to determine as no mitochondrial-specific staining is present. Either need to tone down the conclusion or provide supporting experimental evidence.

First, to avoid the issue of the non-specificity of secondary antibodies, we performed a new set of IF experiments using an HA Tag Alexa Fluor 488-conjugated Antibody (anti-HA-AF488-C Ab) on WT and HA-CCDC146 sperm. These results are now presented in figure 7 panel A (new). The specificity of the signal obtained with the anti-HA-AF488-C Ab on mouse spermatozoa was evaluated by performing a statistical study of the density of dots in the principal piece of the flagellum from HA-CCDC146 and WT sperm. These results are now presented in figure 7 panel B (new). This study was carried out by analyzing 58 WT spermatozoa and 65 CCDC146 spermatozoa coming from 3 WT and 3 KI males. We found a highly significant difference, with a p-value <0.0001, showing that the signal obtained on spermatozoa expressing the tagged protein is highly specific. We have added a paragraph in the MM section to describe the process of image analysis. We finally present new images obtained by ExM showing no staining in the midpiece (figure 7C new). Altogether, these results demonstrate unequivocally the presence of the protein in the flagellum. These experiments are now described lines 271-279

Second, we provide new images of the signal obtained with secondary Abs only that shows more clearly that the secondary Ab gave a non-specific staining (Figure 10-Figure supplement 1B). This point is discussed lines 503-511

1. Figure 9 A - Please relate the white line to Fig. 9B label in X-axis. The information from Fig 9A+D and 9E+F are redundant. The main text nor the figure legends indicate why these specific two sperm were chosen for quantification and demonstrating the outcomes. One of them could be moved to supplementary information or removed, or the two could be combined.

As suggested by the reviewer, we have combined the two sperm to demonstrate that CCDC146 staining is mostly located on microtubule doublets. Moreover, the figure was recomposed to make it clearer.

Minor comments:All of the supplementary figures are referred to as Supp Fig X in the text, however, they are actually titled Appendix - Figure X. This needs to be consistent.

The figures are now referred as figure supplement x in both text and figures

Line 125 - edit spacing.

We think this issue (long internet link) will be curated later and more efficiently by the journal, during the step of formatting necessary for publication.

144 - With which to study with which we studied?

We made the change as suggested.

151 - Supp Fig 1 - the text says that the gene is highly transcribed in human and mouse testes, but the information in the figure states that the level in mouse tissues is "medium"

We have corrected this mistake in the text; See line 176

165 - The two mutations are most likely deleterious. Please specifically mention what analyses done to predict the deleterious nature to support these claims.

Both variants, c.1084C>T and c.2112del, are extremely rare in the general population with a reported allele frequency of 6.5x10-5 and 6.5x10-06 respectively in gnomAD v3. Moreover, these variants are annotated with a high impact on the protein structure (MoBiDiC prioritization algorithm (MPA) score = 10, DOI: 10.1016/j.jmoldx.2018.03.009) and predicted to induce each a premature termination codon, p.(Arg362Ter) and p.(Arg704SerfsTer7) respectively, leading to the production of a truncated protein. This information is now given line 164-169

196-200/Figure 4 - As serum starved cells/basal body (B) are not mentioned in the main text, as is, Fig 4A would be sufficient/is relevant to the text. Please make the text reflect the contents of the whole figure, or re/move to supplement.

We agree with the reviewer that the full description of the figure should be in the text. We added two sentences to describe figure 4B see lines 217-218.

224 - spermatozoa (plural) fits better here, not spermatozoon

OK changed accordingly

236 - According to the figure legend, 6B is only showing data from the epididymal sperm, not postnatal time points; should be referencing 6C. Alignment of Marker label

As indicated above, the figure has been completely remade, including new results. See new figure 6. Figure 6C was suppressed. The corresponding text was changed accordingly see lines 249-266

255-256 - Referenced figure 7B3, however, 7B3 only shows tubulin staining, so no CCDC146 can be observed. Did authors mean to reference fig 7B as a whole?

Sorry for this mistake. We agree and the text is now figure 8B6 (figure 7 and 8 were switched)

305 - "of tubules" - I presume it is meant to be microtubules?

Yes; The text was changed as suggested

317-321 - a diagram of HTCA would be useful here

We have added a reference where HTCA diagram is available see line 363. Moreover, a TEM view of HTCA is presented figure 12A

322/Fig 11A - an arrow denoting the damage might be useful, as A1 and A3 look similar. The size of the marker bar is missing. Please update the information on figure legend.

Concerning, the comparison between A1 and A3, the take home message is that there is a great variability in the morphological damages. This point is now underlined in the corresponding text. We updated the size of the marker bar as suggested (200 nm). See line 365-367

323 - Please mark where capitulum is in the figure

Capitulum was changed for nucleus

Since Fig 11B2 is not referenced in the main text, it does not seem to add anything to the data, and could be removed/moved to supplement.

We added a sentence to describe figure 11B2 line 370

342-343 - manchette in step I is not seen clearly - the figure needs to be annotated better. However, DPY19L2 is absent in step I in the KO, but the main text does not reflect that - why is that?

We do not understand the remark of the reviewer “manchette in step I is not seen clearly”. The figure shows clearly the manchette (red signal) in both WT and KO (Figure 13 D1/D2).

For steps 13-15 WT spermatids, the size of the manchette decreases and become undetectable. In KO spermatids, the shrinkage of the manchette is hampered and in contrast continue to expand (Figure 13D2). We also provide a new Figure 13-figure supplement 1 for other illustrations of very long manchettes and a statistical analysis. In the meantime, the acrosome is strongly remodeled, as shown in figure 16-new, with detached acrosome (panel H). This morphological defect may induce a loss of the DPY19L2 staining (Figure 13 D2 stage I-III). This explanation is now inserted in the text line 396399

Figure 15B and 15C only show KO, corresponding images from the WT should be present for comparison.

WT images are now provided in Figure 1-figure supplement 1 new

Figure 12 - Figure 12 - JM?.

JM was removed. It does not mean anything

Figure 12C and Supplementary Fig 10 - structures need to be labelled, as it is unclear what is where

Done

338 - text mentions step III, but only sperm from step VII are shown in Figure 13

As suggested by reviewer 3, we changed stage by step. The text was modified to take into account this remark see lines 388-396

360 - This is likely supposed to say Supp Figure 11E-G, not 13??

Yes, it is a mistake. Corrected

388 Typo "in a in a".

Yes, it is a mistake. Corrected

820 - Fig 3 legend - in KO spermatid nuclei were elongated - could this be labelled by arrows? I am not convinced this phenotype is that different from the WT.

In fact, the nuclei of elongating KO spermatids are elongated and also very thin, a shape not observed in the WT; We have added arrow heads and modified the text to indicate this point line 200.

836 - Figure 5 legend says that in yellow is centrin, but that is not true for 5A, where the figure shows labelling for y-tubulin (presumably, according to the figure itself).

We have modified the text of the legend to take into account the remark

837- 5A supposedly corresponds to synchronized HEK293T cells, but the reasoning behind using synchronized cells is not mentioned at all in the main text; furthermore, how this synchronization is achieved is not explained in materials and methods (serum starvation? Thymidine block?).

Yes, figure 5A was obtained with synchronized cells. We have added one paragraph in the MM section.For cell synchronization experiments, cells underwent S-phase blockade with thymidine (5 mM, SigmaAldrich) for 17 h followed by incubation in a control culture medium for 5 h, then a second blockade at the G2-M transition with nocodazole (200 nM, Sigma-Aldrich) for 12 h. Cells were then fixed with cold methanol at different times for IF labelling. See line 224 for changes made in the result section and lines 700-704 for changes made in the MM section.

845- figure legend says that the RT-PCR was done on CCDC146-HA tagged mice, but the main text does not reflect that.

We made changes and the description of the KI is now presented before (line 240) the RT-PCR experiment (line 257).

949 - it is likely supposed to say A2, not B1 (B1 does not exist in Fig 15)

Yes, it is a mistake. Corrected

971 - Appendix Fig 3 legend - I believe that the description for B and C are swapped.

Yes, it is a mistake. Corrected

Furthermore, some questions to address in A would be: Which cross sections were from which animal/points? How many per animal? Were they always in the same location?

Yes, we have a protocol for arranging and orienting all testes in the same way during the paraffin embedding phase. The cross-sections are therefore not taken at random, and we can compare sections from the same part of the testis. The number of animals was already indicated in the figure legend (see line 1128)

**Reviewer #3 (Recommendations For The Authors):**
1. There are a number of grammatical and orthographical errors in the text. Careful proofreading should be performed.

We have sent the manuscript to a professional proofreader

1. The author should also check for redundancies between the introduction and the discussion.

The discussion has modified to take into account reviewers’ remarks. Nevertheless, we did our best to avoid redundancies between introduction and discussion.

1. Can the authors provide a rationale why they have chosen to tag their gene with an HA tag for localisation? One would rather think of fluorescent proteins or a Halo tag.

Because the functional domains of the protein are unknown, adding a fluorescent protein of 24 KDa may interfere with both the localization and the function of CCDC146. For this reason, we choose a small tag of only 1.1 KDa, to limit as such as possible the risk of interfering with the structure of the protein. This rational is now indicated in the manuscript lines 251-254. It is worth to note, that the tagged-strain shows no sperm defect, demonstrating that the HA-tag does not interfere with CCDC146 function.

1. In the abstract, line 53, "provide evidence" is not the right term for something that is just suggestive. The term "suggests" would be more appropriate.

The text was modified to take into account this remark

1. Line 74: "genetic deficiency" sounds strange here, do the authors mean simply "mutation"?

Infertility may be due to several genetic deficiency such as chromosomal defects (XXY (Klinefelter syndrome)), microdeletion of the Y chromosome or mutations in a single gene. Therefore, mutation is too restrictive. Nevertheless, we modified the sentence which is now “…or a genetic disorder including chromosomal or single gene deficiencies”

1. Lines 163-164: the authors describe the mutations (premature stop mutations) and say that they could either lead to complete absence of the gene product, or the expression of a truncated protein. Did they test this, for example, with some immuno blot analyses?

As stated above, unfortunately, we were unable to verify the presence of RNA-decay in these patients for lack of biological material.

1. Line 184 and Fig 2E: the sperm head morphologies should be quantitatively assessed.

We provide now a full statistical analysis of the observed defects: see new panel in Figure 2 F

1. Fig 3: The annotation should be more precise - KO certainly means CDCC146-KO. The colours of the IH panels is different, which attracts attention but is clearly a colour-adjustment artefact. Colours should be adjusted for the panels to look comparable. It would be also helpful to add arrowheads into the figure to point at the phenotypes that are highlighted in the text.

We have added Ccdc146 KO in all figures. We have added arrow heads to point out the spermatids showing a thin and elongated nucleus. Concerning adjustment of colors, we attempted to make images of panel B comparable. See new figure 3.

1. Fig 6A: the authors use RT PCR to determine expression dynamics of their gene of interested, and use actin (apparently) as control. However, actin and CDCC146 expression levels follow the same trend. How is the interpreted?

The reviewer did not understand the figure. The orange bars do not correspond to actin expression and the grey bars to Ccdc146 expression but both bars represent the mRNA expression levels of Ccdc146 relative to Actb (orange) and Hprt (grey) expression in CCDC146-HA mouse pups’ testes. We tested two housekeeping genes as reference to be sure that our results were not distorted by an unstable expression of a housekeeping gene. We did not see significant difference between both house keeping genes. Actin was not used.

1. In line 235, the authors suggest posttranslational modifications of their protein as potential cause for a slightly different migration in SDS PAGE as predicted from the theoretical molecular weight. This is not necessarily the case, some proteins do migrate just differently as predicted.

We have changed the text accordingly and now provide alternative explanation for the slightly different migration. See lines 258-259

1. The annotation of Fig 6 panels is problematic. First, why do the authors write "Laemmli" as description of the gel? It would be more helpful to write what is loaded on the gel, such as "sperm". Second, in panels B and C it would be helpful to add the antibodies used. It is not clear why there is a signal in the WT lane of panel B, but not in the HA lane (supposing an anti-HA antibody is used: why has WT a specific HA band?). In panel C, it is not clear why the blot that has so beautifully shown a single band in panel B suddenly gives such a bad labelling. Can the authors explain this? Also, they cut off the blot, likely because to too much background, but this is bad practice as full blots should be shown. In the current state, the panel C does not allow any clear conclusion. To make it conclusive, it must be repeated.

Several mistakes were present in this figure. This figure was recomposed. The WB on testicular extract was suppressed and we now present a new WB allowing to compare the presence of CCDC146 in the flagella and head fractions from WT and HA-CCDC146 sperm. Using an anti-HA Ab, we demonstrate that in epididymal sperm the protein is localized in the flagella only. See new figure 6. The corresponding text was changed accordingly.

1. The authors have raised an HA-knockin mouse for CDCC146, which they explained by the unavailability of specific antibodies. However, in Fig 7, they use a CDCC146 antibody. Can they clarify?

The commercial Ab work for HUMAN CCDC146 but not for MOUSE CCDC146. We have added few words to make the situation clearer, we have added the following information “the commercial Ab works for human CCDC146 only”. See line 240

1. In Fig 7A (line 258), the authors hypothesise that they stain mitochondria - why not test this directly by co-staining with mitochondria markers?

We chose another solution to resolve this question:

To avoid the issue of the non-specificity of secondary antibodies, we performed a new set of IF experiments using an HA Tag Alexa Fluor 488-conjugated Antibody (anti-HA-AF488-C Ab) on WT and HA-CCDC146 sperm. These results are now presented in figure 7 panel A (new). The specificity of the signal obtained with the anti-HA-AF488-C Ab on mouse spermatozoa was evaluated by performing a statistical study of the density of dots in the principal piece of the flagellum from HA-CCDC146 and WT sperm. These results are now presented in figure 7 panel B (new). This study was carried out by analyzing 58 WT spermatozoa and 65 CCDC146 spermatozoa coming from 3 WT and 3 KI males. We found a highly significant difference, with a p-value <0.0001, showing that the signal obtained on spermatozoa expressing the tagged protein is highly specific. We have added a paragraph in the MM section to describe the process of image analysis. We finally present new images obtained by ExM showing no staining in the midpiece (figure 7C new). Altogether, these results demonstrate unequivocally the presence of the protein in the whole flagellum.

1. It seems that in both, Fig 7 and 8, the authors use expansion microscopy to localise CDCC146 in sperm tails. However, the staining differs substantially between the two figures. How is this explained?

In figure 8 we used the commercial Ab in human sperm, whereas in figure 7 we used the anti-HA Abs in mouse sperm. Because the antibodies do not target the same part of the CCDC146 protein (the tag is placed at the N-terminus of the protein, and the HPA020082 Ab targets the last 130 amino acids of the Cter), their accessibility to the antigenic site could be different. However, it is important to note that both antibodies target the flagellum. This explanation is now inserted see lines 304-312

1. Fig 8D and line 274: the authors do a fractionation, but only show the flagella fraction. Why?Showing all fractions of their experiment would have underpinned the specific enrichment of CDCC146 in the flagella fraction, which is what they aim to show. Actually, given the absence of control proteins, the fact that the band in the flagellar fraction appears to be weaker than in total sperm, one could even conclude that there is more CDCC146 in another (not analysed) fraction of this experiment. Thus, the experiment as it stands is incomplete and does not, as the authors claim, confirm the flagellar localisation of the protein.

We agree with the reviewer’s remark. We provide now new results showing both flagella and nuclei fractions in new figure 6A. This experiment is presented lines 253-256

1. Line 283, Fig 9D,F: The description of the microtubules in this experiment is not easy to understand. Do the authors mean to say that the labelling shows that the protein is associated with doublet microtubules, but not with the two central microtubules? They should try to find a clearer way to explain their result.

As suggested by reviewer 2, we have changed the figure to make it clearer. The text was changed accordingly. See new figure 9 and new corresponding legend lines 1006.

1. Fig 9G - how often could the authors observe this? Why is the axoneme frayed? Does this happen randomly, or did the authors apply a specific treatment?

Yes, it happens randomly during the fixation process.

1. Line 300 and Fig 10A - the authors talk about the 90-kDa band, but do say anything about what they think this band is representing.

We have now added the following sentence lines 340-342: “This band may correspond to proteolytic fragment of CCDC146, the solubilization of microtubules by sarkosyl may have made CCDC146 more accessible to endogenous proteases.”

1. Fig 11A, lines 321-322: the authors write that the connecting piece is severely damaged. This is not obvious for somebody who does not work in sperm. Perhaps the authors could add some arrow heads to point out the defects, and briefly describe them in the text.

We realized from your remark that our message was not clear. In fact, there is a great variability in the morphological damages of the HTCA. For instance, the HTCA of Ccdc146 KO sperm presented in figure 10A2 is quite normal, whereas that in figure 10A4 is completely distorted. This point is now underlined in the corresponding text. See lines 367-369

We also added the size of the marker bar (200 nm), which were missing in the figure’s legend.

1. Line 323: it will be important to name which tubulin antibody has been used to identify centrioles, as they are heavily posttranslationally modified.

The different types of anti-tubulin Abs are described in the corresponding figure’s legend

1. Fig 11B - phenotypes must be quantified to make these observations meaningful.

We agree that a quantification would improve the message. However, testicular sperm are obtained by enzymatic separation of spermatogenic cells and the number of testicular sperm are very low. Moreover, not all sperm are stained. Taking these two points into account, it seems to us that quantification could be difficult to analyze. For this reason, the quantification was not done; however, it is important to note that these defects were not observed in WT sperm, demonstrating that these defects are cased by the lack of CCDC146. We have added a sentence to underline this point; See lines 374-375

1. Line 329: Figure 12AB - is this a typo - should it read Figure 12B?

We have split the panel A in A1 and A2 and changed the text accordingly. See line 378

1. Why are there not wildtype controls in Fig 12B, C?

We provide now as Figure 12-figure supplement 1, a control image for fig 12B. For figure 12C, the emergence of the flagellum from the distal centriole in WT is already shown in Fig 12A1

1. Fig 13: the authors write that the manchette is "clearly longer and wider than in WT cells" (lines 342-343). How can they claim this without quantitative data?

We now provide a statistical analysis of the length of the manchette. See figure 13-figure supplement 1A. We also provide a new a new image illustrating the length of the manchette in Ccdc146 KO spermatids; See Figure 13-figure supplement 1B.